# High Fidelity Visualization of What Your Self-Supervised Representation Knows About

**Florian Bordes**                                                    *florian.bordes@umontreal.ca*
*Meta AI Research*
*Mila, Université de Montréal*

**Randall Balestriero**
*Meta AI Research*

**Pascal Vincent**
*Meta AI Research*
*Mila, Université de Montréal*
*Canada CIFAR AI Chair*

**Reviewed on OpenReview:** *https://openreview.net/forum?id=urfWb7VjmL*

## Abstract

Discovering what is learned by neural networks remains a challenge. In self-supervised learning, classification is the most common task used to evaluate how good a representation is. However, relying only on such downstream task can limit our understanding of what information is retained in the representation of a given input. In this work, we showcase the use of a Representation Conditional Diffusion Model (RCDM) to visualize in data space the representations learned by self-supervised models. The use of RCDM is motivated by its ability to generate high-quality samples —on par with state-of-the-art generative models— while ensuring that the representations of those samples are faithful i.e. close to the one used for conditioning. By using RCDM to analyze self-supervised models, we are able to clearly show visually that i) SSL (backbone) representation are *not* invariant to the data augmentations they were trained with – thus debunking an often restated but mistaken belief; ii) SSL post-projector embeddings appear indeed invariant to these data augmentation, along with many other data symmetries; iii) SSL representations appear *more robust to small adversarial perturbation* of their inputs than representations trained in a supervised manner; and iv) that SSL-trained representations exhibit an inherent structure that can be explored thanks to RCDM visualization and enables image manipulation. Code and trained models are available at `https://github.com/facebookresearch/RCDM`.

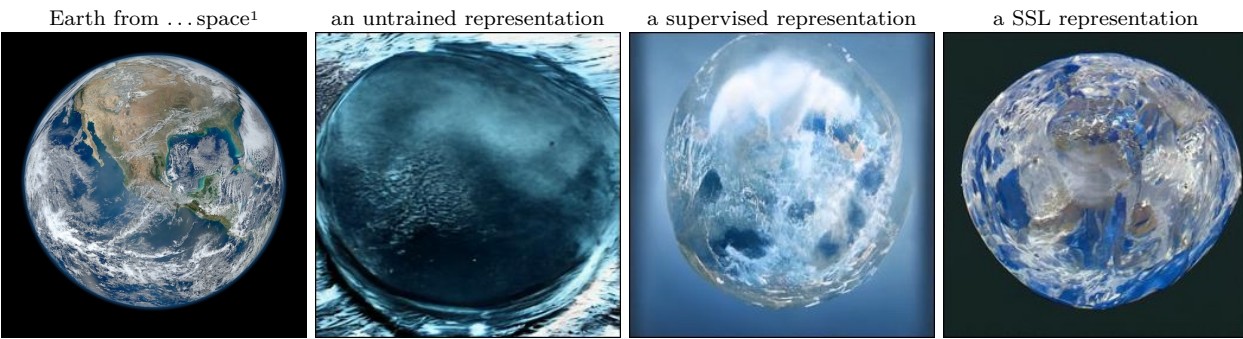

Earth from . . . space[1]    an untrained representation    a supervised representation    a SSL representation

[1]We use representations of the real picture of Earth on the left (source: NASA) as conditioning for RCDM. We show samples (resolution $256 \times 256$) in cases where the representations (2048-dimensions) were obtained respectively with a random initialized ResNet50, a supervised-trained one, and a SSL-trained one. More samples in Fig. 36.

## 1 Introduction

Approaches aimed at learning useful representations, from unlabeled data, have a long tradition in machine learning. These include probabilistic latent variable models and variants of auto-encoders (Ackley et al., 1985; Hinton et al., 2006; Salakhutdinov et al., 2007; Vincent et al., 2008; Kingma & Welling, 2014; Rezende et al., 2014), that are traditionally put under the broad umbrella term of *unsupervised learning* (Bengio et al., 2013). More recent approaches, under the term of *self-supervised learning* (SSL) have used various kinds of "pretext-tasks" to guide the learning of a useful representations. Filling-in-the-blanks tasks, proposed earlier in (Vincent et al., 2008; 2010), later proved remarkably successful in learning potent representations for natural language processing (Vaswani et al., 2017; Devlin et al., 2019). Pretext tasks for the image domain include solving Jigsaw-puzzles (Noroozi & Favaro, 2016), predicting rotations or affine transformations (Gidaris et al., 2018; Zhang et al., 2019b) or discriminating instances (Wu et al., 2018; van den Oord et al., 2018). The latest, most successful, modern family of SSL approaches for images (Misra & Maaten, 2020; Chen et al., 2020; Chen & He, 2020; He et al., 2020; Grill et al., 2020; Caron et al., 2020; 2021; Zbontar et al., 2021; Bardes et al., 2021), have two noteworthy characteristics that markedly distinguish them from traditional unsupervised-learning models such as autoencoder variants or GANs (Goodfellow et al., 2014): a) their training criteria are not based on any input-space reconstruction or generation, but instead depend only on the obtained distribution in the representation or embedding space b) they encourage invariance to explicitly provided input transformations a.k.a. data-augmentations, thus injecting important additional domain knowledge.

Despite their remarkable success in learning representations that perform well on downstream classification tasks, rivaling with supervised-trained models (Chen et al., 2020), much remains to be understood about SSL algorithms and the representations they learn. How do the particularities of different algorithms affect the representation learned and its usefulness? What information does the learned representation contain? Answering this question is the main focus of our work. Since SSL methods mostly rely on learning invariances to a specific set of handcrafted data augmentations, being able to evaluate these invariances will provide insight into how successful the training of the SSL criteria has been. It is also worth to be noted that recent SSL methods use a **projector** (usually a small MLP) on top of a **backbone** network (resnet50 or vit) during training, where the projector is usually discarded when using the model on downstream tasks. This projector trick is essential to get competitive performance on ImageNet i.e we often observe a performance boost of 10 to 30 percentage accuracy point when using the representation at the backbone level instead of the ones at the projector level on which SSL criteria are applied during training. Since SSL criteria are not applied at the backbone level, there remains a mystery regarding what the backbone does learn that make it better for classification than the projector. This is another question that we will be able to answer in this study.

Empirical analyses have so far attempted to analyse SSL algorithms almost exclusively through the limited lens of the numerical performance they achieve on downstream tasks such as classification. *Contrary to their older unsupervised learning cousins, modern SSL methods do not provide any direct way of mapping back the representation in image space, to allow* visualizing *it. The main goal of our work is thus to enable the visualization of representations learned by SSL methods, as a tool to improve our understanding.*

In our approach (Section 3), we propose to generate samples conditioned on a representation such that (i) the representation of those samples match the one used for conditioning, and (ii) the visual quality of the sample is as high as possible to maximize the preciseness of our visual understanding. For this, we employ a conditional generative model that (implicitly) models. For reasons that we will explain later, we opted for a conditional diffusion model, inspired by Dhariwal & Nichol (2021).

This paper's main contributions are:
- To demonstrate how recent diffusion models are suitable for conditioning on large vector representations such as SSL representations. The conditionally generated images, in addition to being high-quality are also highly representation-faithful i.e. they get encoded into a representation that closely matches the representation of the images used for the conditioning (Tab. 1b, Fig. 23).
- To showcase its potential usefulness for qualitatively analyzing SSL representations and embeddings (also in contrast with supervised representations), by shedding light on what information about the input image is or isn't retained in them.

Specifically, by repeatedly sampling from a same conditioning representation, one can observe which aspects are common to all samples, thus identifying what is encoded in the representation, while the aspects that vary greatly show what was *not retained* in the representation. We make the following observations: (i) SSL projector embeddings appear most invariant, followed by supervised-trained representation and last SSL representations[2] (Fig. 2). (ii) SSL-trained representations retain more detailed information on the content of the background and object style while supervised-trained representations appear oblivious to these (Fig. 3). (iii) despite the invariant training criteria, SSL representations appear to retain information on object scale, grayscale vs color, and color palette of the background, much like supervised representation (Fig. 3). (iv) Supervised representations appear more susceptible to adversarial attacks than SSL ones (Fig. 4,29). (v) We can explore and exploit structure inside SSL representations leading to meaningful manipulation of image content (such as splitting representation in foreground/background components to allow background substitution) (Fig, 5, 30, 31).

## 2 Related Work

**Visualization methods:** Many works (Erhan et al., 2009; Zeiler & Fergus, 2014; Simonyan et al., 2013; Mahendran & Vedaldi, 2015; Selvaraju et al., 2016; Smilkov et al., 2017; Olah et al., 2017) used gradient-based techniques to visualize what is learned by neural networks. Some of them maximize the activation of a specific neuron to visualize what is learned by this neuron, others offer visualization of what is learned at different layers by trying to "invert" neural networks. All of these use some form of regularization, constraint or prior to guide the optimization process towards *realistic* images. Dosovitskiy & Brox (2016) learn to map back a representation to the input space by using a Generative Adversarial Networks (Goodfellow et al., 2014) which is trained to reconstruct an input given a representation. Since the mapping is deterministic, they obtain only a single image with respect to a specific conditioning. In contrast, we use a stochastic mapping that allows us to visualize the diversity of the images associated to a specific representation. Nguyen et al. (2016) also use GANs but instead of trying to invert the entire vector of representation, they try to find which images (by using an optimization process in the latent space of the generator) maximize a specific neuron. The following work (Nguyen et al., 2017) demonstrates how using this conditional iterative optimization in the latent space of the generator lead to high quality conditional image generation. Finally Lučić et al. (2019) leverage SSL methods to create discrete cluster that are used as conditioning for a GAN. Our work focuses only on continuous representation vectors. More recently, Caron et al. (2021) used the attention mask of transformers to perform unsupervised object segmentation. By contrast, our method is not model dependent, we can plug any type of representation as conditioning for the diffusion model. Another possibility, explored in Zhao et al. (2021); Appalaraju et al. (2020); Ericsson et al. (2021), is to learn to invert the DN features through a Deep Image Prior (DIP) model. In short, given a mapping $f$ that produces a representation of interest, the DIP model $g_\theta$ learns $\min_\theta d(g_\theta(f(\boldsymbol{x})), \boldsymbol{x})$. However, as we will demonstrate in Figure 7 this solution not only requires to solve an optimization problem for each generated sample but also leads to low-quality generation.

**Generative models:** Several families of techniques have been developed as generative models, that can be trained on unlabeled data and then employed to generate images. These include auto-regressive models (Van Den Oord et al., 2016), variational auto-encoders (Kingma & Welling, 2014; Rezende et al., 2014), GANs (Goodfellow et al., 2014), autoregressive flow models (Kingma et al., 2016), and diffusion models (Sohl-Dickstein et al., 2015). Conditional versions are typically developed shortly after their unconditional versions (Mirza & Osindero, 2014; van den Oord et al., 2016). In principle one could envision training a conditional model with any of these techniques, to condition on an SSL or other representation for visualization purpose, as we are doing in this paper with a diffusion model. One fundamental challenge when conditioning on a rich representation such as the one produced by a SSL model, is that for a given conditioning $\boldsymbol{h}$ we will usually have available only a *single* corresponding input instance $\boldsymbol{x}$, By contrast a particularly successful model such as the conditional version of BigGAN (Brock et al., 2019) conditions on a categorical variable, the class label, that for each value of the conditioning has a large number of associated $\boldsymbol{x}$ data. One closely related work to ours is the recent work on Instance-Conditioned GANs (IC-GAN) of Casanova et al. (2021). Similar to us it also uses SSL or supervised representations as conditioning when training a conditional

---

[2]The representation that is produced by a Resnet50 backbone, before the projector.

generative model, here a GAN (Goodfellow et al., 2014), specifically a variant of BigGAN (Brock et al., 2019) or StyleGAN2 (Karras et al., 2020). However, the model is trained such that, from a specific representation, it learns to generate not only images that should map to this representation, but a much broader neighborhood of the training data. Specifically up to 50 training points that are its nearest neighbors in representation space. It remains to be seen whether such a GAN architecture could be trained successfully without resorting to a nearest neighbor set. IC-GAN is to be understood as a conditional generative model of an image's broad *neighborhood*, and the primary focus of the work was on developing a superior quality controllable generative model. By contrast we want to sample images that map as closely as possible to the original image in the representation space, as our focus is to build a tool to analyse SSL representations, to enable visualising what images correspond *precisely* to a representation. (See Fig. 23 for a comparison.) Lastly, a few approaches have focused on conditional generation to unravel the information encoded in representations of supervised models. In Shocher et al. (2020), a hierarchical LSGAN generator is trained with a class-conditional discriminator (Zhang et al., 2019a). While the main applications focused on inpainting and style-transfer, this allowed to visually quantify the increasing invariance of representations associated to deeper and deeper layers. This method however requires labels to train the generator. On the other hand, Nash et al. (2019) proposed to use an autoregressive model, in particular PixelCNN++ (Salimans et al., 2017), to specifically study the invariances that each layer of a DN inherits. In that case, the conditioning was incorporated by regressing a context vector to the generator biases. As far as we are aware, PixelCNN++ generator falls short on high-resolution images e.g. most papers focus on $32 \times 32$ Imagenet. Lastly, Rombach et al. (2020) proposes to learn a Variational Auto Encoder (VAE) that is combined with an invertible neural network (INN) whose role is to model the relation between the VAE latent space and the given representations. To allow for interpretable manipulation, a second invertible network (Esser et al., 2020) is trained using labels to disentangle the factors of variations present in the representation. By contrast we train end-to-end a single decoder to model the entire diversity of inputs that correspond to the conditioning representation, without imposing constraints of a structured prior or requiring labels for image manipulation.

## 3    High-Fidelity Conditioning with Diffusion Models

We base our work on the Ablated Diffusion Model (ADM) developed by Dhariwal & Nichol (2021) which uses a UNet architecture (Ronneberger et al., 2015) to learn the reverse diffusion process. Our conditional variant – called *Representation-Conditionned Diffusion Model* (RCDM) – is illustrated in Fig. 6b. To suitably condition on representation $h = f(x)$, we followed the technique used by Casanova et al. (2021) for IC-GAN which rely on conditional batch normalization (Dumoulin et al., 2017). More precisely, we replaced the Group Normalization layers of ADM by conditional batch normalization layers that take $h$ as conditioning. We also apply a fully connected layer to $h$ that reduces dimension to a vector of size 512. This vector is then given as input to multiple conditional batch normalization layers that are placed in each residual block of the diffusion model. An alternative conditioning method is to use the original conditioning method built inside ADM, and replace the embedding vector used for class labels by a linear layer that reduces dimension to a vector of the size of the time steps embedding. We didn't observe any differences in term of experimental results between those two methods as presented in Fig. 10. In this paper, we used the first conditioning method for most of the experiments in order to make a proper comparison with IC-GAN.

In contrast with Dhariwal & Nichol (2021) we don't use the input gradient of a classifier to bias the reversed diffusion process towards more probable images (classifier-guidance), nor do we use any label information for training our model – recall that our goal is building a visualization tool for SSL models that train on unlabeled data. Another drawback of classifier-guidance is the need to retrain a classifier on inputs generated by the diffusion process. Since training SSL models can be very costly, retraining them was not an option, thus we had to find a method that could use directly the representation of a pretrained model.

Our first experiments aim at evaluating the abilities of our model to generate realistic-looking images whose representations are close to the conditioning. To do so, we trained our Representation-Conditionned Diffusion Model (RCDM), conditioned on the 2048 dimensional representation given by a Resnet50 (He et al., 2016) trained with Dino (Caron et al., 2021) on ImageNet (Russakovsky et al., 2015). Then we compute the representations of a set of images from ImageNet validation data to condition the sampling from the trained RCDM. Fig. 1a shows it is able to sample images that are very close visually from the one that is used to get

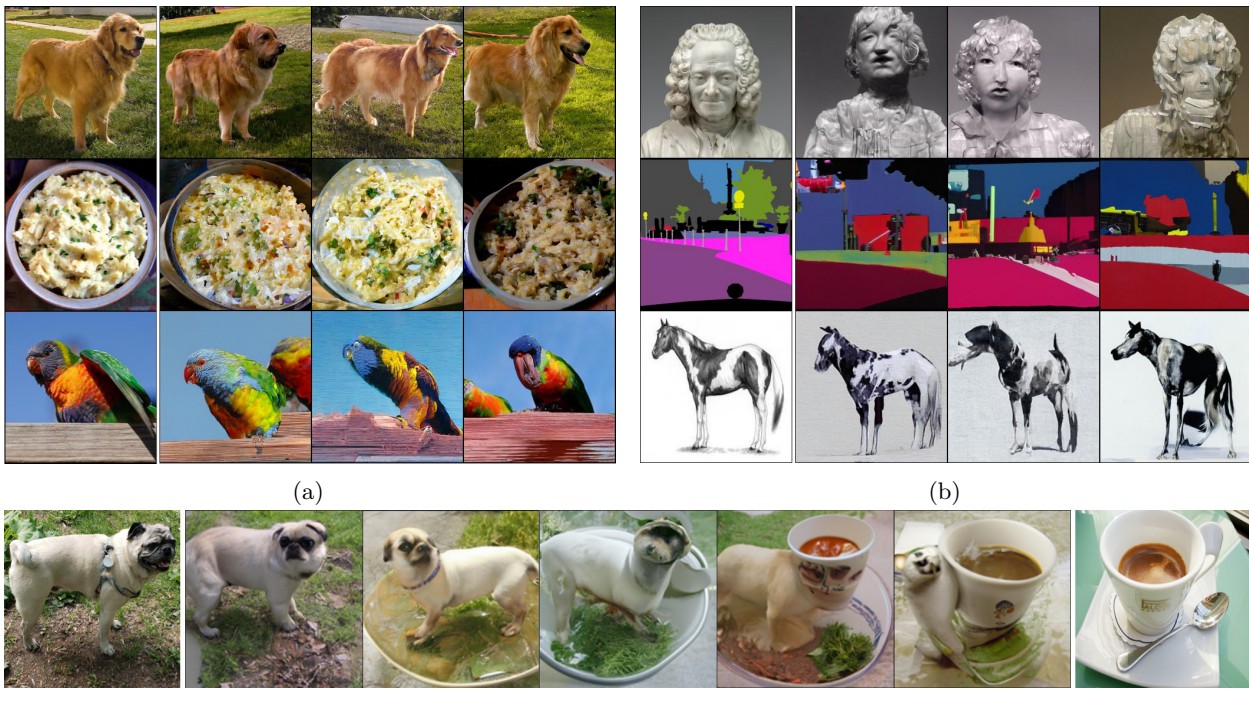

Figure 1: a) In-distribution conditional image generation. An image from ImageNet validation set (first column) is used to compute the representation output by a trained SSL model (Dino backbone). The representation is used as conditioning for the diffusion model. Resulting samples are shown in the subsequent columns (see Fig. 8). We observe that our conditional diffusion model produces samples that are very close to the original image. b) Out of distribution (OOD) conditioning. How well does RCDM generalize when conditioned on representations given by images from a different distribution? (here a WikiMedia Commons image, see Fig. 9 for more). Even with an OOD conditioning, the images produced by RCDM match some characteristics of the original image (which highlights that RCDM is not merely overfitting on ImageNet). c) Interpolation between two images from ImageNet validation data. We apply a linear interpolation between the SSL representation of the images in the first column and the representation of the images in the last column. We use the interpolated vector as conditioning for our model, that produces the samples that are showed in columns 2 to 6. Fig. 15 in appendix shows more sampled interpolation paths.

the conditioning. We also evaluated the generation abilities of our model on out of distribution data. Fig. 1b shows that our model is able to sample new views of an OOD image. We also quantitatively validate that the generated images' representations are close to the original image representation in Tab. 1b, Fig. 17, Fig. 18 and Fig. 19. We do so by verifying that the representation used as conditioning is the nearest neighbor of the representation of its generated sample.

This implies that there is much information kept inside the SSL representation so that the conditional generative model is able to reconstruct many characteristics of the original image. We also perform interpolations between two SSL representations in Fig. 1c. This shows that our model is able to produce interpretable images even for SSL representations that correspond to an unlikely mix of factors. Both the interpolation and OOD generation clearly show that the RCDM model is not merely outputting training set images that it could have memorized. This is also confirmed by Fig. 16 in the appendix that shows nearest neighbors of generated points.

The conditional diffusion model might also serve as a building block to hierarchically build an unconditional generative model. Any technique suitable for modeling and sampling the distribution of (lower dimensional) representations could be used. As this is not our primary goal in the present study, we experimented only with simple kernel density estimation (see appendix B). This allow us to quantify the quality of our generative process in an unconditional manner to fairly compare against state-of-the-art generative models such as ADM;

| Model | SimCLR Trunk | SimCLR Head | Dino Trunk | Dino Head | Barlow T. Trunk | Barlow T. Head | VicReg Trunk | VicReg Head |
|---|---|---|---|---|---|---|---|---|
| **Val acc.** | 69.1 % | 61.2 % | 74.8 % | 64.9 % | 72.6 % | 62.9 % | 72.3 % | 62.2 % |

Table a): ImageNet linear probe validation accuracy on representation given by various SSL models. We observe an accuracy gap between the linear probes at the trunk level and the linear probes trained at the head level of around 10 percentage point of accuracy.

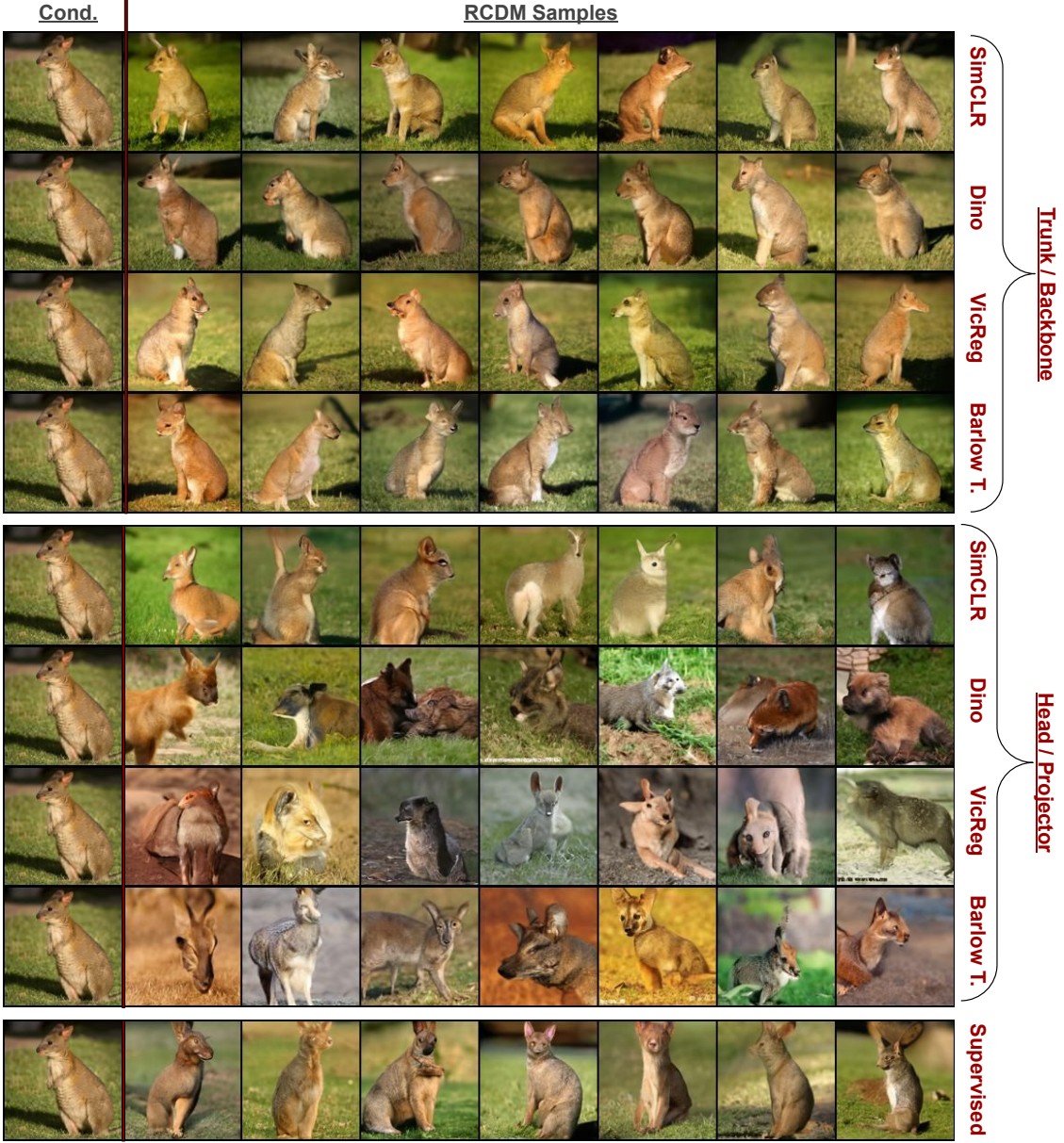

Figure 2: **What is encoded inside various representations?** First to fourth rows show RCDM samples conditioned on the usual resnet50 backbone representation (size 2048) while fifth to eigth rows show samples conditionned on the projector/head representation of various ssl models. (Note that a separate RCDM generative model was trained specifically for each representation). *Common/stable aspects* among a set of generated images reveal *what is encoded* in the conditioning representation. *Aspects that vary* show *what is not encoded* in the representation. We clearly see that the projector representation only keeps global information and not its context, contrary to the backbone representation. This indicates that invariances in SSL models are mostly achieved in the projector representation, not the backbone. Furthermore, it also confirms the linear classification results of Table a) which show that backbone representation are better for classifications since they contain more information about an input than the ones at the projector level. Additional comparisons provided in Fig. 24.

(a) We report results for ImageNet to show that our approach is reliable for generating images which look realistic. Since the focus of our work is not generative modelling but to showcase and encourage the use of such model for representation analysis, we only show results for one conditional generative models. For each method, we computed FID and IS with the same evaluation setup in Pytorch.

(b) For each encoder, we compute the rank and mean reciprocal rank (MRR) of the image used as conditioning within the closest set of neighbor in the representation space of the samples generated from the valid set (50K samples). A rank of one means that all of the generated samples for a given model have their representations matching the representation used as conditioning.

| Method | Res. | ↓FID | ↑IS |
|---|---|---|---|
| ADM (Dhariwal & Nichol, 2021) | 256 | 26.8 | 34.5 ± 1.8 |
| IC-GAN (Casanova et al., 2021)) | 256 | 20.8 | 51.3 ± 2.2 |
| IC-GAN (Casanova et al., 2021) w/ KDE* | 256 | 21.6 | 38.6 ± 1.1 |
| **RCDM w/ KDE* (ours)** | 256 | 19.0 | 51.9 ± 2.6 |

| Model | ↓Mean rank | ↑MRR |
|---|---|---|
| Dino (Caron et al., 2021) | 1.00 | 0.99 |
| Swav (Caron et al., 2020) | 1.01 | 0.99 |
| SimCLR (Chen et al., 2020) | 1.16 | 0.97 |
| Barlow T. (Zbontar et al., 2021)) | 1.00 | 0.99 |
| Supervised | 5.65 | 0.69 |

Table 1: a) Table of results on ImageNet. We compute the FID (Heusel et al., 2017) and IS (Salimans et al., 2016) on 10 000 samples generated by each models with 10 000 images from the validation set of ImageNet as reference. KDE* means that we used the unconditional representation sampling scheme based on KDE (Kernel Density Estimation) for conditioning IC-GAN instead of the method based on K-means introduces by Casanova et al. (2021). b) Table of ranks and mean reciprocal ranks for different encoders. This table show that RCDM is faithful to the conditioning by generating images which have their representations close to the original one.

we provide some generative model metrics in Tab. 1a along some samples in Fig. 8 to show that our method is competitive with the current literature, even in unconditional generation setting.

## 4 Visual Analysis of Representations Learned by Self-Supervised Model

The ability to view generated samples whose representations are very close in the representation space to that of a conditioning image can provide insights into what's hidden in such a representation, learned by self-supervised models. As demonstrated in the previous section, the samples that are generated with RCDM are really close visually to the image used as conditioning. This gives an important indication of how much is kept inside a SSL representation in general. However, it is also interesting to see how much this amount of "hidden" information varies depending on what specific SSL representation is being considered. To this end we train several RCDM on SSL representations given by VicReg (Bardes et al., 2021), Dino (Caron et al., 2021), Barlow Twins (Zbontar et al., 2021) and SimCLR (Chen et al., 2020). In many applications that use self-supervised models, the representation that is subsequently used is the one obtained at the level of the backbone of the ResNet50. Usually, the representation computed by the projector of the SSL-model (on which the SSL criterion is applied) is discarded because the results on many downstream tasks like classification is not as good as when using the backbone representation. However, since our goal is to visualize and better understand the differences between various SSL representations, we also trained RCDM on the representation given by the projector.

In Fig. 2 we condition all the RCDM with the image labelled as conditioning and sample 7 images for each model. We observe that the representation at the backbone level does not allow much variance in the generated samples. Even information about the pose and size of the animal is kept in the representation. In contrast, when looking at the samples generated by using representations at the projector level (also coined as head in the figure), we observe much more variance in the generated samples, which indicates that significant information about the input has been lost. These qualitative differences are correlated [3] with the quantitative experiment we made in Fig. 2 Table a) highlighting that when training a linear probe over the corresponding representations, the performances at the backbone level are better than the ones at the projector level.

### 4.1 Are Self-Supervised Representations Really Invariant to Data-Augmentations?

In Fig. 3, we apply specific transformations (augmentations) to a test image and we check whether the samples generated by the diffusion model change accordingly. We also compare with the behavior of a supervised model.

---

[3]This point is further demonstrated in Figure 12 in the Appendix.

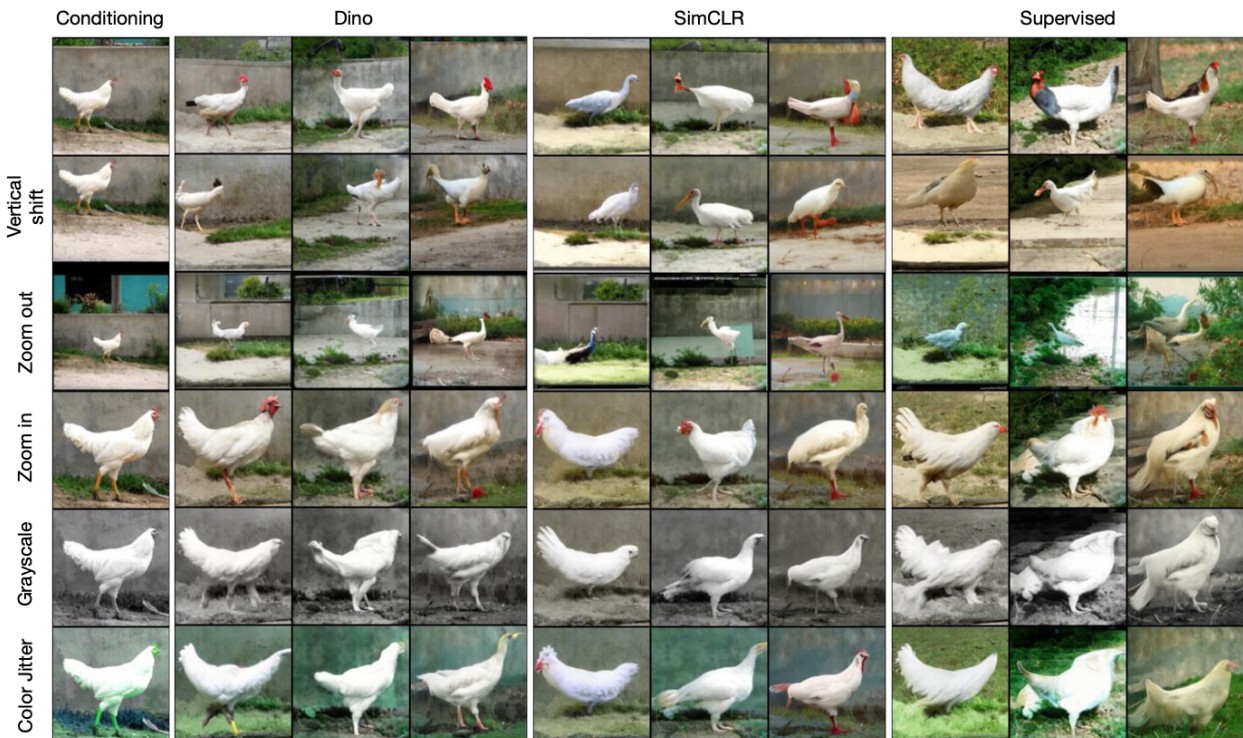

Figure 3: **Using our conditional generative model to gain insight about the invariance (or covariance) of representations with respect to several data augmentations.** On an original image (top left) we apply specific transformations (visible in the first column). For each transformed image, we compute the 2048-dimensional representation of a ResNet50 backbone trained with either Dino, SimCLR, or a fully supervised training. We then condition their corresponding RCDM on that representation to sample 3 images. We see that despite their invariant training criteria, the 2048 dimensional SSL representations appear to retain information on object scale, grayscale vs color, and color palette of the background, much like the supervised-trained representation. They do appear insensitive to vertical shifts. We also see that supervised representation constrain the appearance much less. Refer to Fig. 25 in Appendix for a comparison with using the lower dimensional projector-head embedding as the conditioning representation.

We note that despite their invariant training criteria, the 2048 dimensional SSL representations do retain information on object scale, grayscale status, and color palette of the background, much like the supervised representation. They do appear invariant to vertical shifts. In the Appendix, Fig. 25 applies the same transformations, but additionally compares using the 2048 representation with using the lower dimensional projector head embedding as the representation. There, we observe that the projector representation seems to encode object scale, but contrary to the 2048 representation, it appears to have gotten rid of grayscale-status and background color information. Currently, researchers need to use custom datasets (in which the factors of variation of a specific image are annotated) to verify how well the representations learned are invariant to those factors. We hope that RCDM will help researchers in self-supervised learning to alleviate this concern since our method is "plug and play" and can be easily used on any dataset with any type of representation.

## 4.2 Self-Supervised and Supervised Models See Adversarial Examples Differently

Since our model is able to "project back" representations to the manifold of realistic-looking images, we follow the same experimental protocol as Rombach et al. (2020) to visualize how adversarial examples affect the content of the representations, as seen through RCDM. We apply Fast Gradient Sign attacks (FGSM) (Goodfellow et al., 2015) over a given image and compute the representation associated to the attacked image. When using RCDM conditioned on the representation of the adversarial examples, we can visualize if the generated images still belong to the class of the attacked image or not. In Fig. 4 and 29, the adversarial

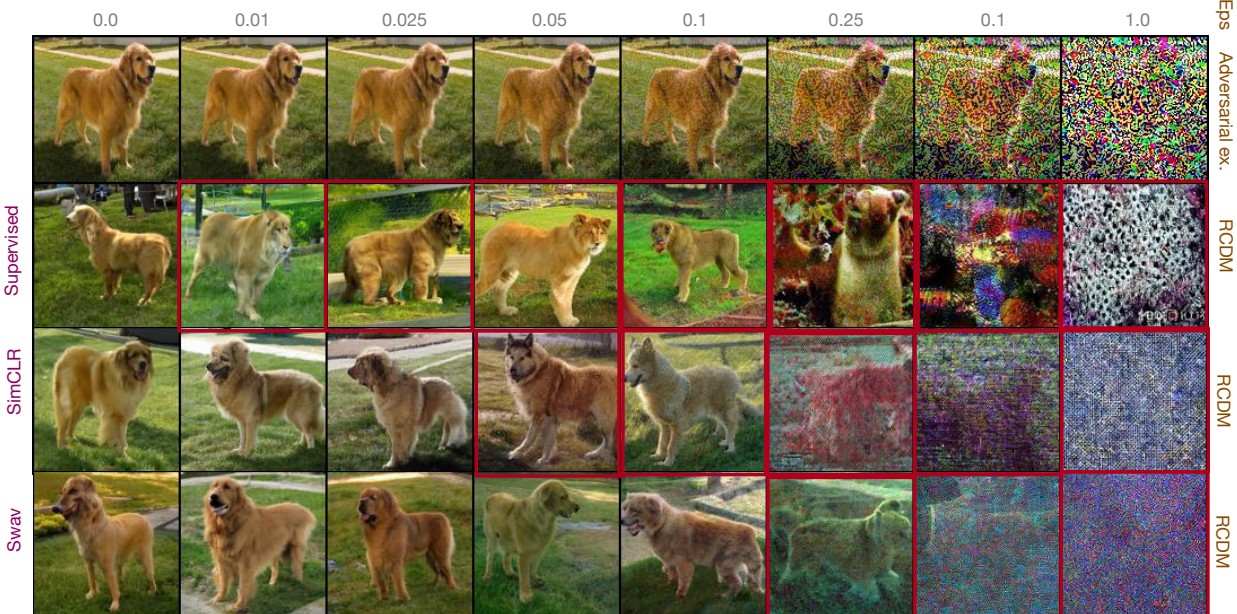

Figure 4: **Using RCDM to visualize the robustness of differently-trained representations to adversarial attacks.** We use Fast Gradient Sign to attack a given image (top-left corner) on different models with various values for the attack coefficient epsilon. In the first row, we only show the adversarial images obtained from a supervised encoder: refer to Fig. 29 in the Appendix to see the (similar looking) adversarial examples obtained for each model. In the following rows we show, for differently trained models, the RCDM "stochastic reconstructions" of the adversarially attacked images, from their ResNet-50 backbone representation. For an adversarial attack on a purely supervised model (second row), RCDM reconstructs an animal that belongs to another class, a lion in this case. Third and forth rows show what we obtain with ResNet50 that was pretrained with SimCLR or Swav in SSL fasion, with only their linear softmax output layer trained in a supervised manner. In contrast to the supervised model, with the SSL-trained models, RCDM stably reconstructs dogs from the representation of adversarially attacked inputs, even with quite larger values for epsilon. Images classified incorrectly by a trained linear probe are highlight with a red square.

attacks change the dog in the samples to a lion in the supervised setting whereas SSL methods doesn't seem to be impacted by the adversarial perturbations i.e the samples are still dogs until the adversarial attack became visible to the human eye.

## 4.3 Self-Supervised Representations Locally Encode background and Object on Different Dimensions

Experimental manipulation of representations can be useful to analyze to what degree specific dimensions of the representation can be associated with specific aspects or factors of variations of the data. In a self-supervised setting in which we don't have access to labelled data, it can be difficult to gain insight as to how the information about the data is encoded in the representation. We showcase a very simple and heuristic setup to remove the most common information in the representations within a set of the nearest neighbors of a specific example. We experimentally saw that the nearest neighbors of a given representation share often similar factors of variation. Having this information in mind, we investigate how many dimensions are shared in between this set of neighbors. Then, we mask the most common non-zero dimensions by setting them to zero and use RCDM to decode this masked representation. In Fig. 5, this simple manipulation visibly yields the removal of all information about the background and the dog, to only keep information about clothing (only one dog had clothes in the set of neighbors used to find the most common dimensions). Since the information about the dog and the background are removed, RCDM produces images of different clothes only. In the third and fourth row, instead of setting the most common dimensions to zeros, we set them to the value they have in other unclothed dog images. By using these new representations, RCDM is able to generate the corresponding dog with clothes. This setup works better with SSL methods, as supervised models learn to

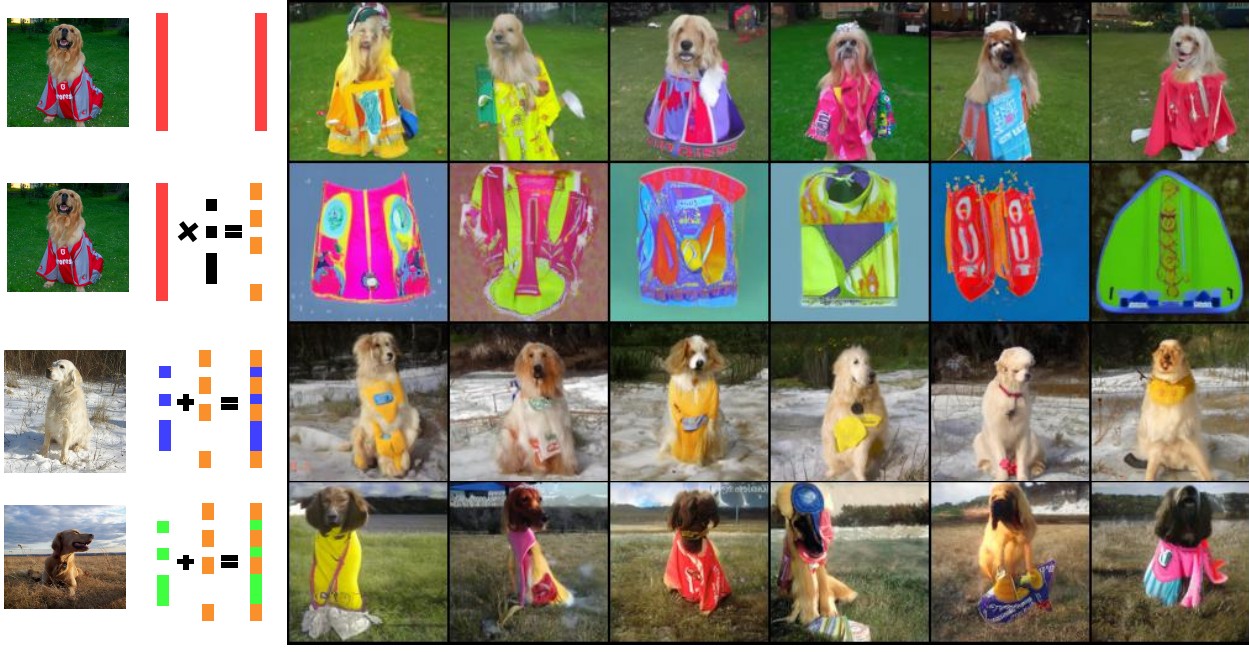

■ zero mask of most common indices where dim of representation is non zero
■ Least common dim of ■ where dim of representation is non zero

Figure 5: Visualization of direct manipulations in the representation space of a ResNet-50 backbone trained with SimCLR. In this experiment, we find the most common non-zero dimensions among the neighborhood (in representation space) of the image used as conditioning (top-left clothed dog). In the second row, we set these dimensions to zero and use RCDM to decode the thus masked representation. We see that RCDM produces a variety of clothes (but no dog): all information about the background and the dog has been removed. In the third and forth row, instead of setting these dimensions to zero, we set them to the value they have in the representation of the unclothed-dog image on the left. As we can see, the generated dog gets various clothes which were not present in the original image. Additional examples provided in Figure 30, 31.

remove from their representation most of the information that is not needed to predict class labels. We show a similar experiment for background removal and manipulation in Figure 30 in the Appendix.

## 5 Conclusion

Most of the Self-Supervised Learning literature uses downstream tasks that require labeled data to measure how good the learned representation is and to quantify its invariance to specific data-augmentations. However one cannot in this way see the entirety of what is retained in a representation, beyond testing for specific invariances known beforehand, or predicting specific labeled factors, for a limited (and costly to acquire) set of labels. Yet, through conditional generation, all the stable information can be revealed and discerned from visual inspection of the samples. We showcased how to use a simple conditional generative model (RCDM) to visualize representations, enabling the visual analysis of what information is contained in a self-supervised representation, without the need of any labelled data. After verifying that our conditional generative model produces high-quality samples (attested qualitatively and by FID scores) and representation-faithful samples, we turned to exploring representations obtained under different frameworks. Our findings clearly separate supervised from SSL models along a variety of aspects: their respective invariances – or lack thereof – to specific image transformations, the discovery of exploitable structure in the representation's dimensions, and their differing sensitivity to adversarial noise.

## 6 Reproducibility statement

The data and images in this paper were only used for the sole purpose of exchanging reproducible research results with the academic community.

Our results should be easily reproducible as:

- RCDM, is based on the same code as Dhariwal & Nichol (2021) (`https://github.com/openai/guided-diffusion`) and uses the same hyper-parameters (See Appendix I of Dhariwal & Nichol (2021) for details about the hyper-parameters).

- To obtain our conditional RCDM, one just needs to replace the GroupNormalization layers in that architecture by a conditional batch normalization layer of Brock et al. (2019) (using the code from `https://github.com/ajbrock/BigGAN-PyTorch`).

- The self-supervised pretrained models we used to extract the conditioning representations were obtained from the model-zoo of VISSL (Goyal et al., 2021) (code from `https://github.com/facebookresearch/vissl`).

- The unconditonal sampling process is straightforward, as explained in Appendix B.

- We are working on cleaning and preparing to release any remaining code glue to easily reproduce the results in this paper.

## 7 Broader impact statement

Our work aims to promote the use of conditional generative models to project back in image space the internal representation learned by latest and future techniques to train deep artificial neural networks – in order to better understand their inner workings. Such improved understanding through qualitative visualizations, in complement with quantitative metrics, is expected to foster the development of more robust and reliable neural network algorithms. Controlled generation of realistic images is not the goal and focus of this work, as we merely use it as a tool for scientific understanding. Yet conditional generative models have already been and will likely continue to be used and improved to generate fake images, including of synthesized imaginary situations and people, that we expect will be increasingly realistic and hard to impossible to distinguish from real photographs. Such technology will be usable for positive creative pursuits, as well as for voluntarily misleading portrayals of fakes passed as truths and facts.

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

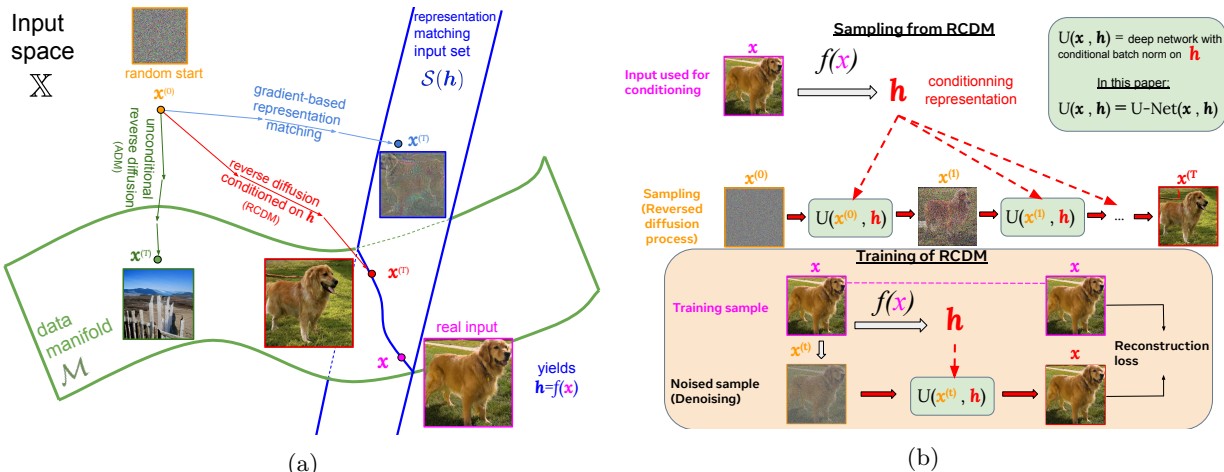

Figure 6: (a) Illustration of considered image generation methods. A real input $\boldsymbol{x}$ yields representation $\boldsymbol{h}$. All methods start from a random noise image $\boldsymbol{x}^{(0)}$. Gradient-based representation matching (light blue arrows) will move it towards $\mathcal{S}(\boldsymbol{h})$ i.e. until its representation matches $\boldsymbol{h}$, but won't land on the data-manifold $\mathcal{M}$. Unconditional reverse diffusion (ADM model, green arrows) will move it towards the data manifold. Our representation-conditioned diffusion model (RCDM, red arrows) will move it towards $\mathcal{M} \cap \mathcal{S}(\boldsymbol{h})$, yielding a different natural-looking image with the same given representation. (b) Representation-Conditionned Diffusion Model (RCDM). From a diffusion process that progressively corrupts an image, the model learns the reverse process by predicting the noise that it should remove at each step. We also add as conditioning a vector $\boldsymbol{h}$, which is the representation given by a SSL or supervised model for a given image $\boldsymbol{x}$. Thus, the network is trained explicitly to denoise towards a specific example given the corresponding conditioning. The diffusion model used is the same as the one presented by Dhariwal & Nichol (2021) with the exception of the conditioning on the representations.

## A    Conditional and super-resolution sampling with RCDM

As presented in the main text, we introduce RCDM to generate samples that preserved well the semantics of the images used for the conditioning. The training of the model is simple and presented in Figure 6b. We show in Figure 8 additional samples of RCDM when conditioning on the SSL representation of ImageNet validation set images (which were never used for training). We observe that the information hidden in the SSL representation is so rich that RCDM is almost able to reconstruct entirely the image used for conditioning. To further evaluate the abilities of this model, we present in Figure 9 a similar experiment except that we use out of distribution images as conditioning. We used cell images from microscope and a photo of a status (Both from Wikimedia Commons), sketch and cartoons from PACS (Li et al., 2017), image segmentation from Cityscape (Cordts et al., 2016) and an image of the Earth by NASA. Even in the OOD scenario, RCDM is able to generate images that share common features to the one used as conditioning because of the richness of ssl representations. However, if the images used as OOD are too far from the training distribution, which is the case when using image segmentation mask from Cityscapes, the model will have more difficulty to reconstruct the images used as conditioning. To investigate if this failure is due to the SSL network used to produced the conditioning, we run the experiment in Figure 11 in which we kept the same SSL model with an RCDM trained on ImageNet and another one on Cityscape. We observe that when using Cityscapes segmentation mask as conditioning with an RCDM trained on Cityscapes segmentation mask, despite using a SSL model trained only on ImageNet, RCDM is able to reconstruct the conditioning very faithfully. This mean that the failure mode observed in OOD are mostly due to the visualization model (RCDM) and not due to the representation used for conditioning.

In those examples we used conditional batch-normalization (which is the same technique as used by Casanova et al. (2021)). However one can also use the sampling technique built-in in the ADM model of Dhariwal & Nichol (2021). Instead of using an embedding layer that take discrete representation, we can use a linear

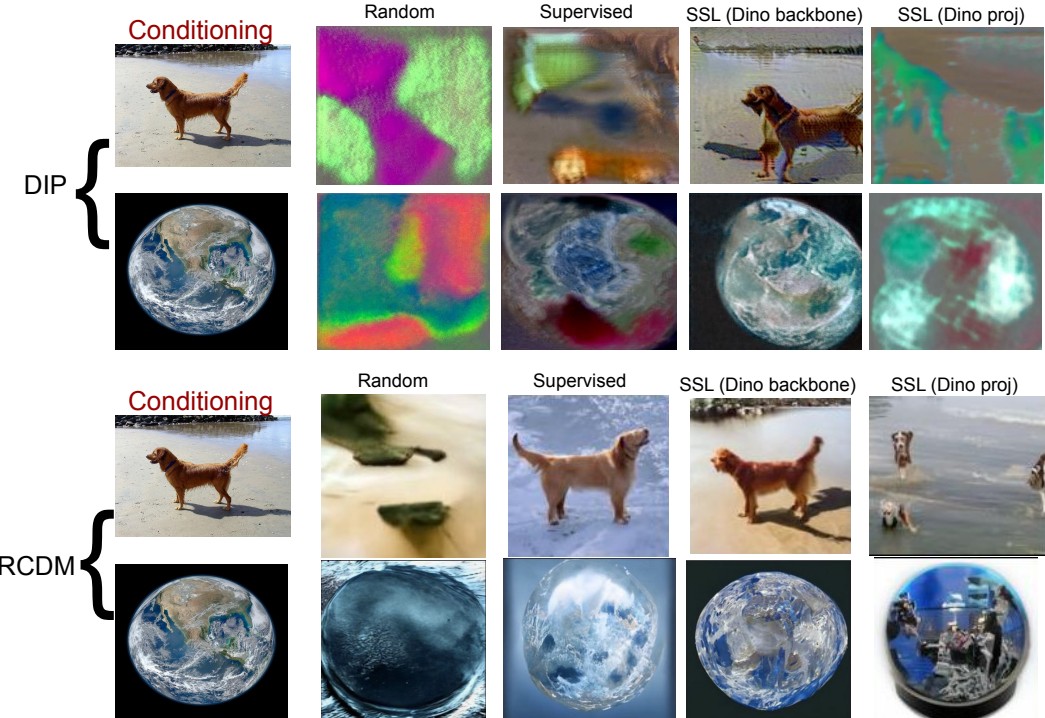

Figure 7: We compare RCDM with the approach of Zhao et al. (2021) that use Deep Image Prior (DIP (Ulyanov et al., 2018)) to visualize the features learn by Self-supervised models. We run this experiment by using as conditioning an In-Distribution image (with the dog from ImageNet validation set) and an Out-Of-Distribution image with an image from the Earth (Source: NASA). For both methods, we compare with representations extracted at the backbone level of a Resnet (after average pooling) from a random initialized network, a supervised network and a network trained with SSL (Dino). We also use the representation extracted at the projector level for Dino. We observe that the samples we obtained with RCDM have a better quality than the ones generated with DIP and are also more insightful about the properties of the representations. In this instance, we clearly see with RCDM that the supervised representation is invariant to background which is something more difficult to assess with DIP.

layer to map a representation to the dimension of the time steps embedding and add it along the time step conditioning. A comparison with these two conditioning methods is shown in Figure 10.

We also use the super-resolution model presented by Dhariwal & Nichol (2021) to generate images of higher resolutions. In Figure 13, we use the small images on the top of the bigger images as conditioning for a RCDM trained on images of size 128x128. Then, we feed the 128x128 samples into the super-resolution model of Dhariwal & Nichol (2021) to get images of size 512x512. Since the model of Dhariwal & Nichol (2021) is conditional and need labels, we used a random label when upsampling from RCDM. Despite using the "wrong" label, the high resolution samples are still very close to the conditioning. This show that RCDM can be used jointly with a super-resolution model to sample high fidelity images in the close neighborhood of the conditioning.

To verify how well our model can produce realistic samples from different combinations of representations, we take two images from which we compute their representations and perform a linear interpolation between those. This give us new vectors of representation that can be used as conditioning for RCDM. We can see on Figure 14 and Figure 15 that RCDM is able to generate samples that contains the semantic characteristics of both images.

Finally, in Figure 16, we search the nearest neighbors of a series of samples in the ImageNet training set. As demonstrated by Figure 16, RCDM samples images that are new and far enough from images belonging to the training set of ImageNet.

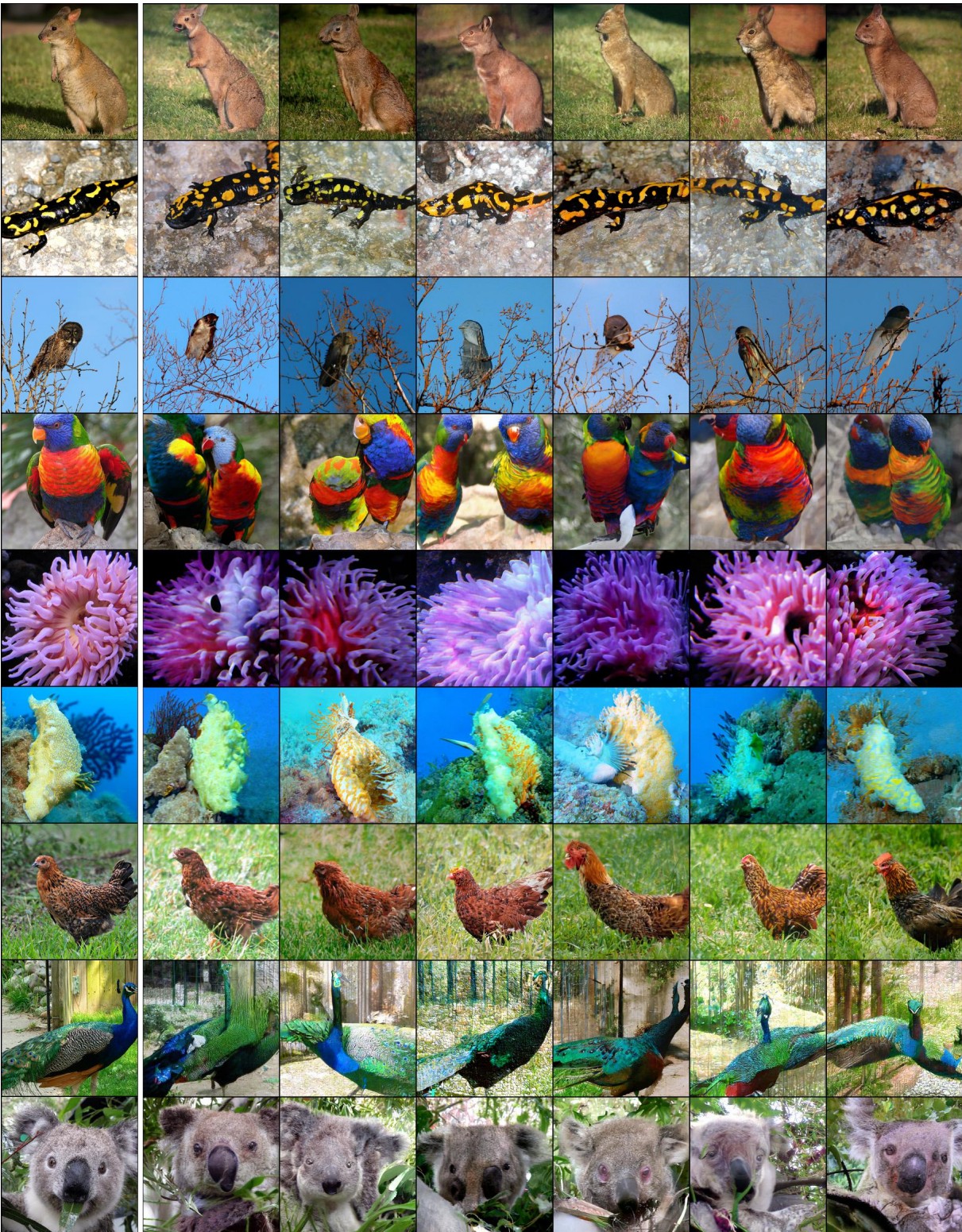

Figure 8: Generated samples from RCDM on 256x256 images trained with representations produced by Dino. We put on the first column the images that are used to compute the representation conditioning. On the following column, we can see the samples generated by RCDM. It is worth to denote our generated samples are qualitatively close to the original image.

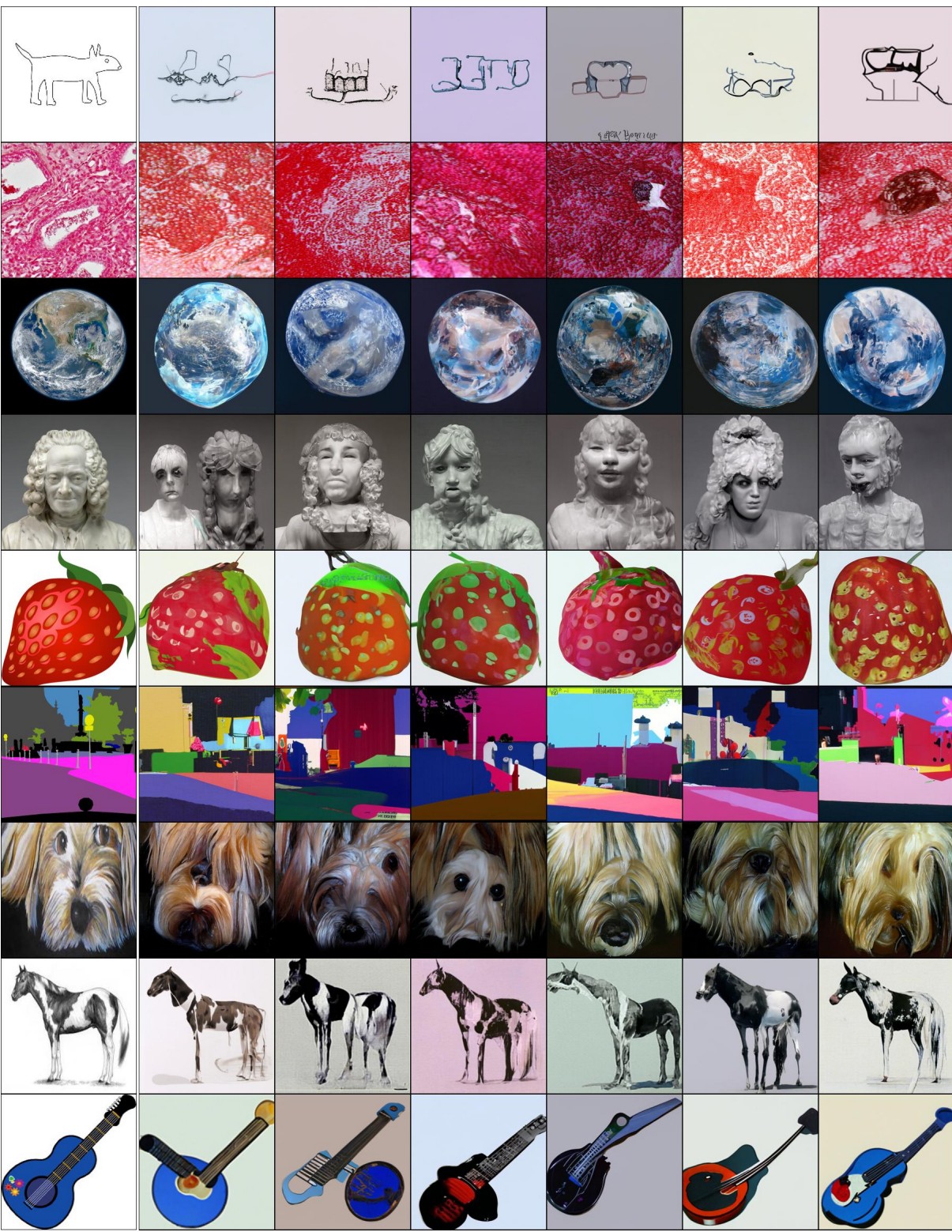

Figure 9: Generated samples from RCDM model on 256x256 images trained with representations produced by Dino on Out of Distribution data. We put on the first column the images that are used to compute the representation. On the following column, we can see the samples generated by RCDM. It is worth to denote our generated sample are close to the original image. The images used for the conditioning are from Wikimedia Commons, Cityscapes (Cordts et al., 2016), PACS (Li et al., 2017) and the image of earth from NASA.

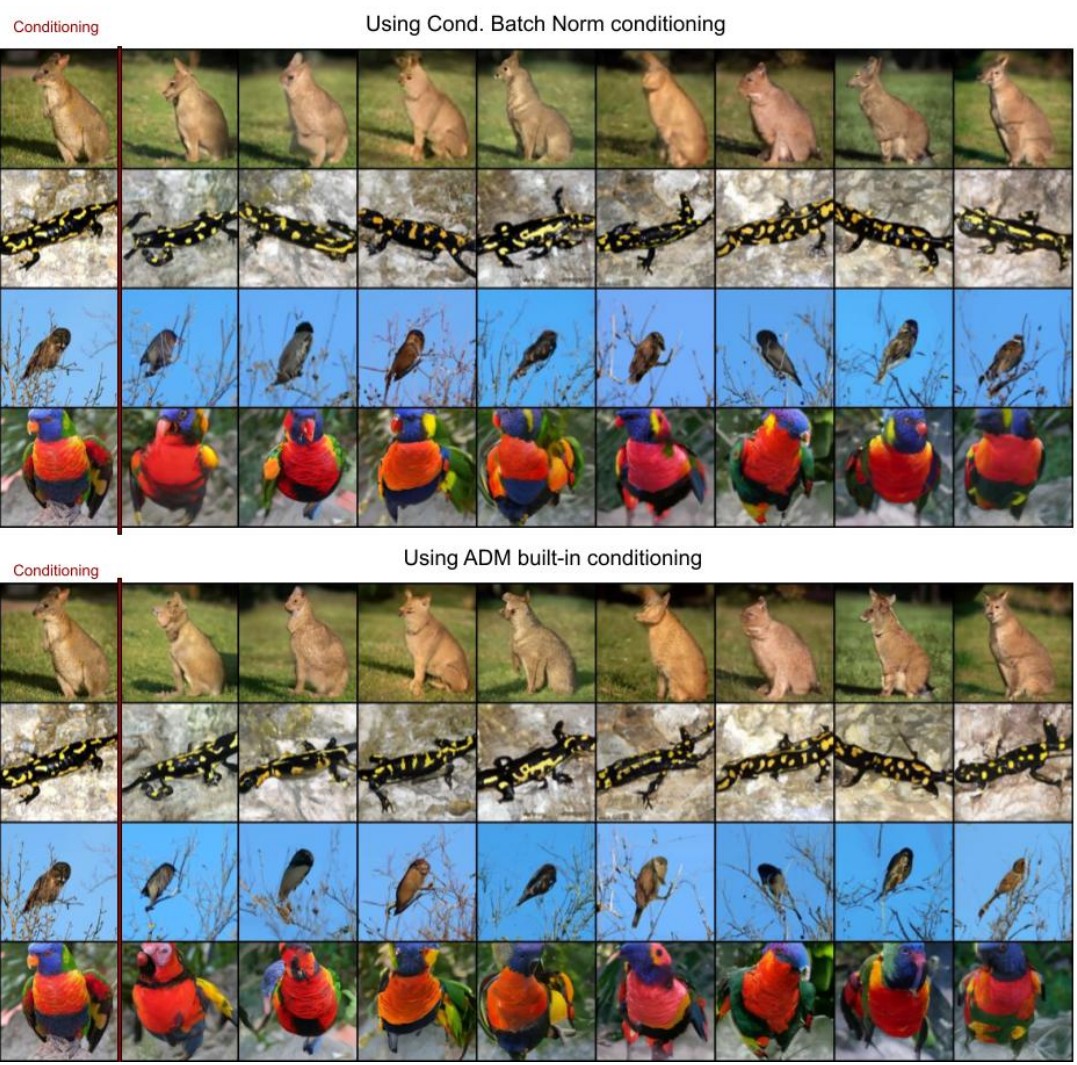

Figure 10: Comparison between conditioning RCDM with batch normalization and the built-in conditioning mechanism offered by ADM. For this example, we took the representation backbone of dino trained on ImageNet with resolution 128x128. There doesn't seem to be any significant differences between both methods.

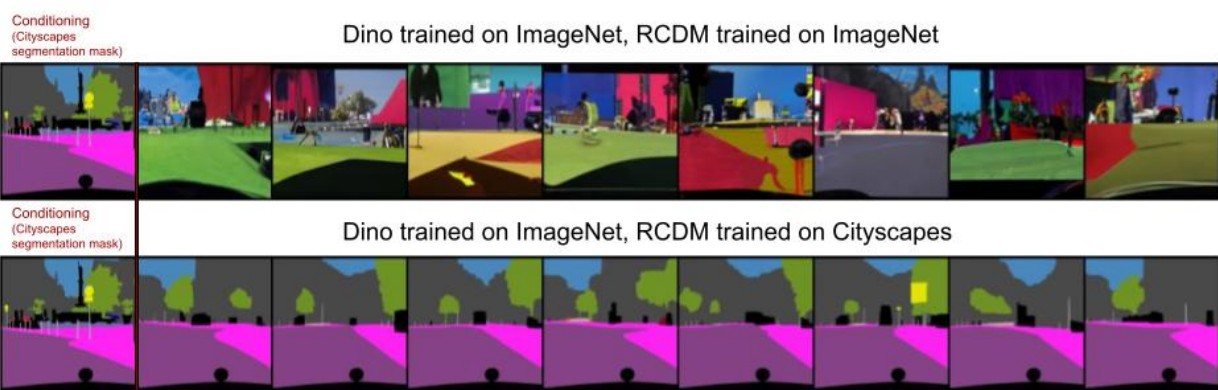

Figure 11: We perform this experiment to see if the failures mode on OOD, especially when conditioning on segmentation mask of Cityscapes, are due to the self-supervised representations not containing enough information to reconstruct the image, or are due to RCDM not being able to reconstruct OOD images. On the first line, we show the samples generated by an RCDM trained on ImageNet with the self-supervised representation of Dino that was also trained on ImageNet. On the second line, we show the samples generated by an RCDM trained on the segmentation masks of cityscapes that use the same self-supervised model of Dino that was trained on ImageNet. We can clearly see that despite using a SSL model trained on ImageNet, when RCDM is trained on CityScapes, the reconstruction almost match the original conditioning. Hence, one should train or fine-tune RCDM on any target dataset to then use it to sample representation conditioned images from a (frozen) pre-trained model.

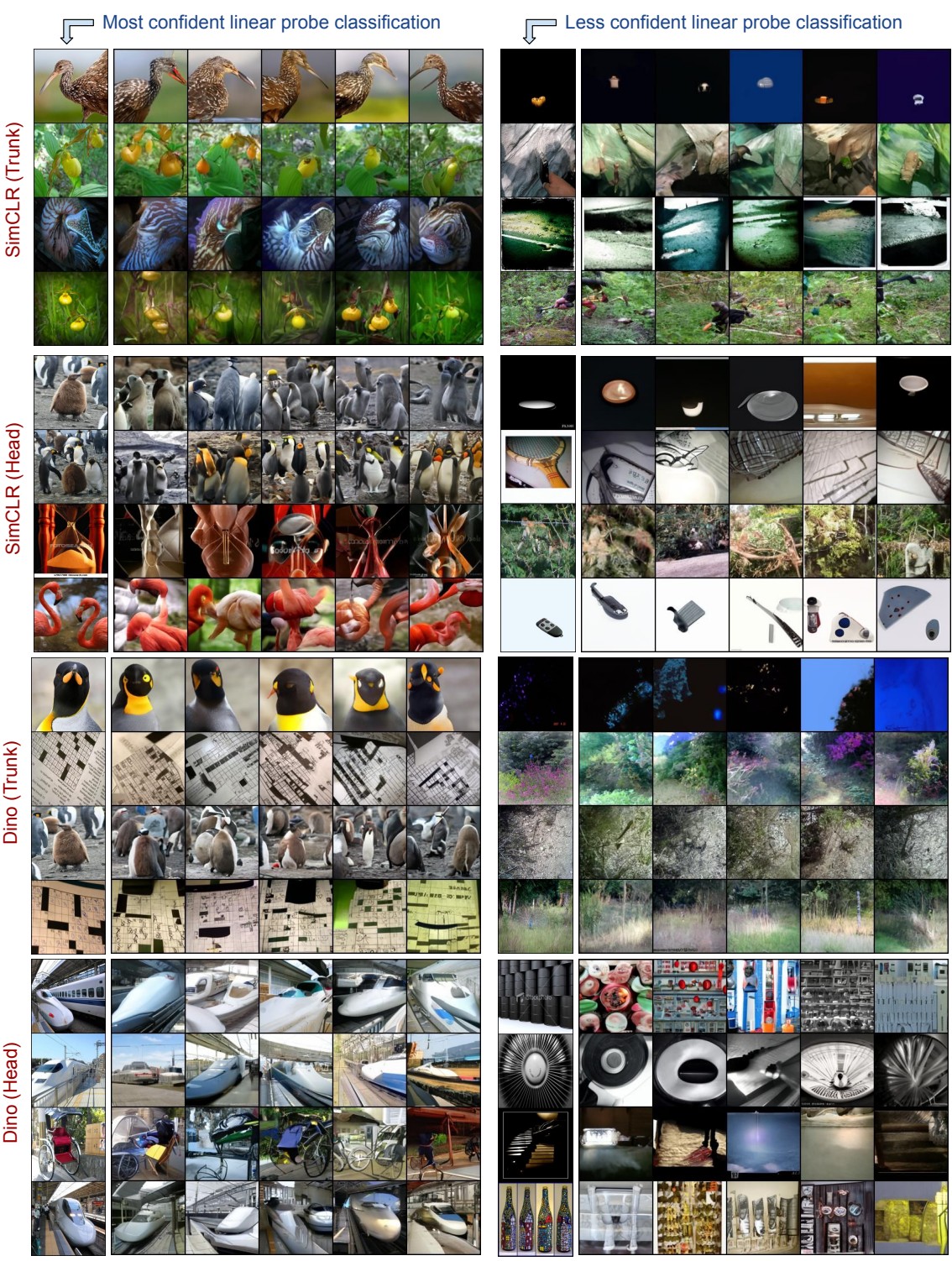

a) RCDM samples using a conditioned-representation having the highest probabilities under a linear classifier.

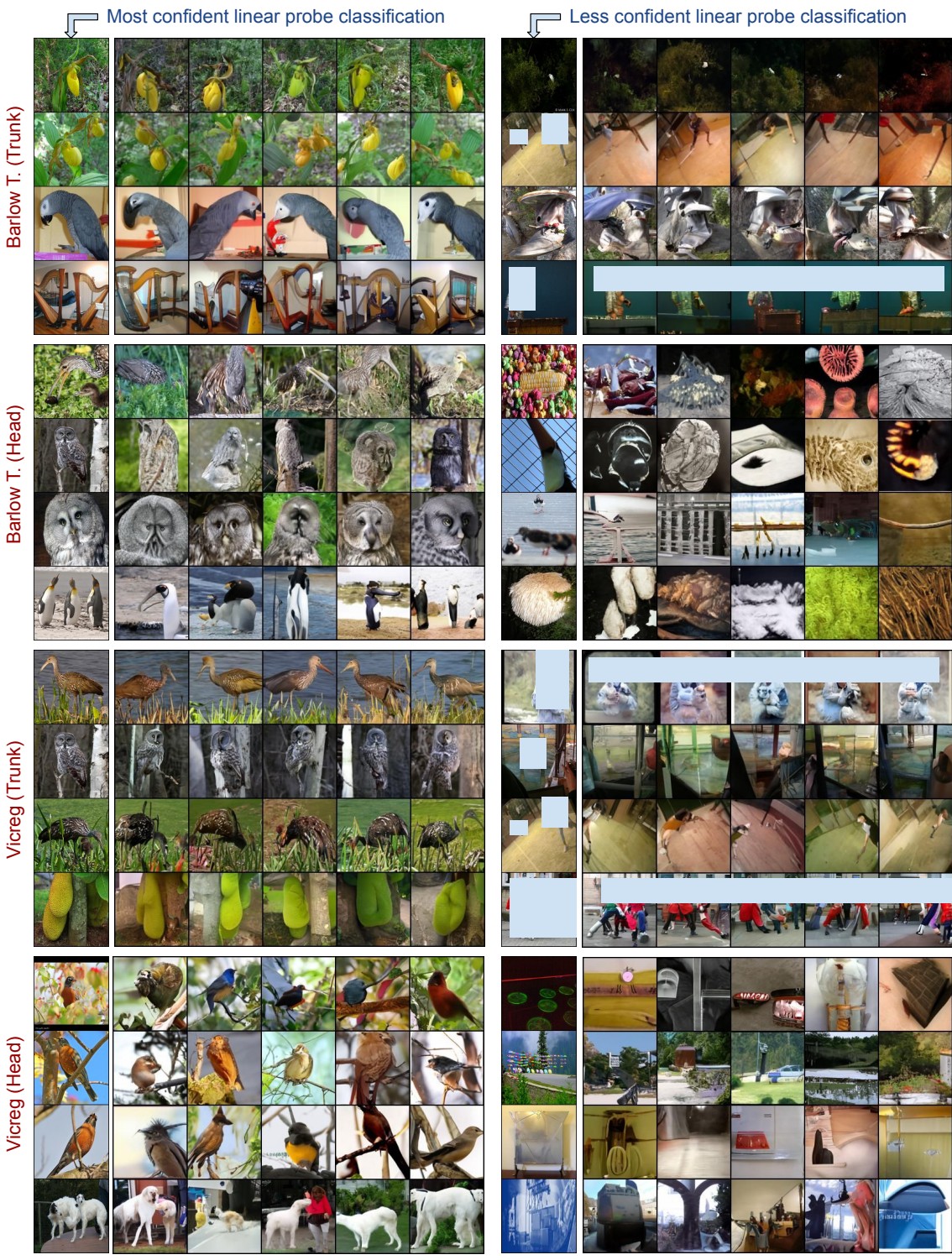

b) RCDM samples on conditioning that have the lowest probabilities under a linear classifier.

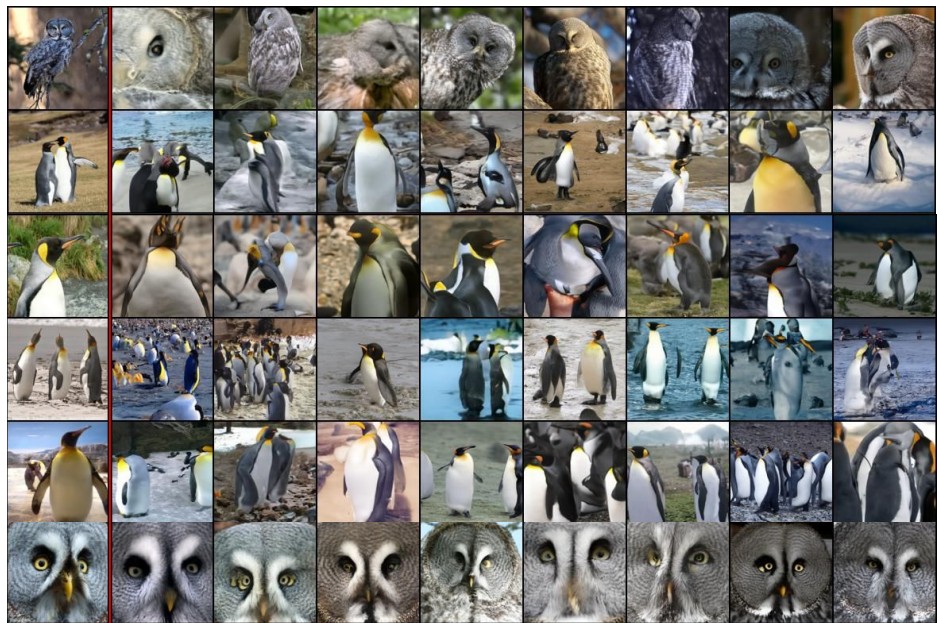

c) More RCDM samples on conditioning that have the highest probabilities under a linear classifier.

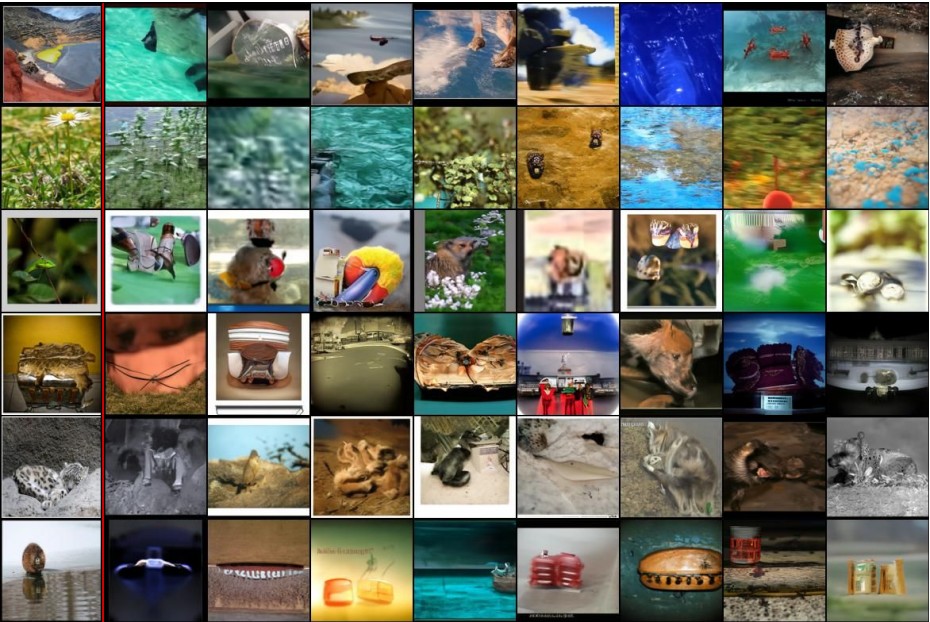

d) More RCDM samples on conditioning that have the lowest probabilities under a linear classifier.

Figure 12: We trained a linear probe for classification by using the representation (at the backbone and projector level) given by various SSL models. Then we find, among the ImageNet validation set, the images that yield the highest softmax probability under this linear probe and use RCDM to generate samples with respect to their representations. We also find the images that yield the lowest softmax probabilities. On the first column and seventh column in a) and b), we present the images used as conditioning (thus, the ones with highest and lowest softmax probabilities). In the following columns we show the corresponding RCDM samples. We observe that all generated images belong to the same class as the one that was used as conditioning when looking at the most confident representation . However when looking at the least confident representation, the generated samples does not seem to belong to a unique class. This experiment shows that the uncertainty in the predictions of a downstream classification task can also be predicted by simple visual inspection of samples produces by RCDM.

## B    A hierarchical diffusion model for unconditional generation

We provided a novel and conditional generative model based on a given latent representation e.g. from a SSL embedding, and a diffusion model. This allows visualizing and thus provides insight regarding what is or isn't encoded in a particular representations. We can go one step further and augment this conditional model with an unconditional one that can generate those representations. This will provide us with the ability to generate new samples without the need to condition on a given input. As a by-product, it will allow us to quantify the quality of our generative process in an unconditional manner to fairly compare against state-of-the-art generative models.

We shall recall that our goal is to employ the conditional generative model to provide understanding into learned (SSL) representations. The unconditional model is only developed to compare our generative model and ensure that its quality is reliable for any further down analysis. As such, we propose to learn the representation distribution in a very simple manner via the usual Kernel Density Estimation (KDE). That is, the distribution is modeled as

$$p(\boldsymbol{h}) = \frac{1}{N} \sum_{n=1}^{N} \mathcal{N}(\boldsymbol{h}; f(\boldsymbol{x}_n); I\sigma)$$

with $\sigma$ set to 0.01. By using the above distribution, we are able to sample representations $\boldsymbol{h}$ to then sample images $\boldsymbol{x}$ conditionally to that $\boldsymbol{h}$ using our diffusion model. We provide some samples in Figure 20 to show that even with our very simple conditioning, our method is still able to generate realistic images.

## C    On the closeness of the samples in the representation space

Even if we show that RCDM is able to generate images that seems visually close to the image used for the conditioning, it's still unclear how close those images are in the representation space. We can compute euclidean distances but to know how close the generated samples are to the conditioning, we need to have references that can be used to compare this distance with. As references, we compute the euclidean distance between a conditioning image and random images in the validation set of ImageNet, random images belonging to the same class as the conditioning, the closest images in the training set, the conditioning image on which we applied single data augmentations and the conditoning image on which we applied the data augmentation performed by Swav and Dino (Caron et al., 2020; 2021). The results can be seen in Figure 21 for a RCDM trained with Dino representations and in Figure 22 for a RCDM trained with SimCLR representations. On both Figure, we observe that the generated images with RCDM are closer to the conditioning than the closest neighbors in the entire training set of ImageNet. We also computed the mean and reciprocal mean rank in the main paper (Table 1b) which show that for most SSL models the closest examples in the representation space of the generated images is the image used as conditioning. We also added Figure 17 to show which rank is associated to samples generated by RCDM. For SimCLR, the rank is mostly always 1 whereas we got more diversity for the supervised case. This difficulty of RCDM to generated samples which have their representation that map back to the one used for the conditioning can be explain by the nature of a supervised training. In such scenario, the encoder is trained to map a big set of images (often a specific class) to a specific type of representation whereas SSL models are explicitly train to push each examples farther away from each others. Thus, it seems more likely that a little perturbation on the supervised representation induces a change of nearest neighbor. This hypothesis is supported by Figure 29 which show that small adversarial attack are enough to induces a change of class in the representation which is not the case for SSL encoders.

## D    Analysis of representations learned with Self-Supervised model

Having generated samples that are close in the representation space to a conditioning image can give us an insight on what's hidden in the representations learned with self-supervised models. As demonstrated in the previous section, the samples that are generated with RCDM are really close visually to the image used as conditioning. This give an important proof of how much is kept inside a SSL representation. However, it's also important to consider how much this amount of "hidden" information varied depending on the SSL representation that is used. Therefore, we train several RCDM on SSL representations given by VicReg

(Bardes et al., 2021), Dino (Caron et al., 2021), Barlow Twins (Zbontar et al., 2021) and SimCLR (Chen et al., 2020). In many applications that used self-supervised models, the representation that is used is the one corresponding to the backbone of the ResNet50. Usually, the representation given by the projector of the SSL-model (on which the SSL criterion is applied) is discarded because the results on many downstream tasks like classification is not as good as the backbone. However, since our work is to visualize and better understand the differences between SSL representations, we also trained RCDM on the representation given by the projector of Dino, Barlow Twins and SimCLR. In Figure 24 we condition all the RCDM with the image labelled as conditioning and sample 9 images for each model. If we look at the projector of the SSL models, the generated samples have a higher variance.

To further compare and analyse the different SSL models, we visualize how much SSL representations can be invariant with respect to a transformation that is applied on the conditioning image. In Figure 25, we apply several Data Augmentation: Vertical shift, Zoom out, Zoom In, Grayscale and a Collor Jitter on a given conditioning image. Then we compute the SSL representations of the transformed image with different SSL models and use our corresponding RCDM to see how much the samples have changed with respect to the samples generated on the vanilla conditioning image. We observe that the representation (the 2048 backbone one) of all SSL methods are not invariant to scale and change of colors. Whereas the representation of the projector doesn't seem to take into account any small transformation in the original conditioning outside the scale for Dino. For SimCLR, there is still some information about the background that is kept in the representation however the samples are not as close visually with respect to the 2048 representation. Barlow Twins is interesting because there isn't much differences between the backbone representation (2048) one and the representation of the projector (Size 8192). With the exception that this last representation seems to be more invariant to color shift than the backbone one.

We also perform an experiment in Figure 28 using OOD images to ensure that the conclusions drawn with our methods about SSL representation are not specific to ImageNet.

### D.1 Visualization of adversrial examples

We use RCDM to visualize adversarial examples for different models. For each model, we trained a linear classifier on top of their representations to predict class labels for the ImageNet dataset. Then, we use FGSM attacks over the trained model using a NLL loss to generate adversarial examples. In Figure 29 we show the adversarial examples that are created for each model, the samples generated by RCDM with respect to the representation of the adversarial perturbed example and the class label predicted by the linear classifier over the adversarial examples. The supervised model is very sensitive to the attack whereas SSL models seems more robust.

### D.2 Manipulation of SSL representations

It is also possible to manipulate SSL representations to generate new images. We try to apply addition and subtractions over SSL representations (similarly to what has been done in NLP). From two different images, we compute the difference between the two corresponding representations and add the difference vector to a third image. Figure 32 shows that it is possible to apply such transformations meaningfully in the SSL space. We also used another setup where we choose specific dimensions in the representation based on how many times these dimensions are non zero in the representation space of a set of neighbors. Then we set this dimension to zero which surprisingly induces the removing of the background in the generated images. We also replace them by the same corresponding dimension of another images which induces a change of background toward the one of the new image. Results are shown in Figure 30.

### D.3 Experiments with vision transformers

All the experiments in this paper were conducted with Resnet50 since most of the SSL baselines are available with this model. However, RCDM can work with any type of architecture, including vision transforms. In Figure 33, we show RCDM samples using representations of Dino trained with a VIT-B 16 (Kolesnikov et al., 2021).

### D.4 Why is my model over-fitting on the training set ?

By enabling the visualization of what is learned in a representation, RCDM can help researchers to get a better understanding of the failures modes of their models. In one of our experiments, we trained a SSL models with VicReg by using only cropping as data augmentation (thus discarding the traditional colorjiterring/grayscaling and other transforms that change the colors). Training a linear probe on such network resulted in a training accuracy of 95% on the training set while the validation accuracy was only about 20%. To better understand how the model was able to overfit on the training set, we trained RCDM on the representations of this model. The samples obtained are shown in Figure 34. This experiment validate the hypothesis that removing color related augmentations during the training of SSL models leads to learn representations that are only colors and textures based.

### D.5 Visualizing how representations are changing during training

Another way one can use RCDM, is to consider how representations are changing during training. In this experiment, we trained 3 RCDM models on the representation given by SSL models (VicReg) trained after 1 epoch, 5 epochs and 50 epochs. In this experiment, we want to visualize what is changing in the representation during training. The hypothesis was that at the beginning of the training, the network is learning some easy feature, like some color information, and later in the training more complex features, probably containing more shape based information. In Figure 35, we observe that after 1 epoch of SSL training, the information retain in the representation is mostly color/texture based while after only 5 epochs, we can see that the shape are better defined.

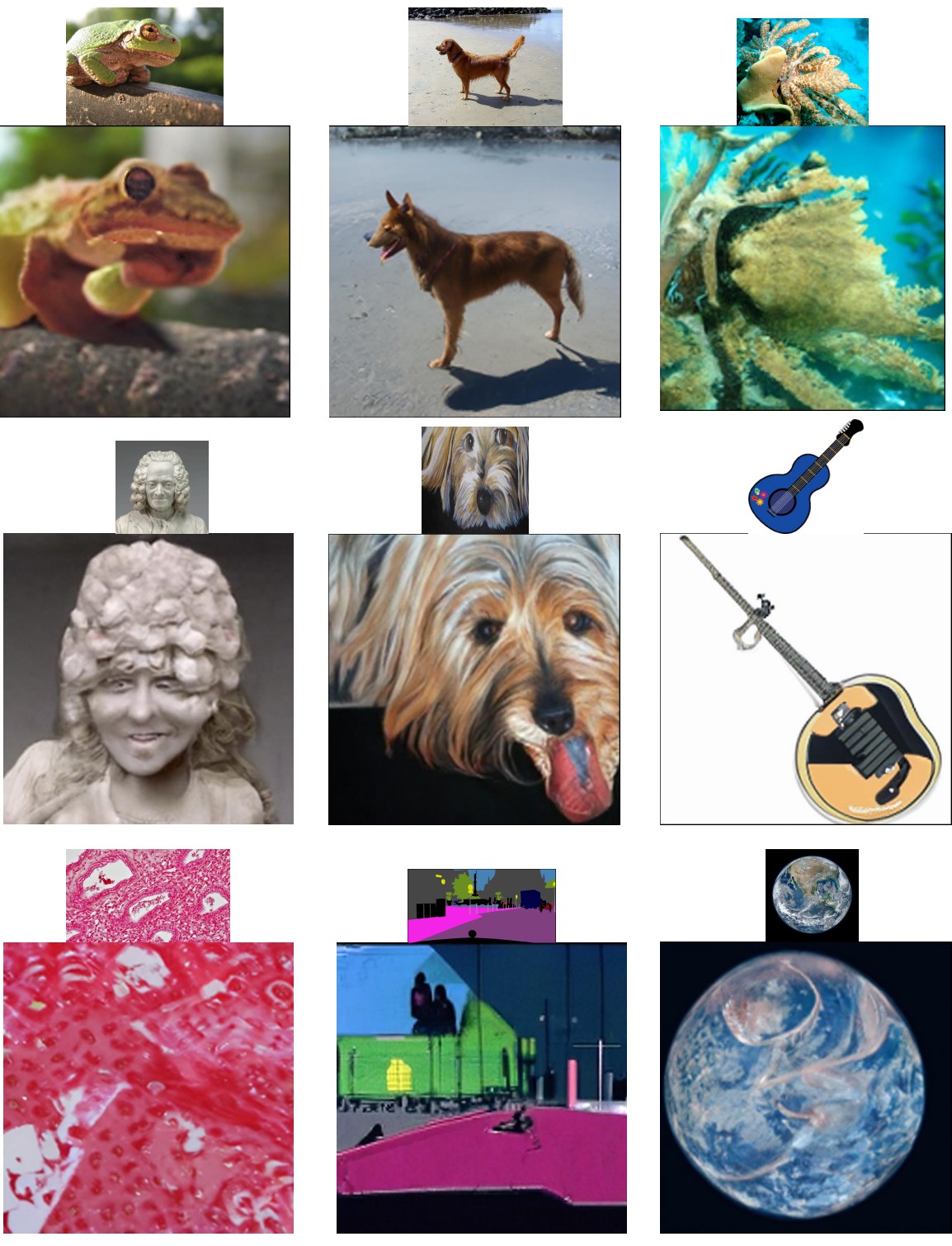

Figure 13: High resolution samples from our conditional diffusion generative model using the super resolution model of Dhariwal & Nichol (2021). We use the small images on the top of each bigger image as conditioning (source NASA for the earth picture) for a diffusion model trained with Dino representation on 128x128 images. Then, we feed the samples generated to the super resolution model of Dhariwal & Nichol (2021) which produces images of size 512x512. Since the super resolution model is conditional, we sample a random label. We note that the high resolution samples are still very close to the conditioning.

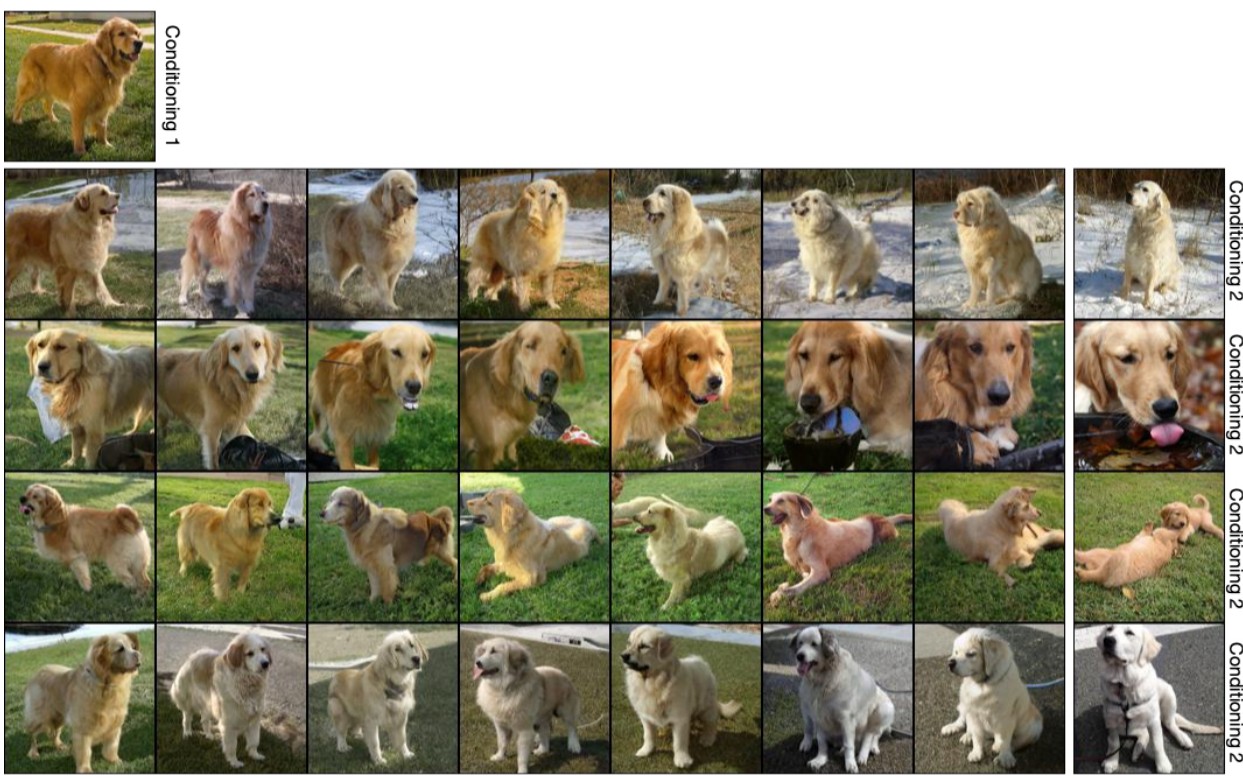

(a) Linear interpolation between the image of the golden retriever in conditioning 1 with various other images belonging to the same class as conditioning 2.

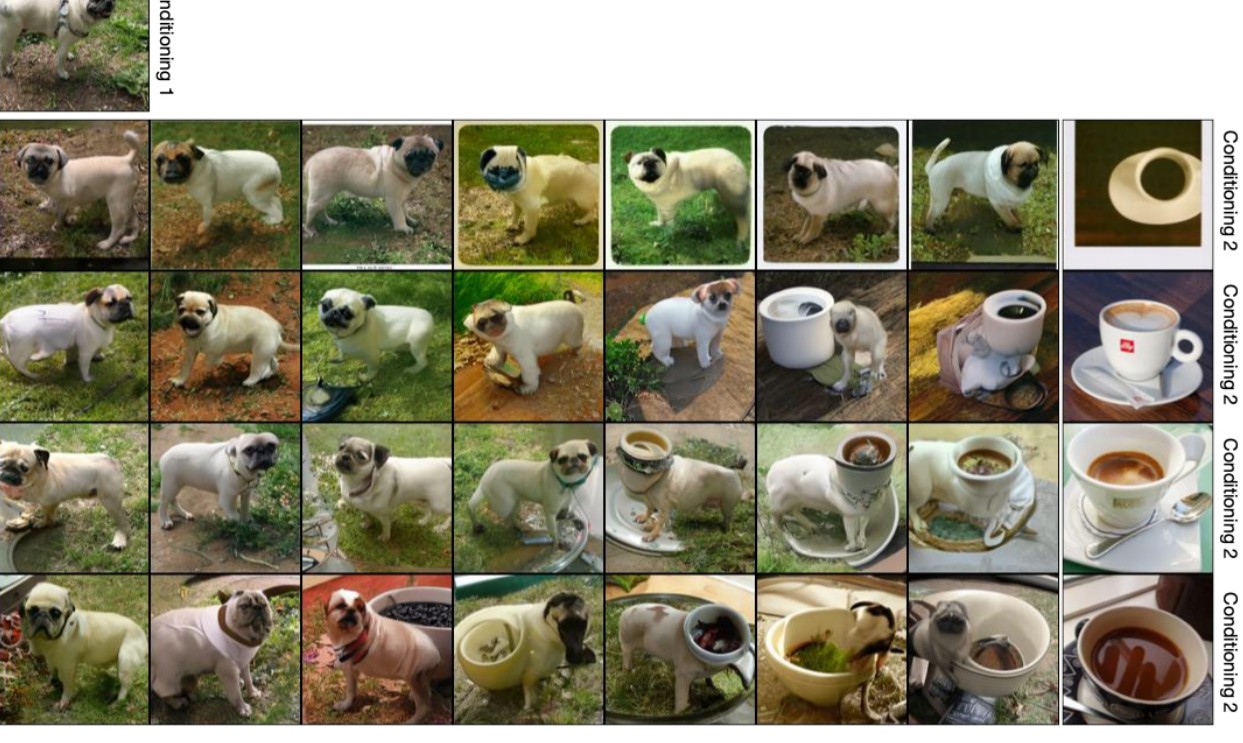

(b) Linear interpolation between the image of the pug in conditioning 1 with various other images belonging to the espresso class as conditioning 2.

Figure 14: Each vectors that result from the linear interpolation is feed to a RCDM trained with Barlow Twins representation.

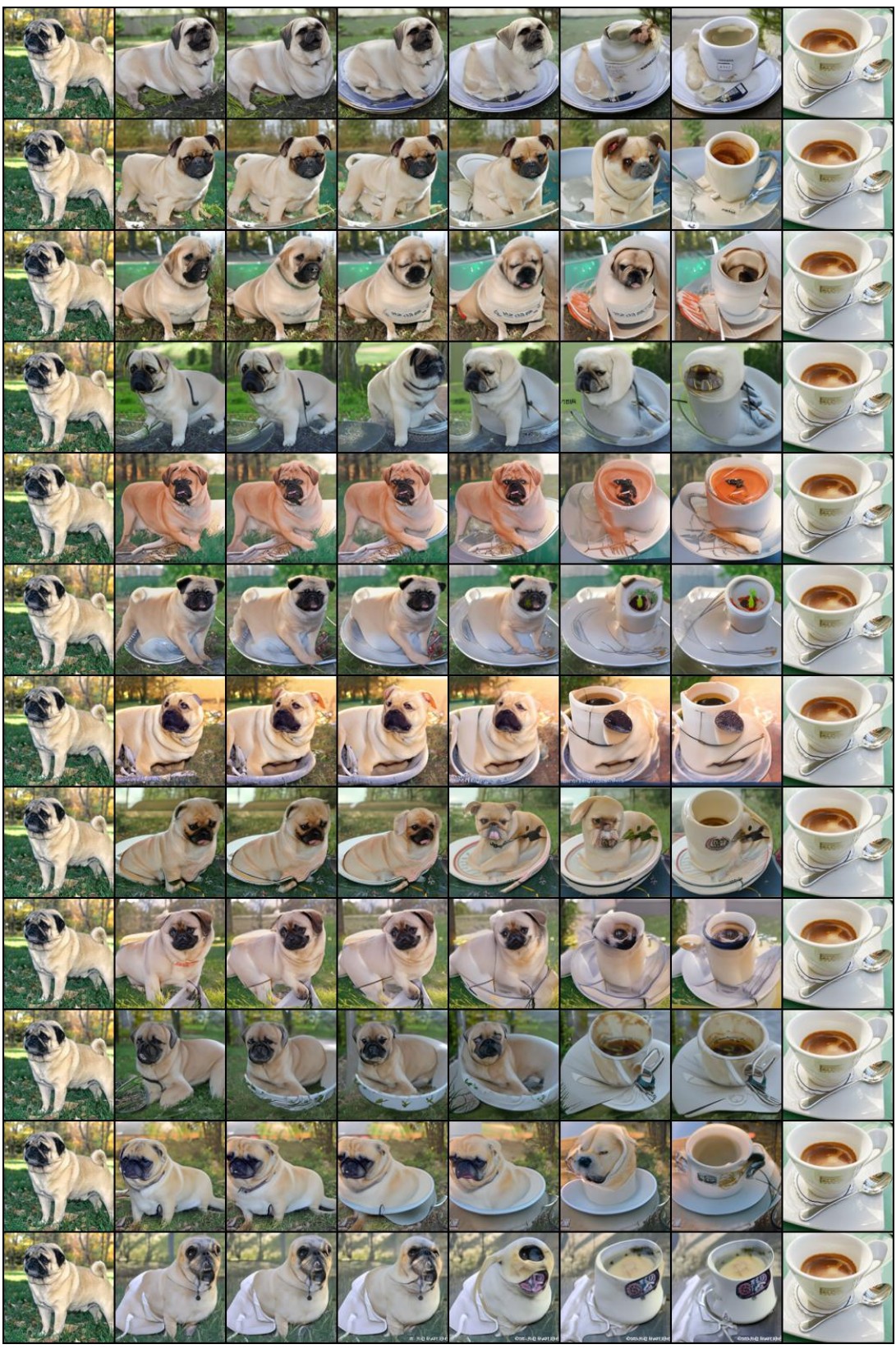

Figure 15: Diversity of the samples generated by RCDM on interpolated representations. Each row corresponds to different random noise for the same conditioning. On the first and last column are the real images used for the interpolation. All of the images in-between those rows are samples from RCDM.

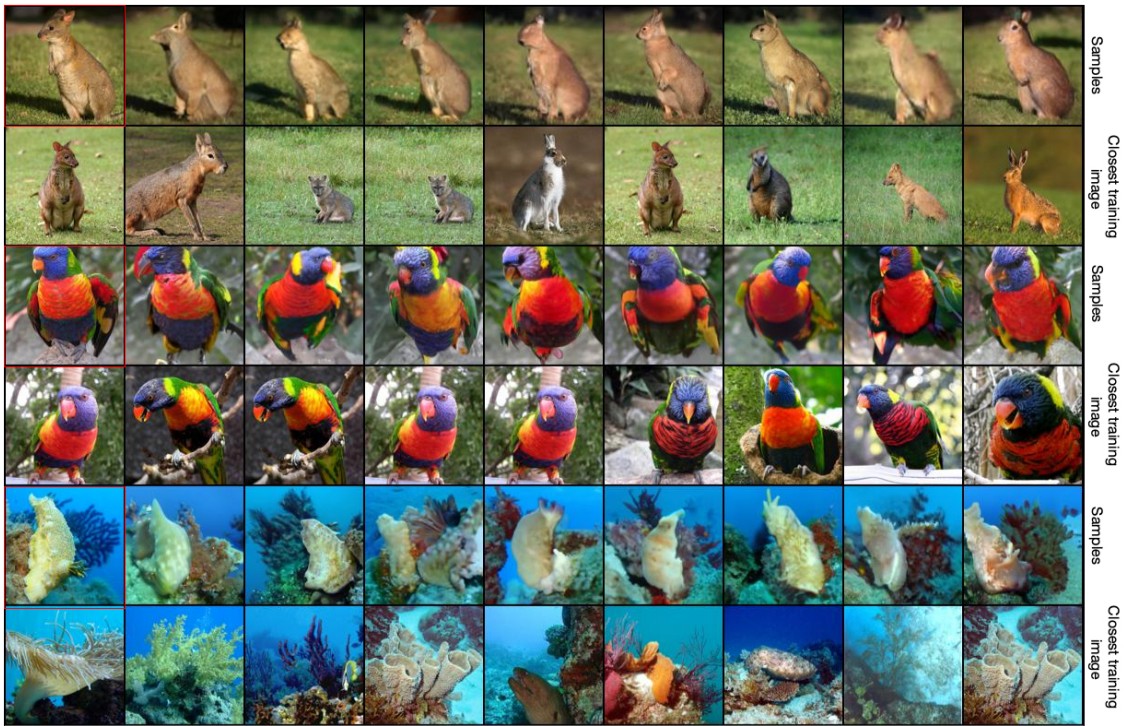

(a) Closest real images (ImageNet training set) from images sampled with RCDM trained on Dino (backbone) representation (2048).

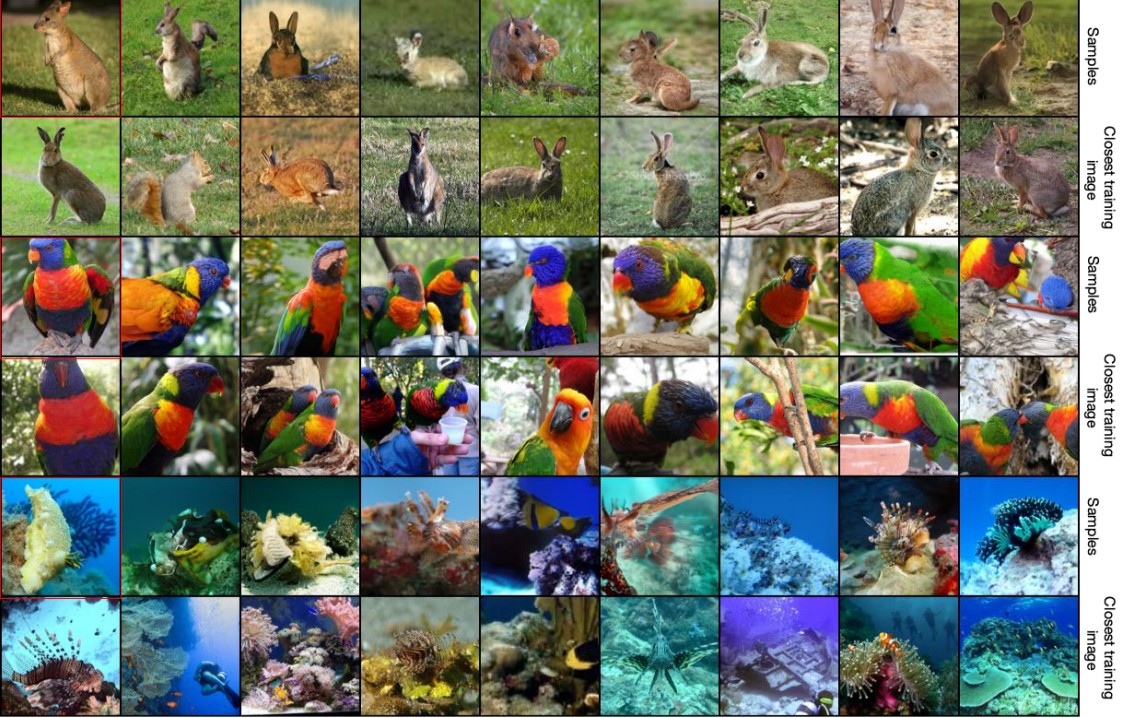

(b) Closest real images (ImageNet training set) from images sampled with RCDM trained on Dino projector representation (256).

Figure 16: We find the nearest neighbors in the representation space of samples generated by RCDM. The images in the red squared are the ones used for conditioning.

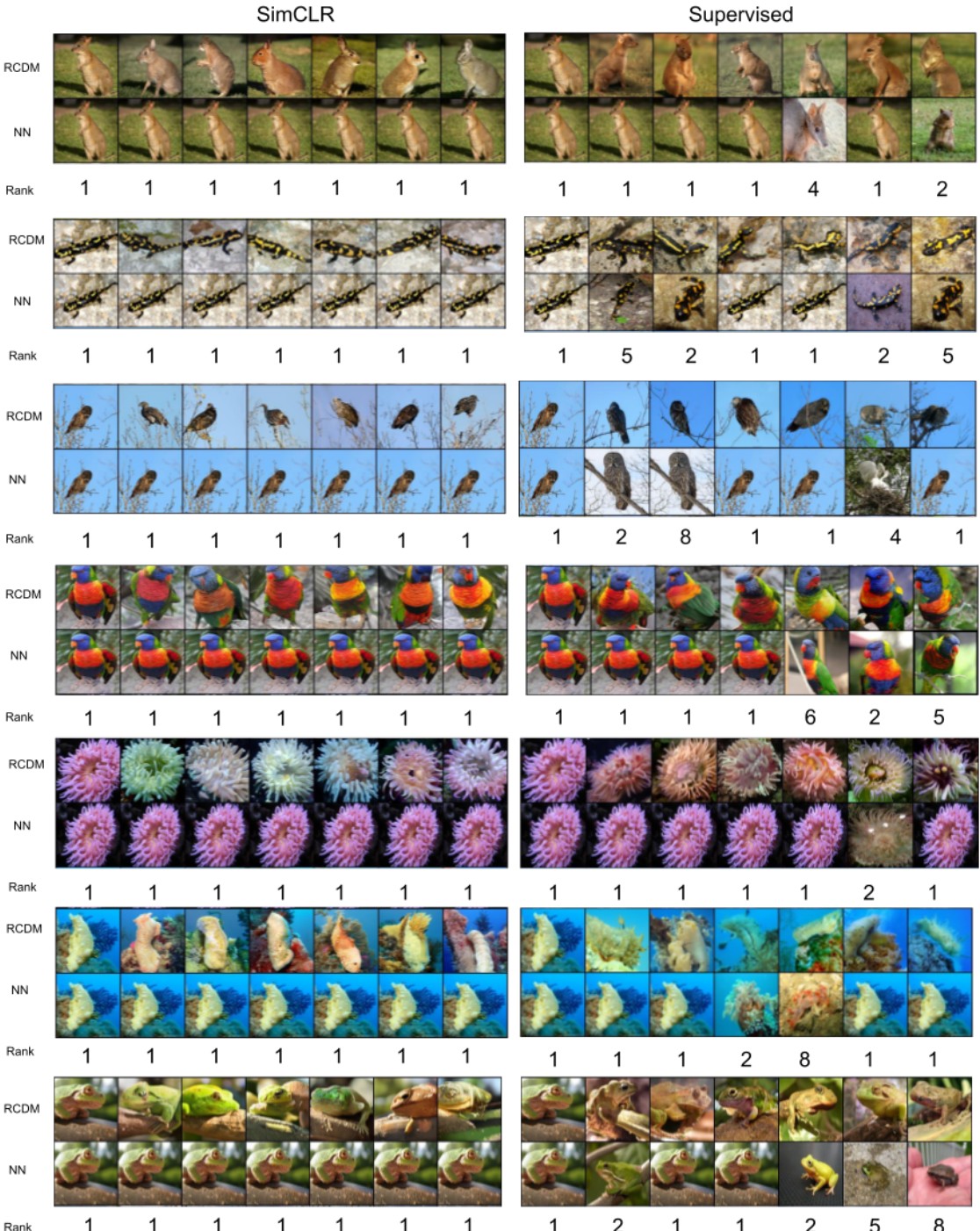

Figure 17: After generating samples with respect to a specific conditioning, we compute back the representation of the generated samples and find which are the closest neighbors in the validation set. Then, we compute the rank of the original image that was used as conditioning within the set of neighbors. When the rank is one, it implies that the nearest neighbors of the generated samples is the conditioning itself, meaning that the generated samples have their representation that is very close in the representation space to the one used as conditioning. We can see that for SimCLR, the generated samples are much closer in the representation space to their conditioning than the supervised representation. This is easily explain by the fact that supervised model learn to map images from a same class toward a similar representation whereas SSL models try to push further away different examples.

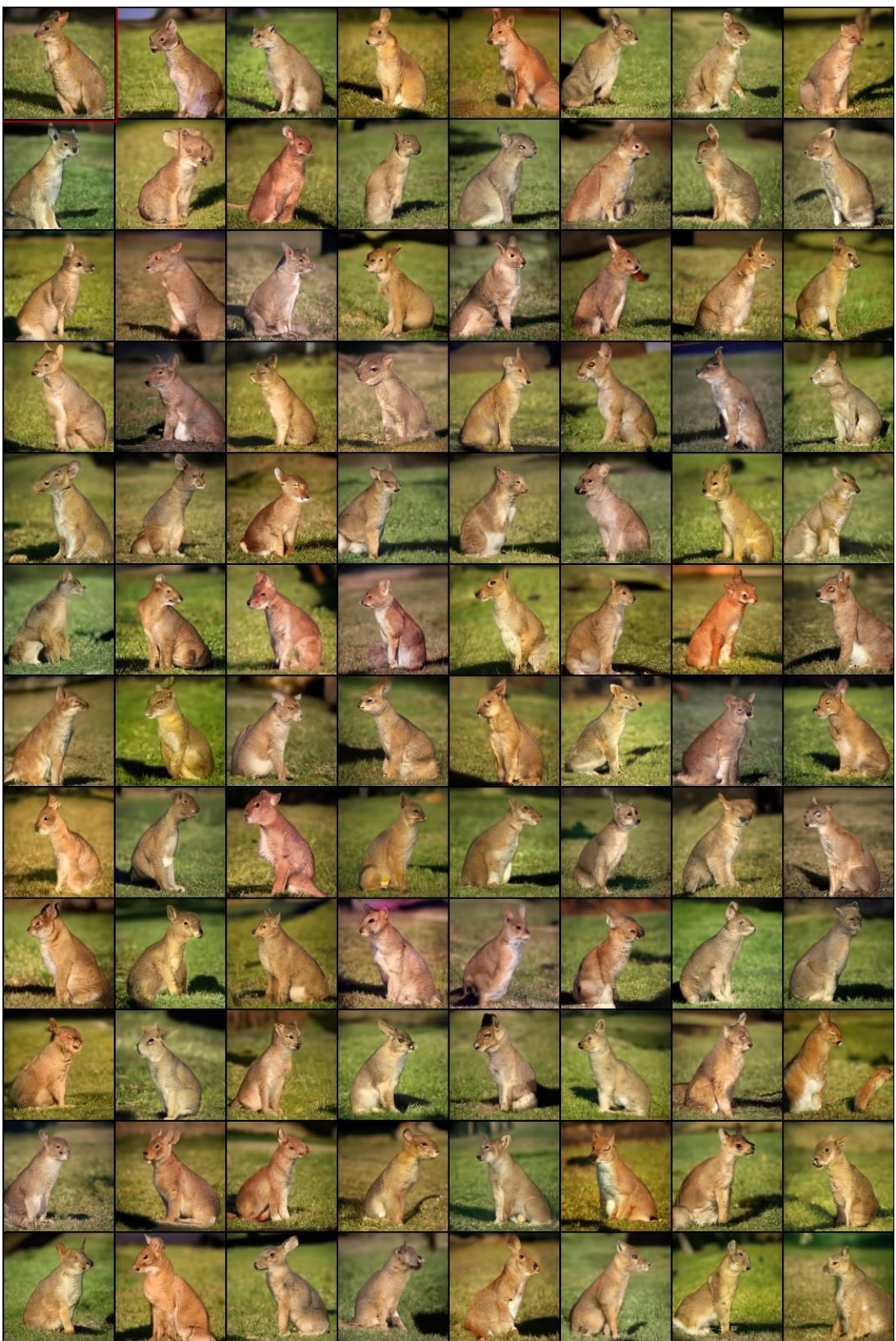

Figure 18: Visual analysis of the variance of the generated samples for a specific image when using the trunk/backbone of a Barlow Twins encoder. The first image (in red) in the one used as conditioning.

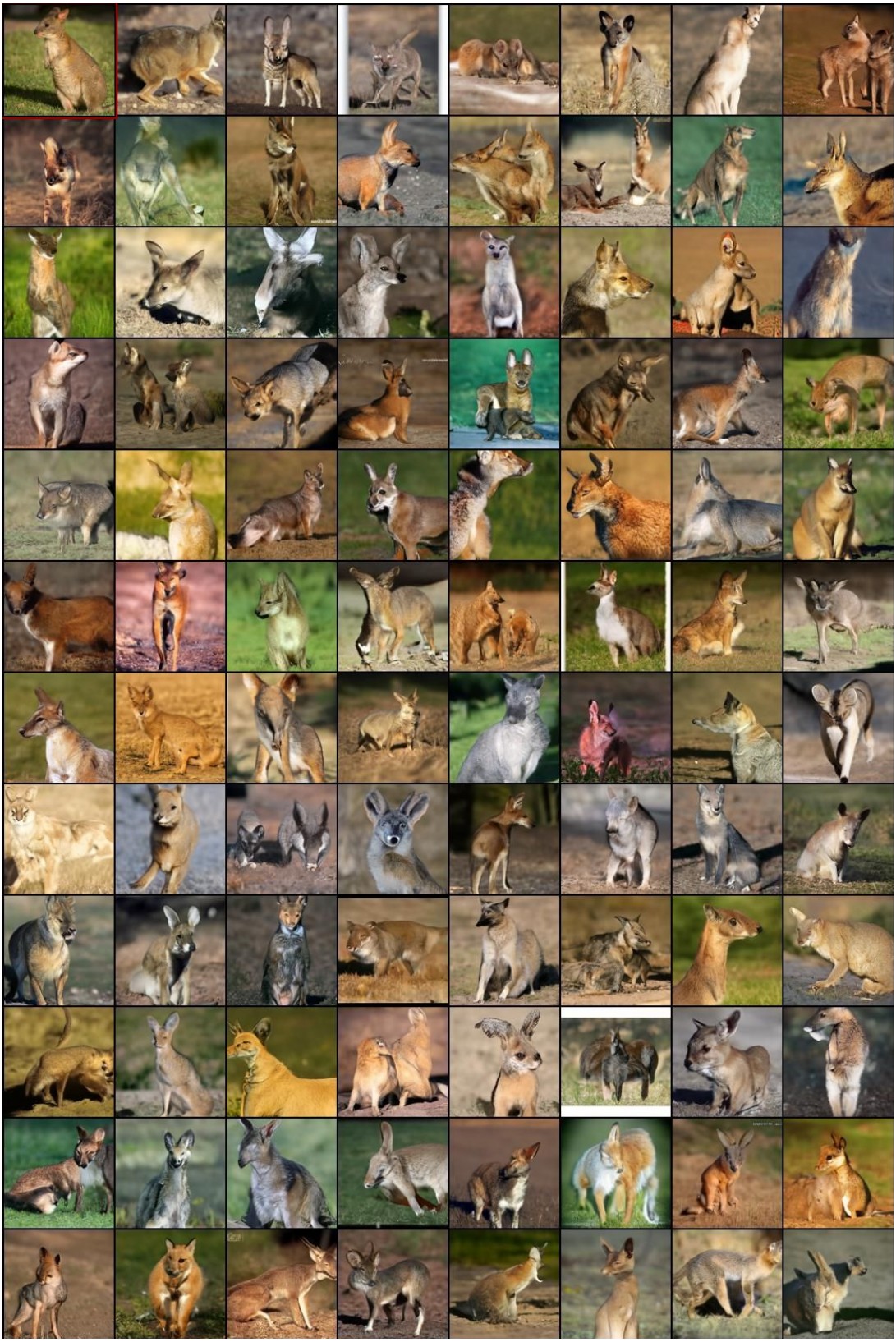

Figure 19: Visual analysis of the variance of the generated samples for a specific image when using the projector/head of a Barlow Twins encoder. The first image (in red) in the one used as conditioning.

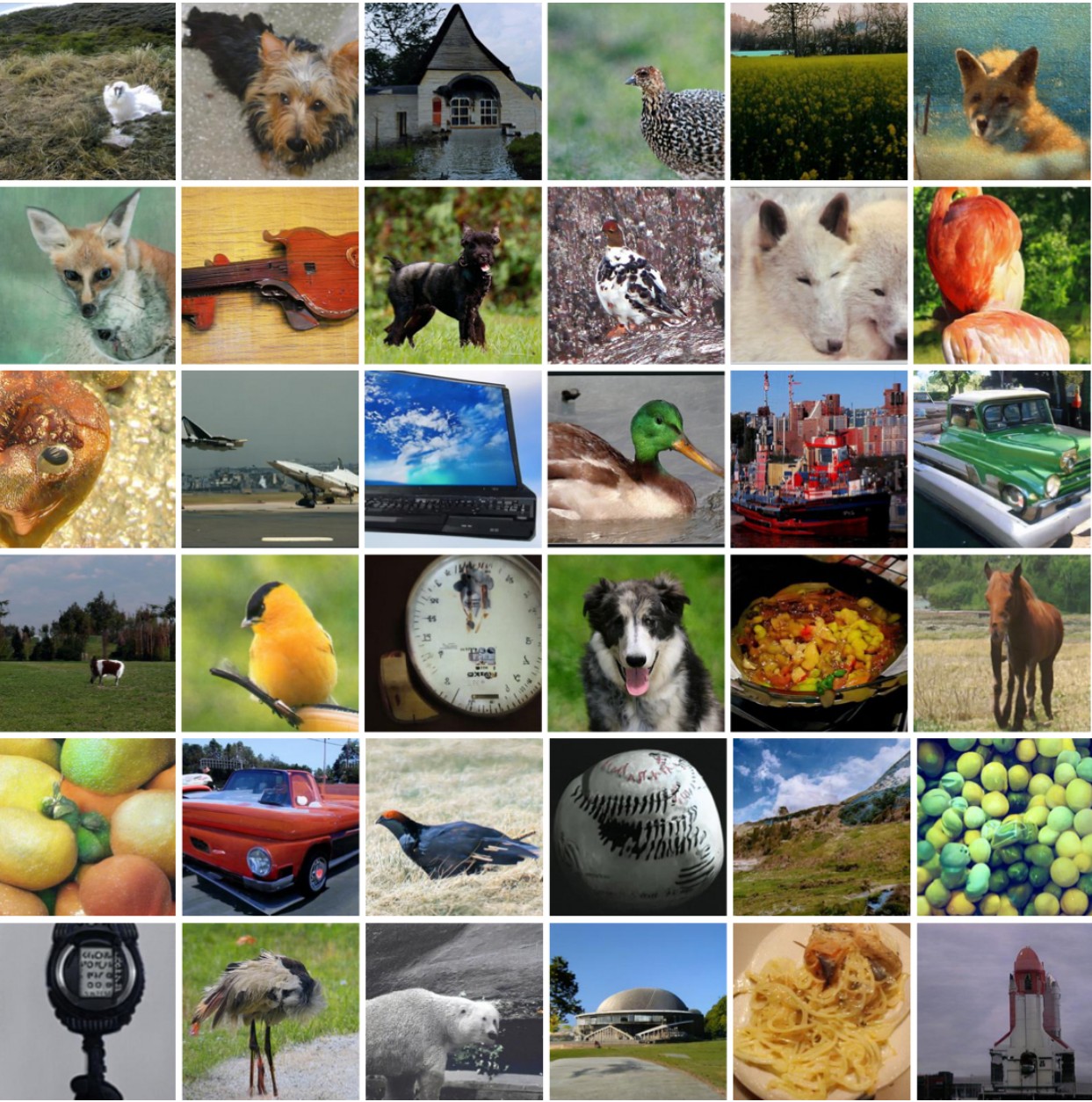

Figure 20: Unconditional generation following the protocol of section B. Our simple generative model of representations consists in applying a small Gaussian noise over representation computed from random training images of ImageNet. We use these noisy vector as conditioning for our 256x256 RCDM trained with Dino representations. We note that the generated images looks realistic despite some generative artefact like a two-headed dog and an elephant-horse.

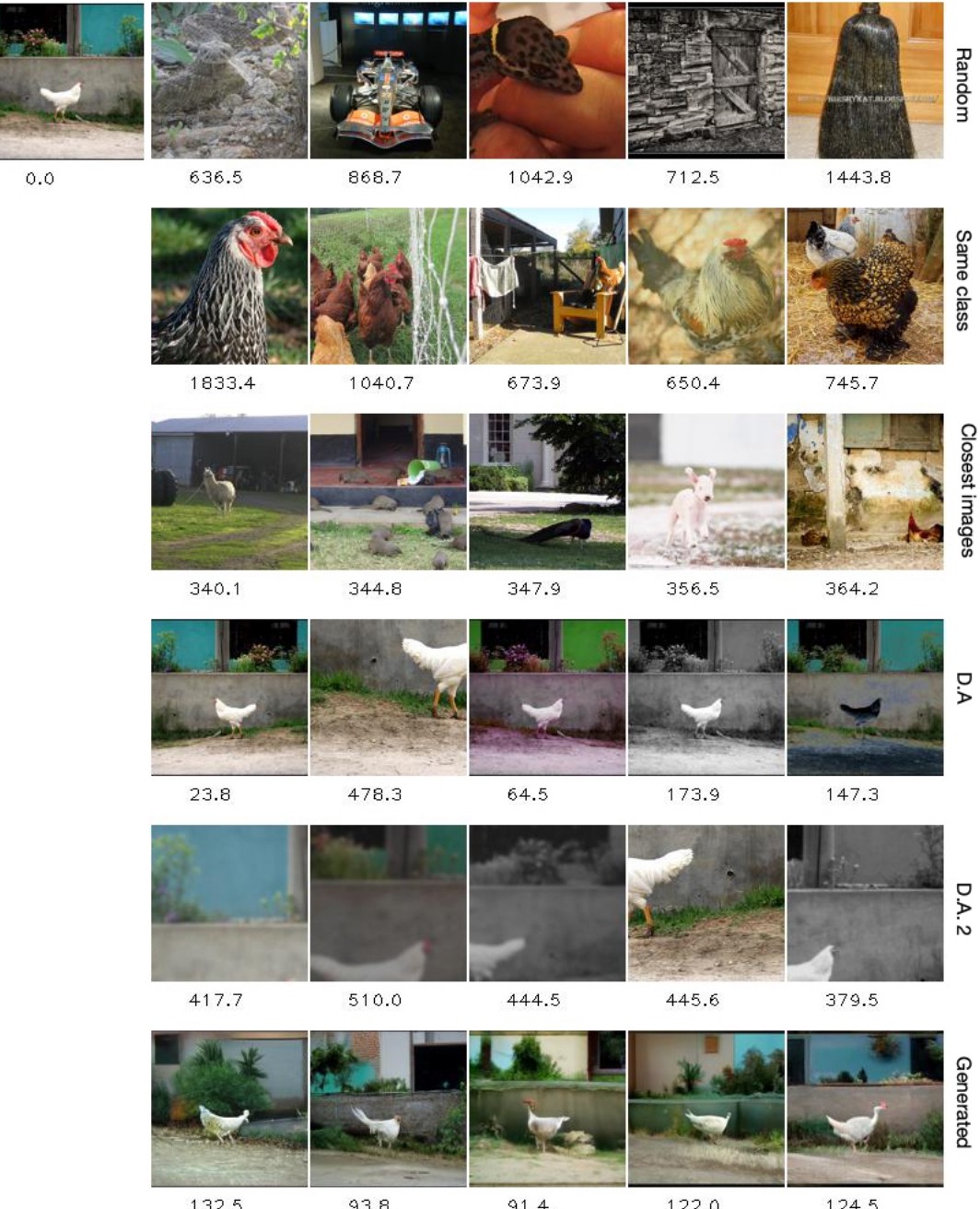

Figure 21: **Squared Euclidean distances in the Dino representation space.** We show the squared euclidean distance between the conditioning image on the leftmost column on first row and different images to get an insight about how close the samples generated by the diffusion model stay close to the representation used as conditioning. The distances with the conditioning is printed below each images. On the first row, we show random images from the ImageNet validation data. On the second row, we take random validation examples belonging to the same class as the conditioning. On the third row, we find the closest training neighbors of the conditioning in the representation space. On forth row, D.A. means Data Augmentation which consist in horizontal flip, CenterCrop, ColorJitter, GrayScale and solarization. On fifth row (D.A. 2), we use the random data Augmentation used in the paper of (Caron et al., 2020; 2021). On the last row, we show the generated samples from our conditional diffusion model that use **Dino representation**. The samples produces by our model are much closer to the conditioning than other images.

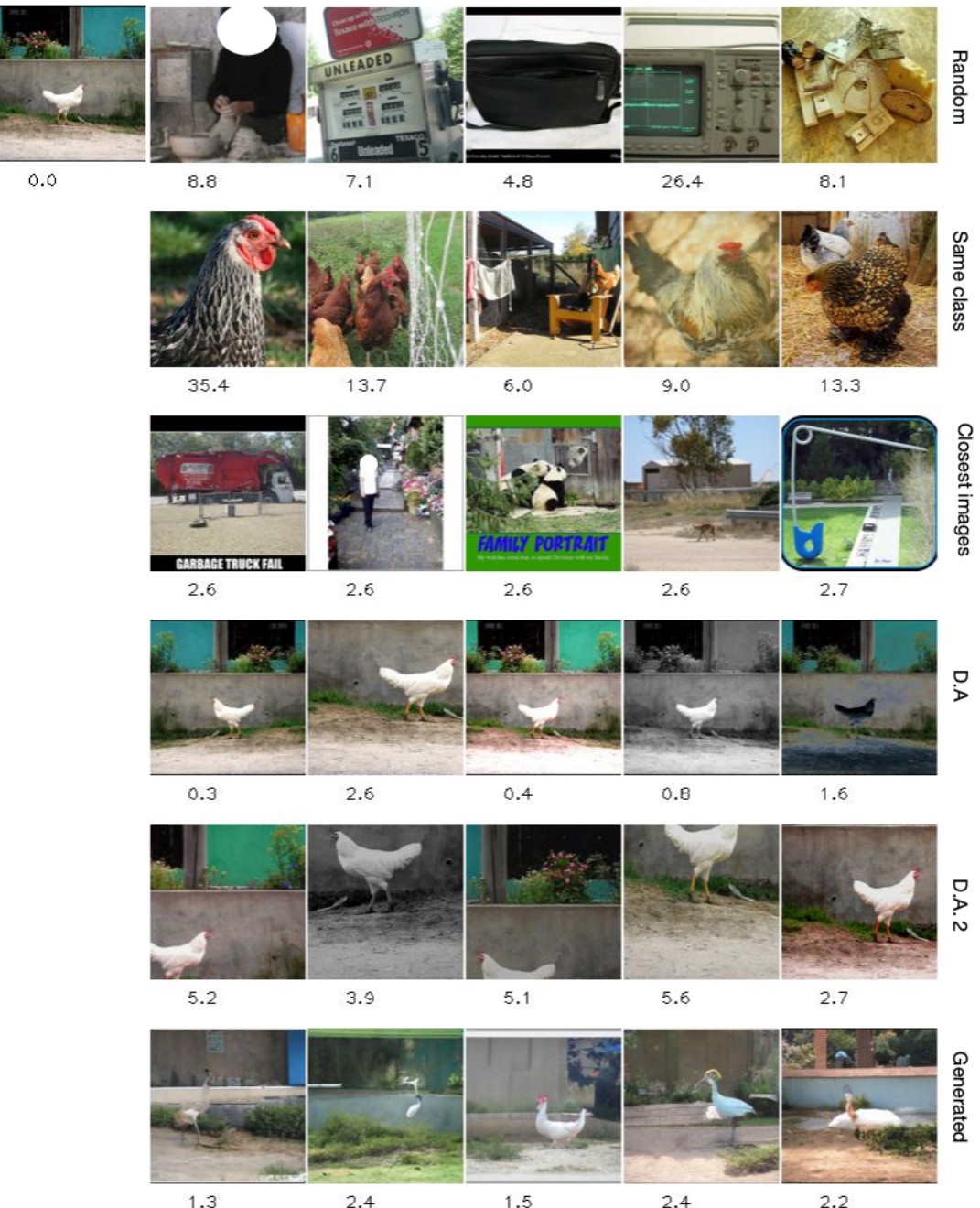

Figure 22: **Squared Euclidean distances in the SimCLR projector head representation space.** We show the squared euclidean distance between the conditioning image on the leftmost column on first row and different images to get an insight about how close the samples generated by the diffusion model stay close to the representation used as conditioning. The distances with the conditioning is printed below each images. On the first row, we show random images from the ImageNet validation data. On the second row, we take random validation example belonging to the same class as the conditioning. On third row, we find the closest training neigbords of the conditioning in the representation space. On forth row, D.A. means Data Augmentation which consist in horizontal flip, CenterCrop, ColorJitter, GrayScale and solarization. On fifth row (D.A. 2), we use the random data Augmentation used in the paper of (Caron et al., 2020; 2021). On the last row, we show the generated samples from our conditional diffusion model that use **SimCLR projector head representation**. The samples produces by our model are much closer to the conditioning than other images.

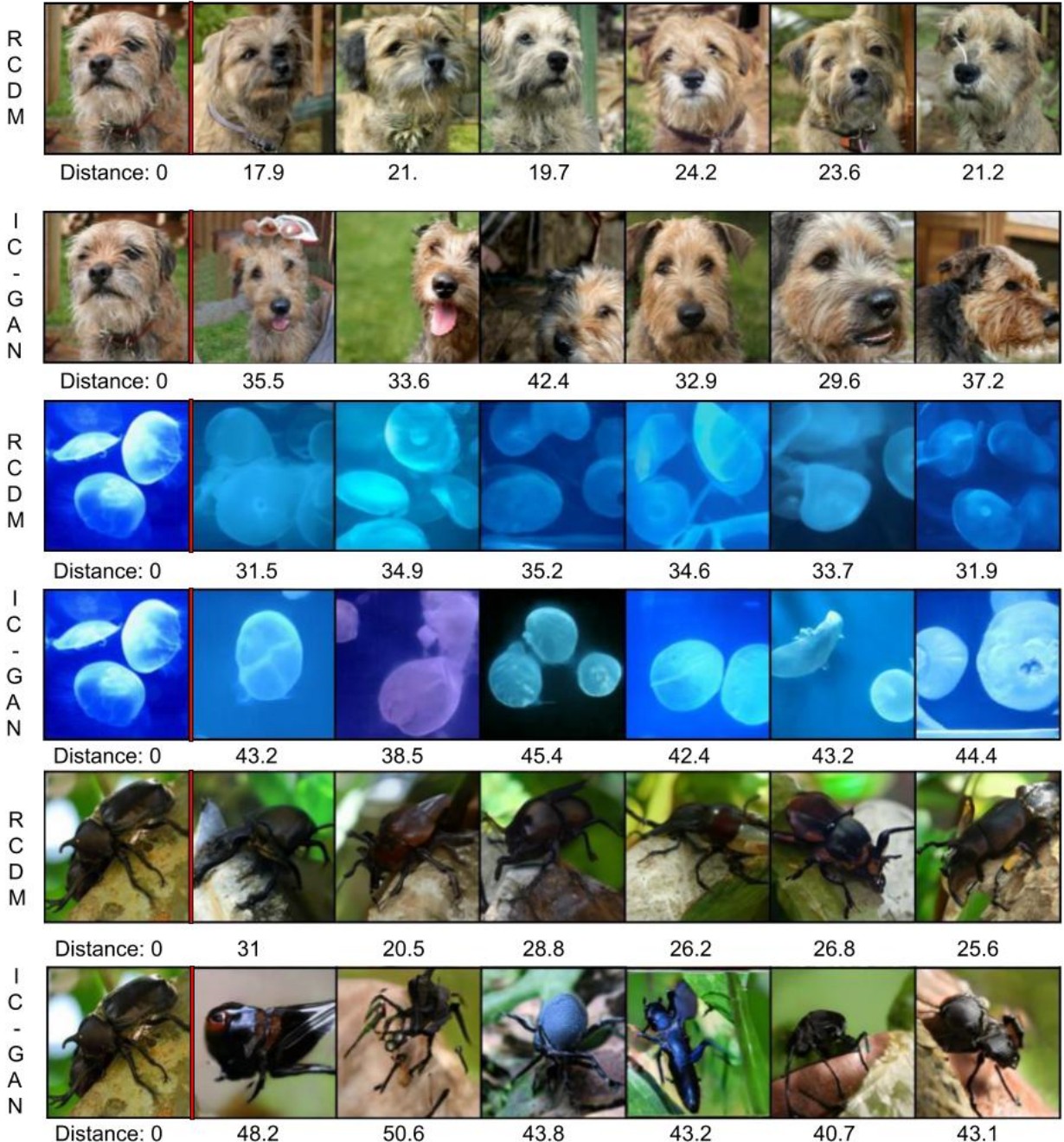

Figure 23: Comparison of the euclidean distance between IC-GAN and RCDM. We use the same self-supervised representation as conditioning (Swav encoder) for RCDM and IC-GAN. We compute the euclidean distance between the representation of the generated images versus the representation used as conditioning. We observe that samples of RCDM are much closer in the representation space (and also visually) to the conditioning. Samples of IC-GAN show a higher variability, thus farther away in the representation space.

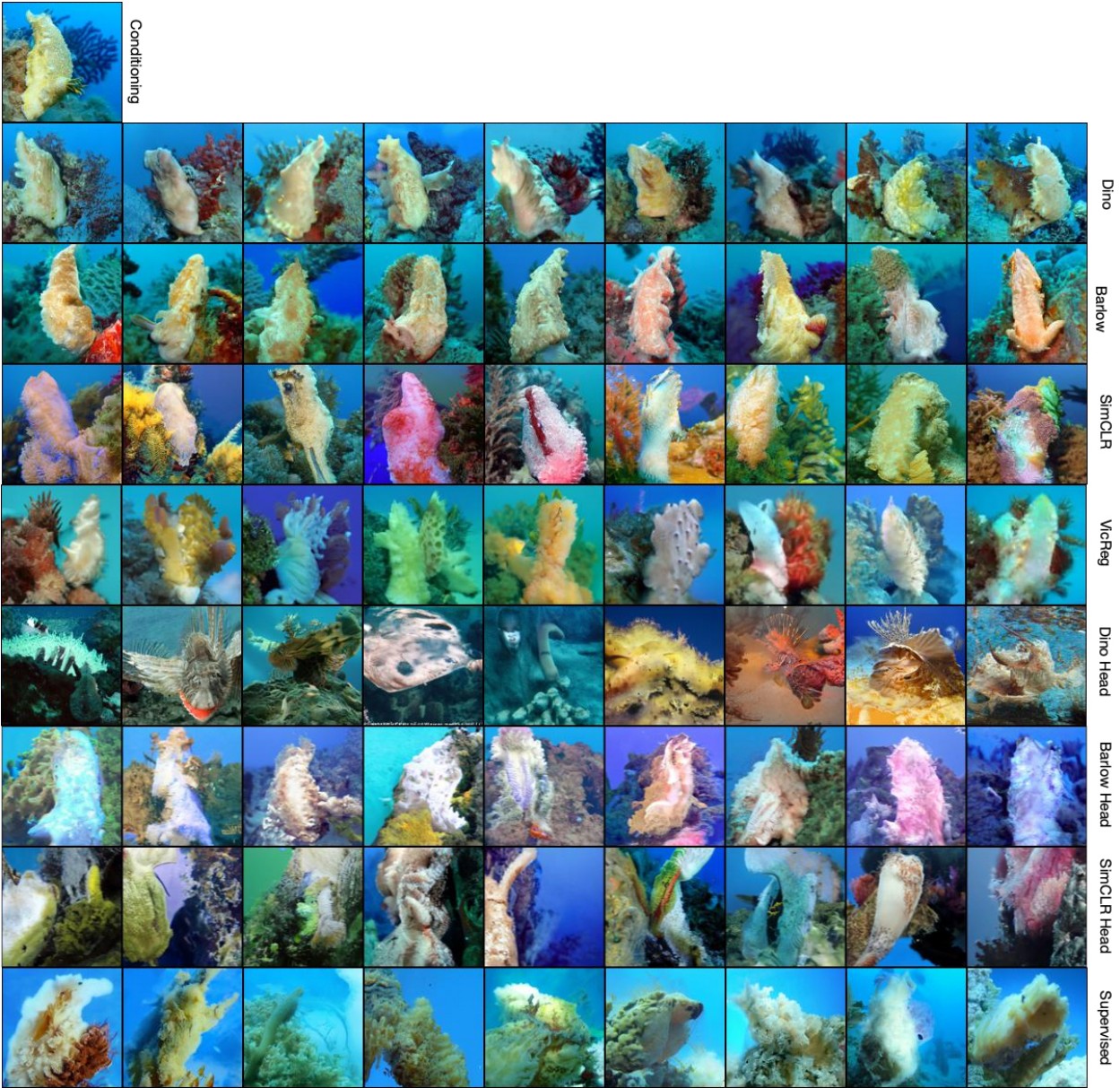

Figure 24: Generated samples from RCDM trained with representation from various self-supervised models. We generate 9 samples for each model with different random seeds. We observe that the representation given by dino isn't very invariant while the one given by SimCLR or VicReg show much better invariance. We also show the samples of RCDM trained on the representation given by the projector (The embedding on which is usually applied the SSL criterion). There is a much higher variability in the generated samples. Maybe too much to be used for a classification task since we can observe class crossing.

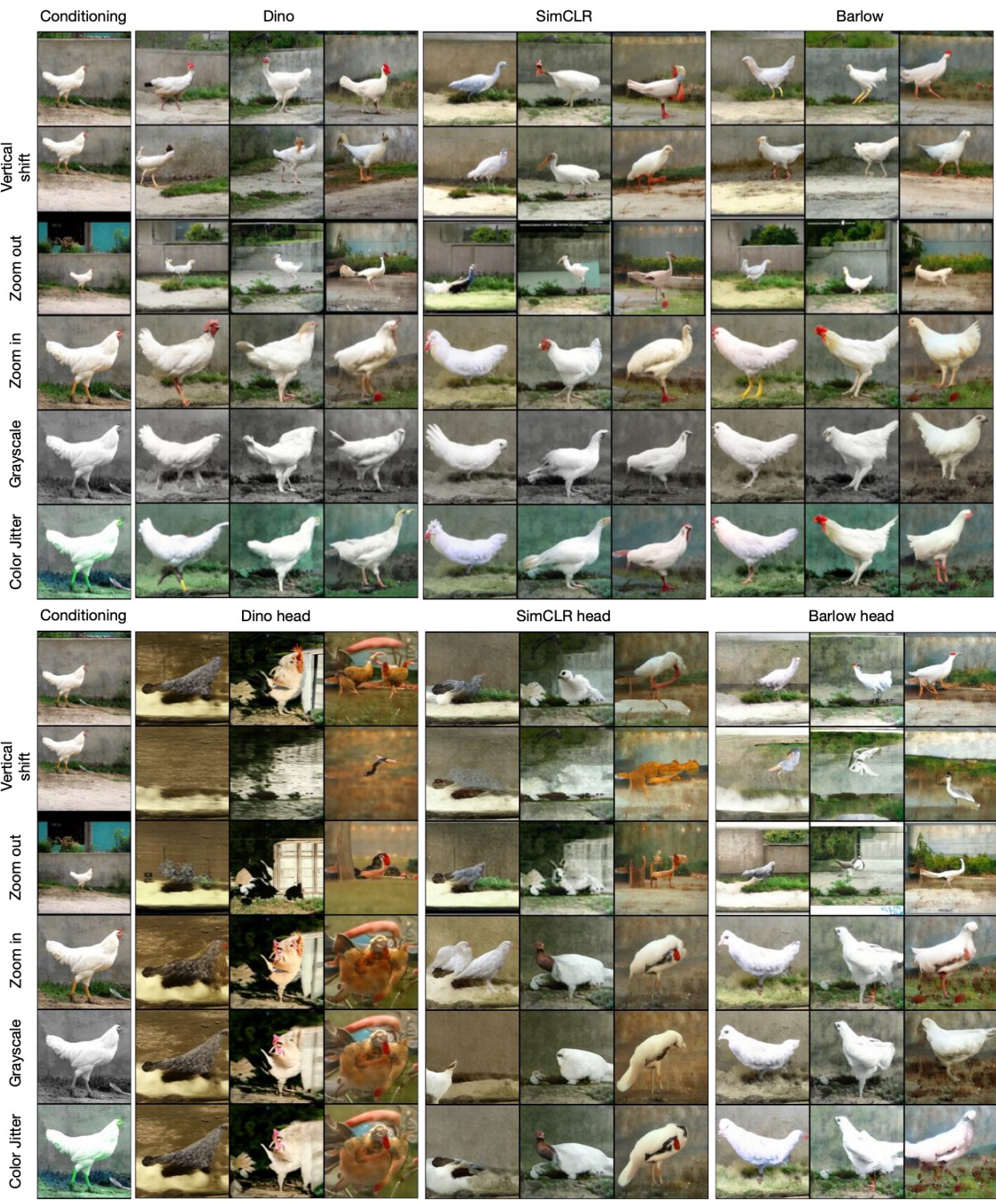

Figure 25: We compare how much the samples generated by RCDM change depending on different transformations of a given image and the model and layer used to produces the representation. Top half uses 2048 representation. Bottom half uses the lower dimensional projector head embedding. We observe that using the projector head representation leads to a much larger variance in the generated samples whereas using the traditional backbone (2048) representation leads to samples that are very close to the original image. We also observe that the projector representation seems to encode object scale, but contrary to the 2048 representation, it seems to have gotten rid of grayscale-status and background color information.

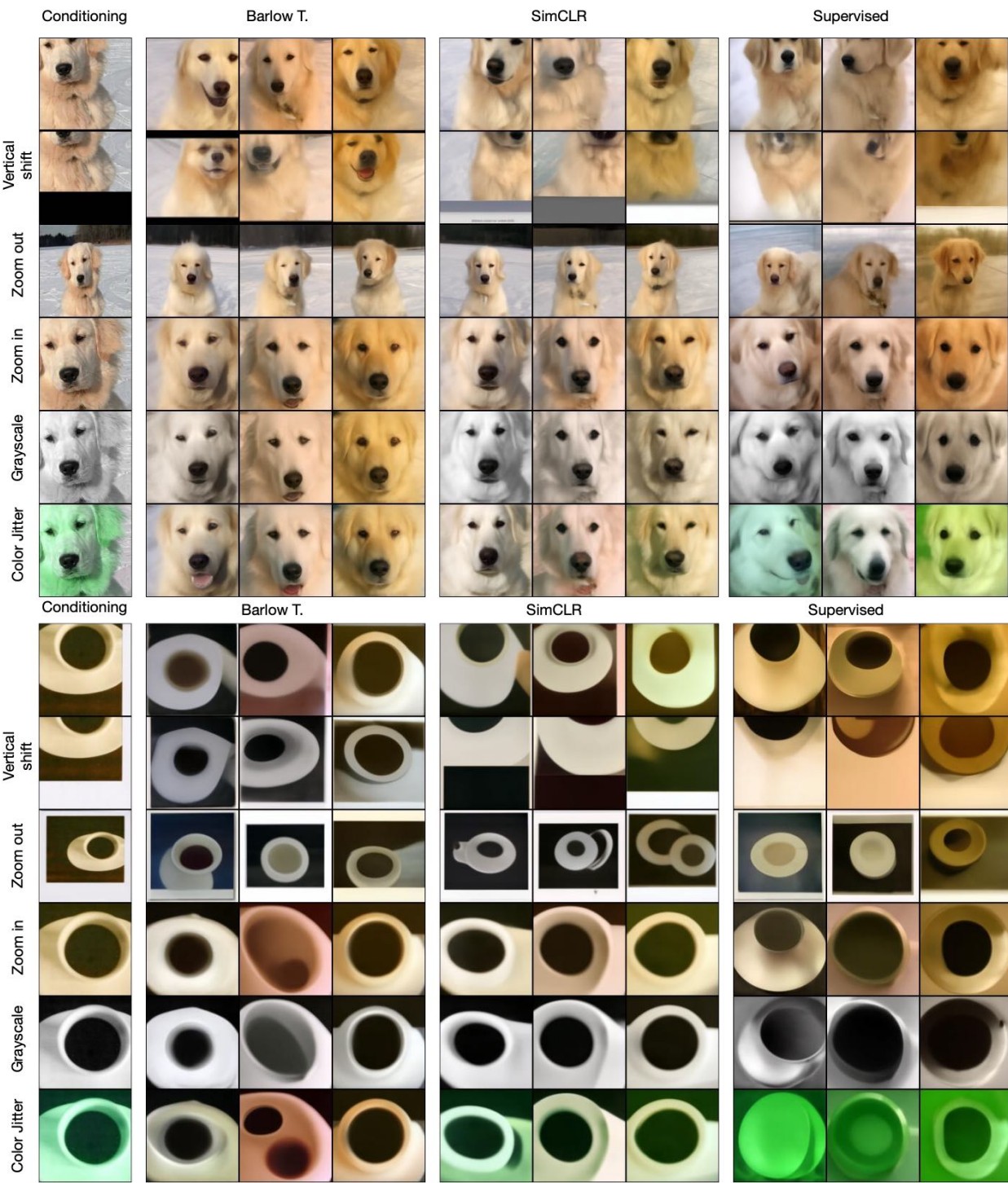

Figure 26: Same setup as Figure 25 except with other images as conditioning.

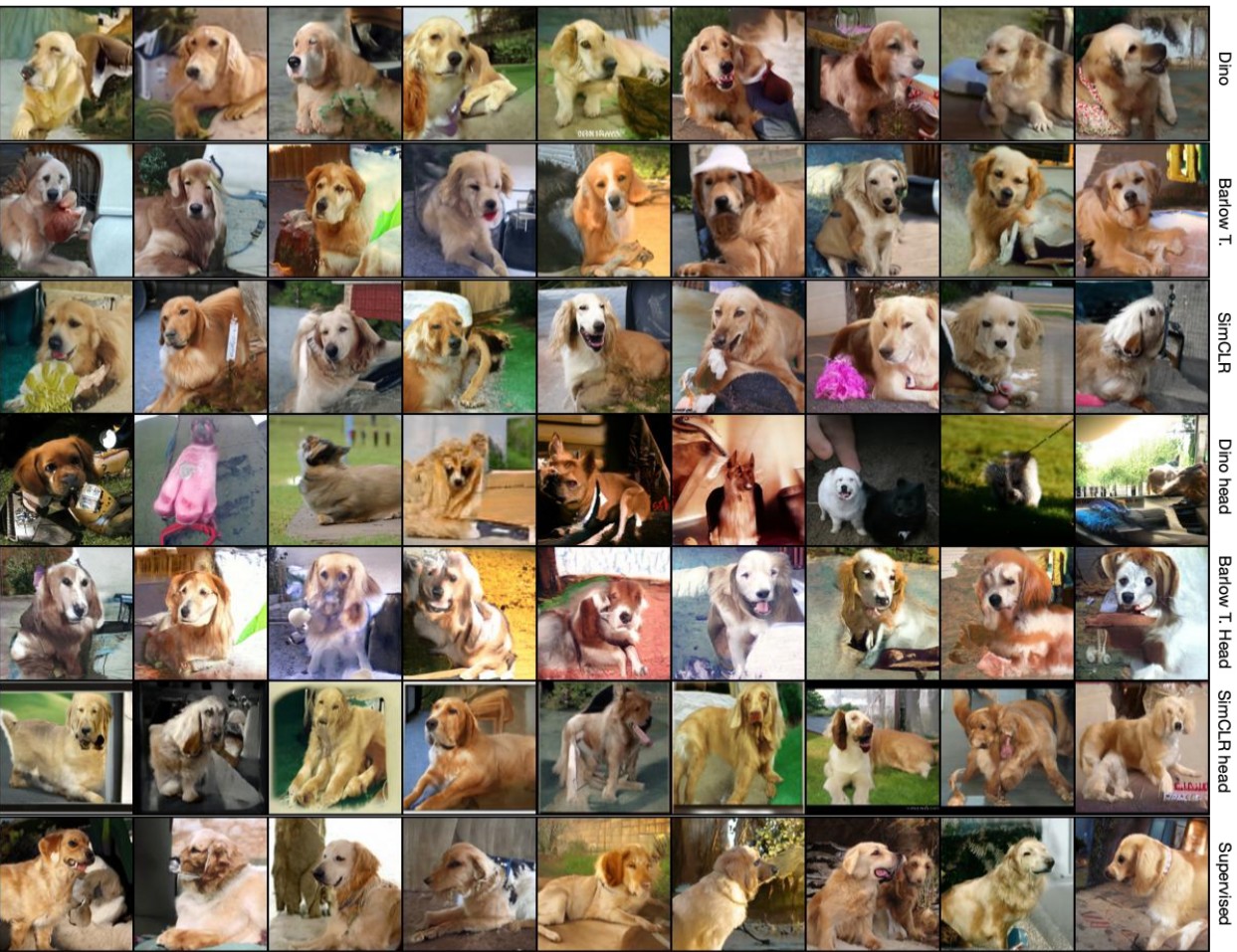

Figure 27: Generated samples from RCDM using the mean representation for a specific class (golden retriever) in ImageNet for various SSL models.

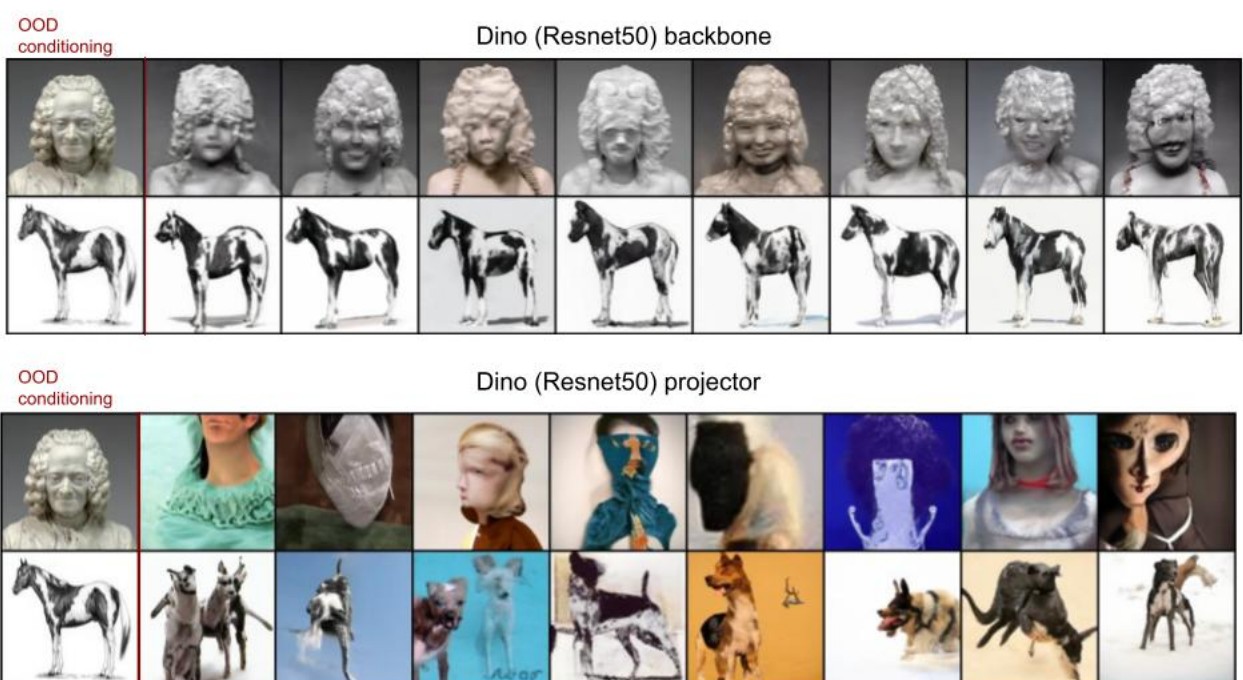

Figure 28: We compare the visualization obtained with representations from Dino (resnet50) at the backbone level and also at the projector level on OOD images to ensure that conclusions drawn with our model about SSL representations are not specific to ImageNet. We confirm that we observe the sames phenomenons in an OOD settings as the ones we could get on an In-Distribution scenario.

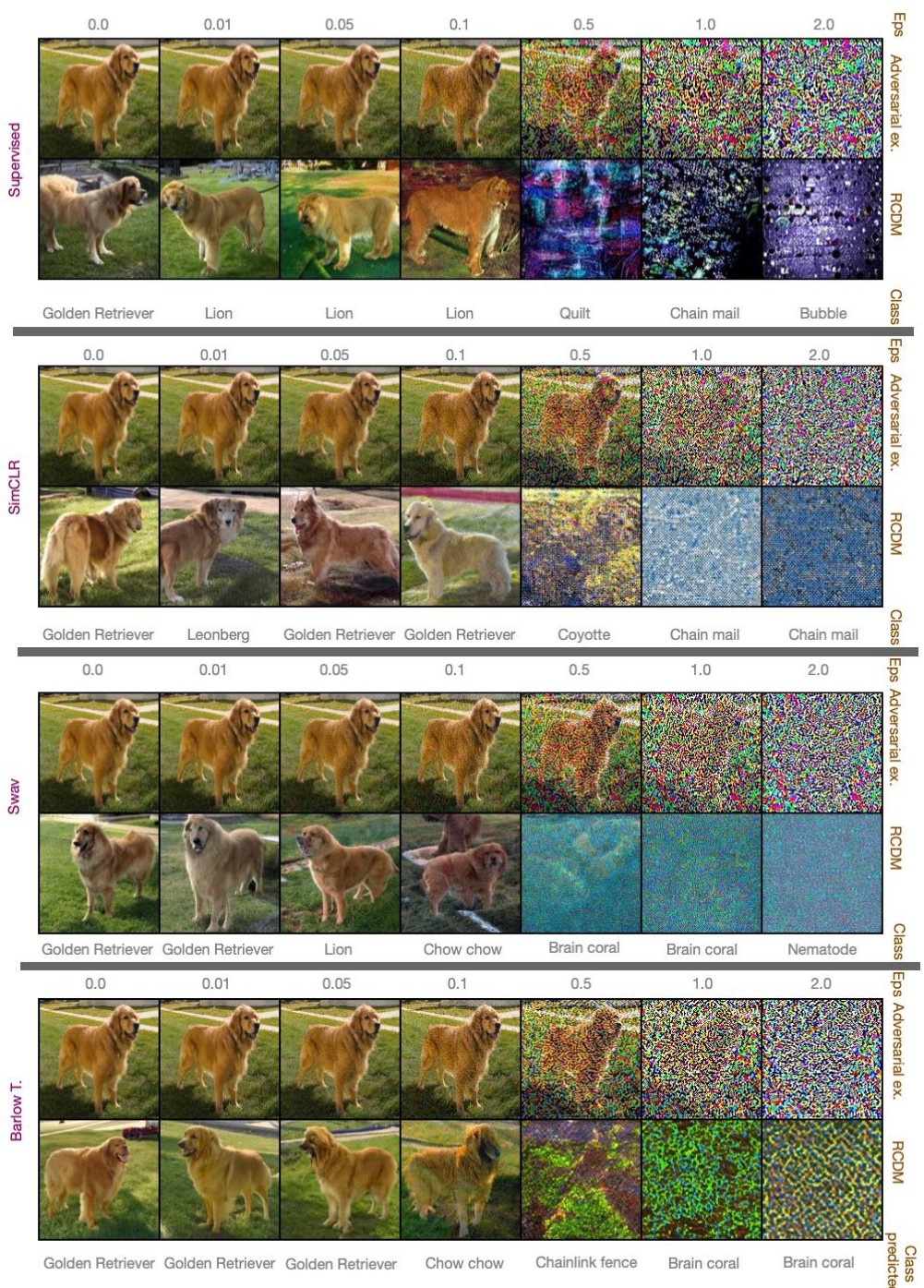

Figure 29: **Visualization of adversarial examples** We use RCDM to visualize adversarial examples for different models. For each model, we trained a linear classifier on top of their representations to predict class labels for the ImageNet dataset. Then, we use FGSM attack over the trained model using a NLL loss to generate adversarial examples towards the class lion. For each model, we visualize adversarial examples for different values of $\epsilon$ which is the coefficient used in front of the gradient sign. In the supervised scenario, even for small values of epsilon which doesn't seem to change the original image, the decoded image as well as the predicted label by the linear classifier becomes a lion. However it's not the case in the self-supervised setting where the dog still get the same class or get another breed of dog as label until the adversarial attack becomes more visible to the human eye (For $\epsilon$ value superior to 0.5).

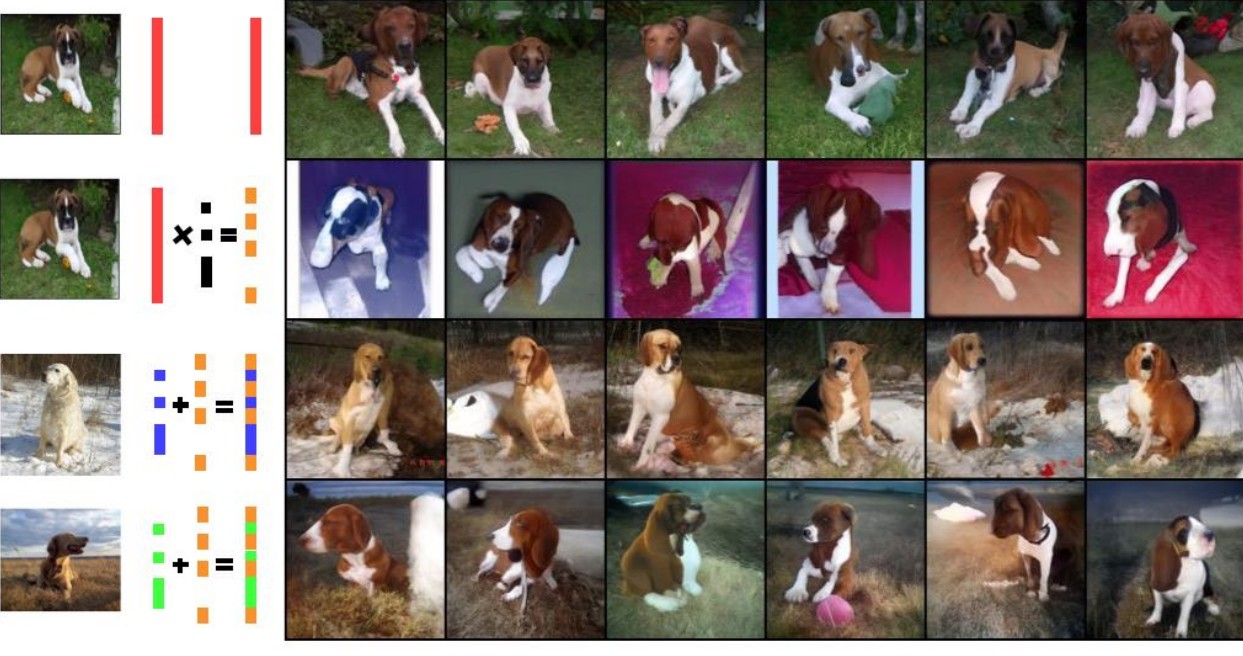

Figure 30: **Background suppression and addition** Visualization of direct manipulations over the representation space. On the first row, we used the full representation of the dog's image on the top-left as conditioning for RCDM. Then, we find the most common non zero dimension across the neighborhood of the image used as conditioning. On the second row, we set these dimensions to zero and use RCDM to decode the truncated representation. We observe that RCDM produces examples of the dog with a high variety of unnatural background meaning that all information about the background is removed. In the third and forth row, instead of setting the most common non zero dimension to zero, we set them to the value of corresponding dimension of the representation associated to the image on the left. As we can see, the original dog get a new background and a new pose.

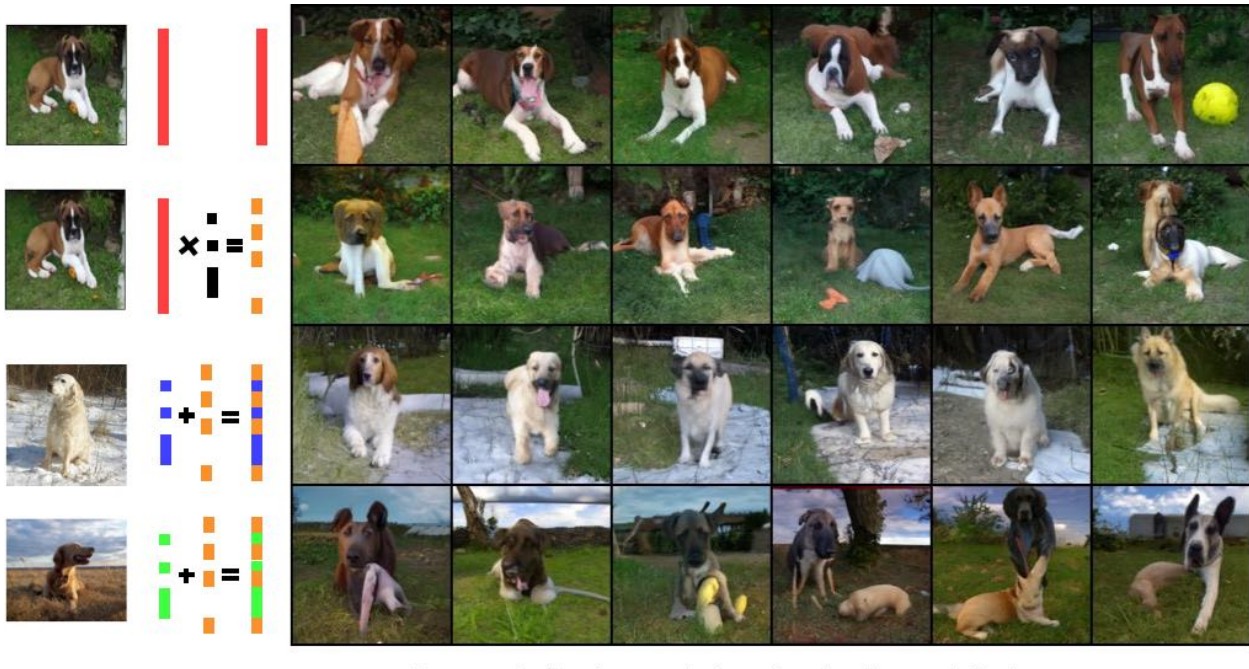

■ zero mask of **least** common indices where dim of representation is non zero
■ Least common dim of ■ where dim of representation is non zero

Figure 31: Same setup as Figure 30 except that instead of using the most common non zero-dimensions as mask, we used the least common non-zero dimensions as mask. On the second row, we observe that some information about the original dog is removed such that in each column, we get a slightly different breed of dog while the background stay fixed. On the third and forth row, we saw that the information about the background (grass) is propagated through the samples (which was not the case in Figure 30).

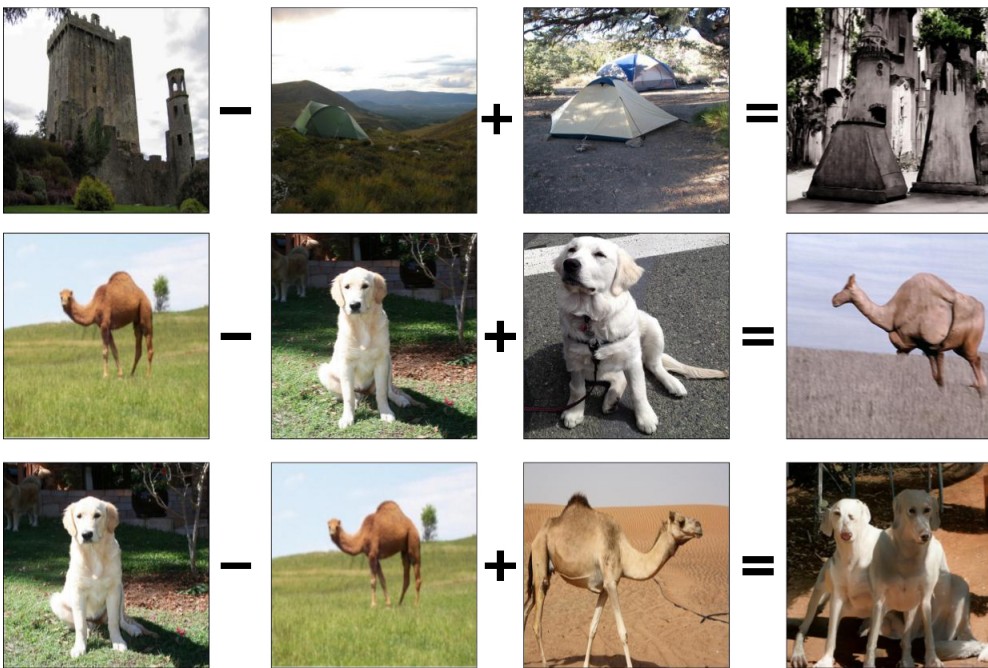

Figure 32: Algebraic manipulation of representations from real images (left-hand side of =) allows RCDM to generate new images with novel combination of factors. Here we use this technique with ImageNet images, to attempt background substitutions.

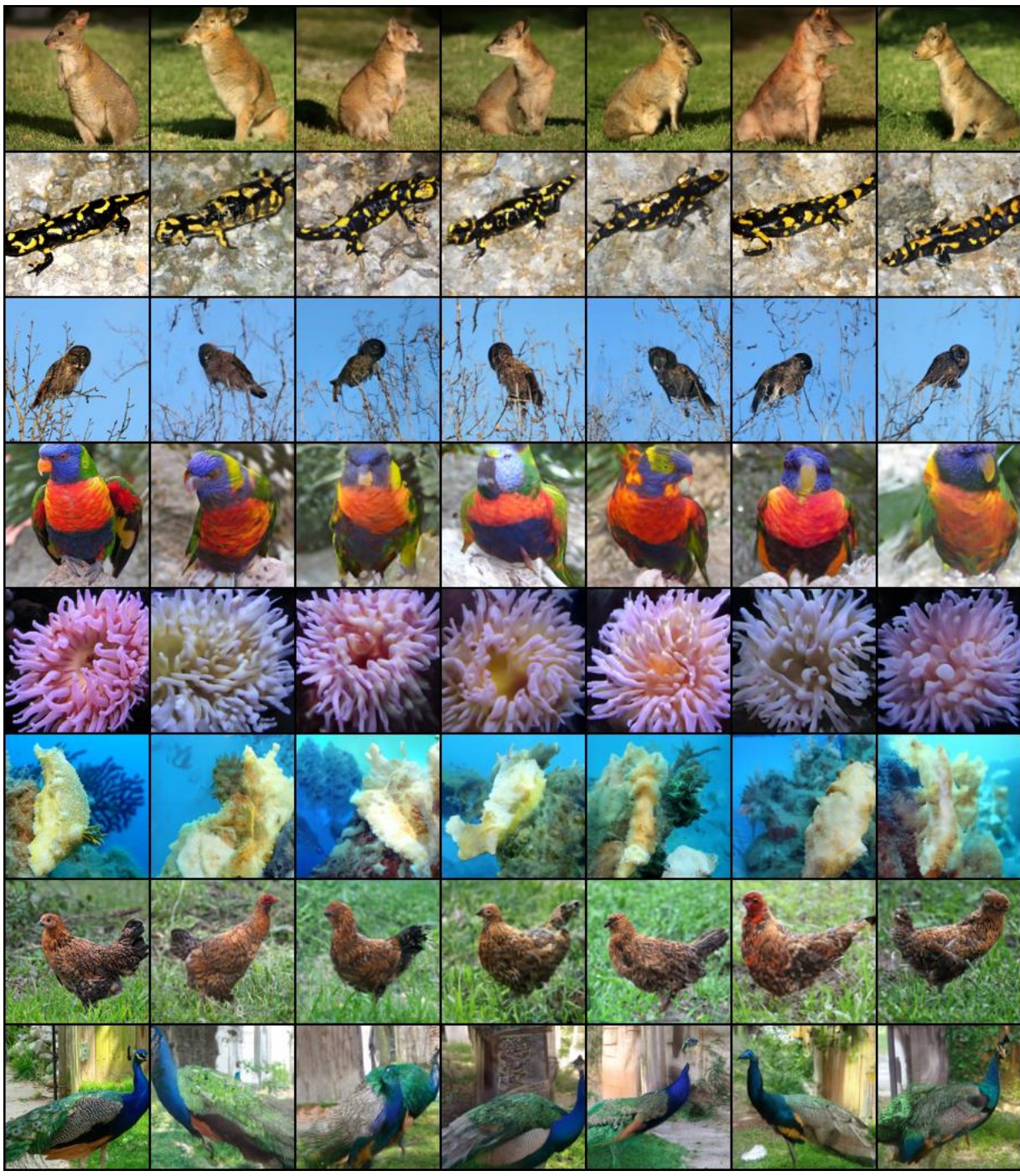

Figure 33: Conditional generation with RCDM using representation extracted from a VIT-B 16 trained with Dino. This experiment shows that RCDM is able to successfully use the representation extracted form different kinds of architectures.

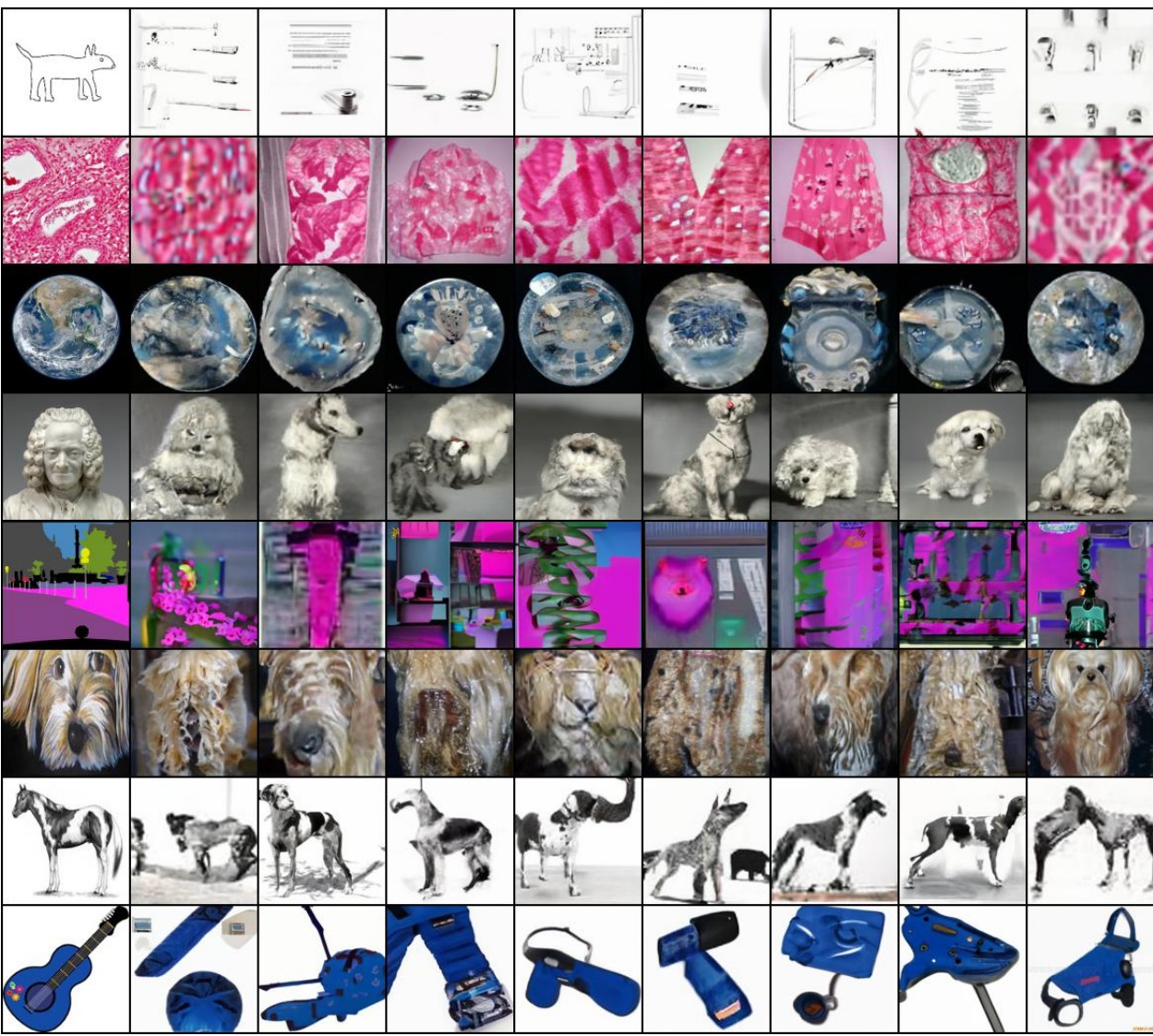

Figure 34: Conditional generation with RCDM using representation extracted from a Resnet50 trained with VicReg using only cropping as data augmentation (thus discarding all transforms related to color change). This experiment shows that training an SSL model without learning any invariances to colors lead to learn only statistics about the colors in the representation. We can clearly see that the samples generated from the guitar are clearly following the same colors statistics as the conditioning but totally fail to reconstruct anything related to the shape information. The source for the picture of the earth is NASA.

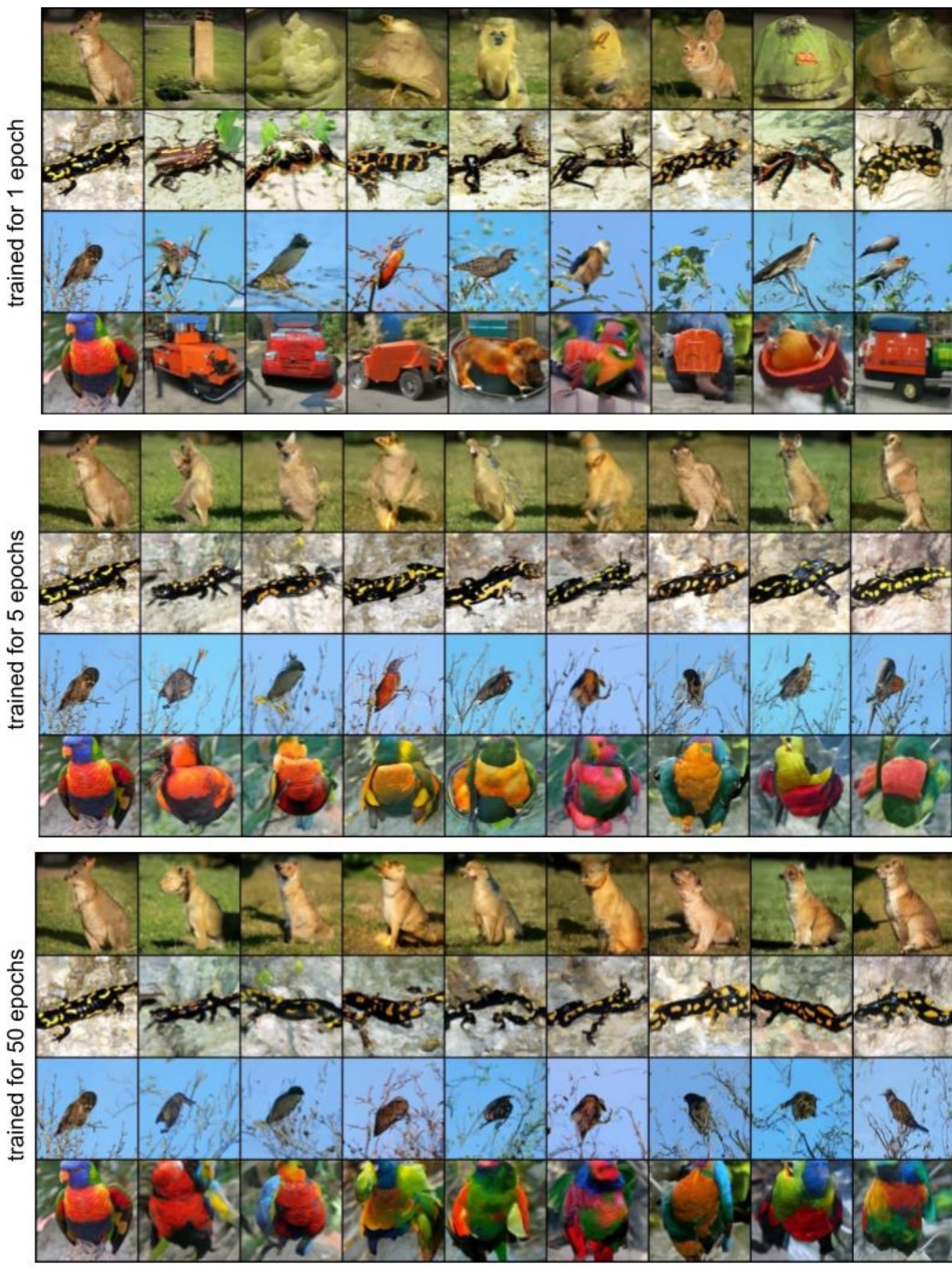

Figure 35: Conditional generation with RCDM using representation extracted from a Resnet50 trained with VicReg for 1, 5 and 50 training epochs (a new RCDM generator is trained fully for each case). This experiment shows that the SSL model first (after 1 epoch) learns to retain mostly information about color and texture in its representation (see e.g. how conditioning on the parrot representation yields *vehicles* with similar color-themes). It encodes accurate information on the more precise shape only later in training.

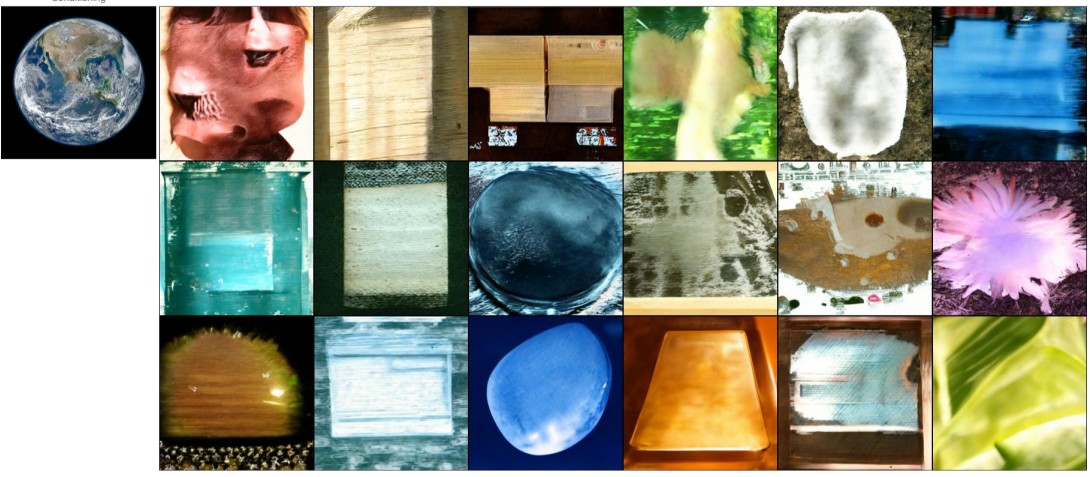

(a) Earth from an untrained representation (Random initialized Resnet 50).

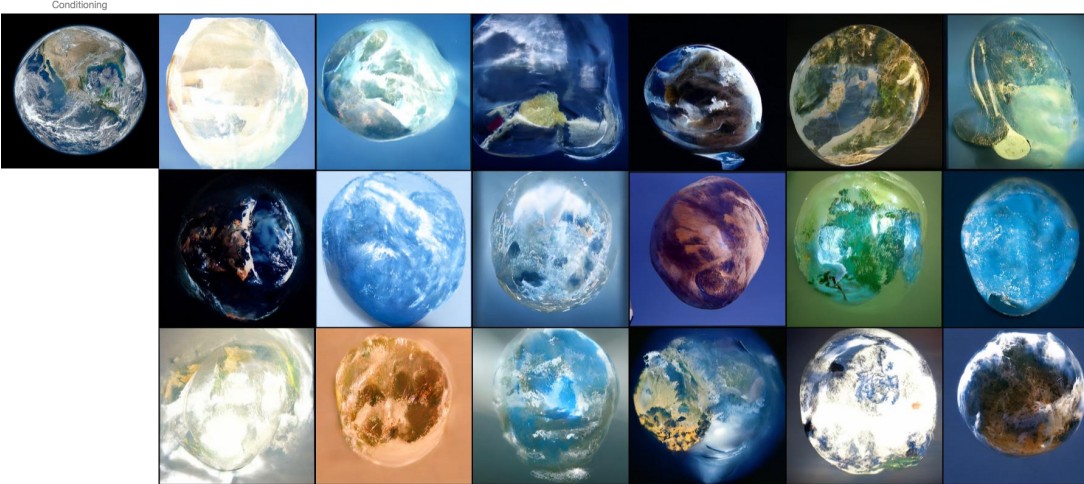

(b) Earth from a supervised representation (Pretrained resnet50 on ImageNet)

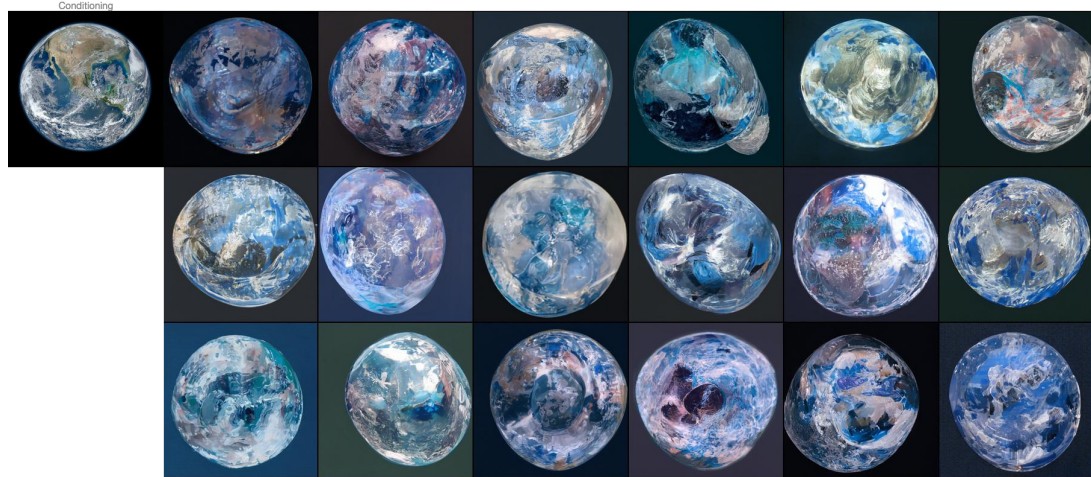

(c) Earth from a SSL representation (Dino Resnet50 backbone).

Figure 36: Different samples of RCDM conditioned on a satellite image of the earth (source: NASA). We show the samples we obtained in a) when using a random initialized network to get representations, b) when using a pretrained resnet50, c) when using a self supervised model (Dino).

