# OpenReview forum: "High Fidelity Visualization of What Your Self-Supervised Representation Knows About"
_TMLR — Accepted by TMLR_

### Review · Reviewer_GRCY · 2022-04-27

**Summary Of Contributions:**

This paper aims to understand the information captured by neural network representations (specifically, self-supervised learning (SSL) methods) by going beyond the standard downstream classification task, and instead visualizing the representations with a class-conditional diffusion model (RCDM). They find that this RCDM helps to uncover the fact that SSL representations are robust to adversarial perturbations and demonstrate invariance to certain kinds of transformations, though with varying degrees across SSL methods.


**Broader Impact Concerns:**

The authors did not include a Broader Impacts statement in their submission. They should discuss the potential harmful impacts of generative models, especially with their potential for misuse in generating deepfakes and synthetic examples of people that may violate their privacy.

**Requested Changes:**

- As mentioned in the “Weaknesses” section, I think the paper would benefit from a more clear explanation
- In Section 3, the authors discuss how to develop an effectively “instance-conditional” generative model which takes in a representation h and learns a conditional probability distribution p(x|h). This section was quite confusing. I think the authors should either commit to a full-length discussion on (conditional) diffusion models, how they are trained, how they generate samples, etc. and refer to their schematic in Figure 1, or choose to leave out such details entirely. The current hybrid approach in Section 3 where the details are extremely high-level actually made it harder to understand what was going on, even though at the end of the day the point of the section was to elaborate upon the authors’ decision to replace a particular activation layer.
- On that note, I think Figure 1 could be explained in more detail. The authors contextualize RCDM within 2 alternative frameworks (unconditional generation, gradient-based controllable generation), and it would have been helpful to highlight this more in the main text.
- How much are the results specific to ImageNet? While it is important to evaluate SSL methods on this dataset, I think it’s important to demonstrate that the method works well on other datasets used in SSL such as CIFAR-10/CIFAR-100, CityScapes.
- Although the authors performed a wide variety of experiments with the visualization of learned representations, the conclusions they drew seemed somewhat inconclusive. For example, if SSL representations do preserve information such as the object scale and color palette of the background, what are the implications of this? What should we do with the knowledge that SSL representations seem to demonstrate some signs of adversarial robustness to perturbations in the input space? I would have appreciated if the authors could have contextualized their findings better for either recommendations in practice, developments of some quantitative metrics to compare between representations, etc. At the moment, the experiments are largely qualitative visual comparisons, which are hard to assess.
- Along this line, I think the paper would benefit from additional experiments that use the insights drawn from RCDM to verify empirical phenomena in downstream tasks. For example, if it is indeed the case that the SSL representations are “invariant to vertical shifts,” does this observation still hold in a downstream task where the test images are shifted downwards? Experiments along these lines would be extremely helpful for convincing the reader that the qualitative observations made from using a tool like RCDM is actually useful/predictive of certain properties of the representations in practice.
- Addition of broader impacts section (see below).


**Strengths And Weaknesses:**

Strengths:
- The paper gave a nice overview of related works, although at times they seemed to cover too much ground. For example, I didn’t think it was necessary to cover all families of generative models (e.g. PixelCNN++), as they weren’t relevant to the method and were distracting at times.
- The paper also conveyed a lot of empirical results (in terms of samples, interpolations, etc.) throughout the main text and the appendix. The authors clearly demonstrated that RCDM worked well as an instance-conditional generative model for their purposes.

Weaknesses:
- The exact contribution of this work was unclear to me. The RCDM model is a pre-existing conditional variant of OpenAI’s diffusion model from (Dhariwal & Nichol 2021), just with a very minor architectural change (where the Group Normalization activations have been replaced with conditional batch normalization layers). Thus I’m not sure that this is a contribution of the work, and the fact that its quality is on-part with “state-of-the-start models” as mentioned on page 2 is not surprising (as well as the analysis in Section 4).
- This implies that the main contribution of the work is centered around analyzing what kinds of information is encoded in SSL representations/embeddings, starting in Section 5. However, I felt that this was actually the weakest part of the paper. Although the authors examine 4 SSL algorithms (VicReg, Dino, Barlow Twins, SimCLR)
- Parts of the text/paper were unclear. For example, what is the distinction between n SSL “projector embedding” and “SSL representation?” This wasn’t clarified until Section 5, which seems to be too late in the main text.
- Parts of the exposition could definitely be improved. See “Requested changes” section.
- There are some examples of overclaiming here: e.g., the OOD views generated by RCDM are noticeably worse quality than those from in-domain (Figure 2).
- I think the authors should demonstrate that their approach works on datasets beyond ImageNet, since the method is pitched as a “general-purpose, plug-and-play visualization tool.” This goes beyond simple OOD synthesis experiments in Figure 2. I would expect the method to work well since diffusion models tend to work well on a variety of different (and complex) image datasets.
- One thing that wasn’t clear to me was what the takeaway should be, even after visualizing such SSL representations. I have elaborated on this in the “Requested changes” section.

---

> ### Author Response · Authors · 2022-05-24
> **Answer to Reviewer GRCY**
>
> Thank you for your review. Before reading our detailed answer below, we invite you to read the general answer to the reviews posted above as well as the list of changes we did in the updated version of our paper. In particular, as we emphasize that our core contribution is to provide a visualization method to better grasp what SSL methods learn. We never claimed that our contribution was to introduce a new type of conditional generative model nor to propose an improvement over OpenAI's diffusion model. Our main goal and contribution is the demonstration that we can actually use the OpenAI's diffusion model to produce high fidelity visualization. This notion of “fidelity” or “faithfulness” – how closely the representation of generated points match the conditioning – is an essential point, underlined through the title, abstract and introduction. We also highlight that we provide a thorough validation (qualitative and quantitative) on how faithful our generated samples are, a study that was e.g. missing from (Dhariwal & Nichol 2021) and that is crucial for drawing conclusions regarding SSL methods (or any representation used for conditioning) through visualizations.
>
> > “The exact contribution of this work was unclear to me. The RCDM model is a pre-existing conditional variant of OpenAI’s diffusion model from (Dhariwal & Nichol 2021), just with a very minor architectural change (where the Group Normalization activations have been replaced with conditional batch normalization layers). Thus I’m not sure that this is a contribution of the work, and the fact that its quality is on-part with “state-of-the-start models” as mentioned on page 2 is not surprising (as well as the analysis in Section 4).
>
> > “In Section 3, the authors discuss how to develop an effectively “instance-conditional” generative model which takes in a representation h and learns a conditional probability distribution p(x|h). This section was quite confusing. I think the authors should either commit to a full-length discussion on (conditional) diffusion models, how they are trained, how they generate samples, etc. and refer to their schematic in Figure 1, or choose to leave out such details entirely. The current hybrid approach in Section 3 where the details are extremely high-level actually made it harder to understand what was going on, even though at the end of the day the point of the section was to elaborate upon the authors’ decision to replace a particular activation layer.”
>
> > “As mentioned in the “Weaknesses” section, I think the paper would benefit from a more clear explanation”
>
> We definitely agree that having an entire dedicated section (section 3.) to motivate the use of the OpenAI’s diffusion model was not necessary and confusing. Since our contribution is not about the model, but about analyzing the fidelity of the samples produced by OpenAI's diffusion model when conditioned on a large representation vector, we decided to merge Section 3 and Section 4 to clarify our message. Nonetheless, we think that our entire analysis on fidelity of the conditioning is novel and should be considered as at least a minor contribution on its own right. In fact, having a generative model that is able to generate high quality images doesn’t mean that it will be able to generate high fidelity images, the model could have totally ignored the conditioning, especially on large representation vectors. So the result that is “surprising” in this paper is not the quality of the samples, but **the fidelity** of the samples.
>
> > -This implies that the main contribution of the work is centered around analyzing what kinds of information is encoded in SSL representations/embeddings, starting in Section 5. However, I felt that this was actually the weakest part of the paper. Although the authors examine 4 SSL algorithms (VicReg, Dino, Barlow Twins, SimCLR)
>
> Our core contribution lies in how to probe SSL representations in novel ways with a conditional diffusion model to gain clear insights (claim 2) We would like to invite you to read the list of evidence that supports this claim2 in our general answer.  We don’t really understand the weakness statement since we provide a powerful visualization tool that is essential for debunking misleading claims that linger in the self-supervised community (such as SSL representations being invariant to the data augmentations used during training).
>
> > “Parts of the text/paper were unclear. For example, what is the distinction between SSL “projector embedding” and “SSL representation?” This wasn’t clarified until Section 5, which seems to be too late in the main text.”
>
> You are right, we have now defined this SSL specific terminology right in the introduction.

---

> > ### Author Response · Authors · 2022-05-24
> > **Answer to Reviewer GRCY**
> >
> > > “There are some examples of overclaiming here: e.g., the OOD views generated by RCDM are noticeably worse quality than those from in-domain (Figure 2).”
> >
> > It is expected there will be a difference since it is OOD data coming from various datasets that are not as object centric as ImageNet. However, we agree that any qualitative judgment will be difficult to agree upon .We removed the following sentence from the paper “and is also suited for generating images conditioned on out-of-distribution representations “, as OOD generation quality is not the focus of this paper.
> >
> >  > “I think the authors should demonstrate that their approach works on datasets beyond ImageNet, since the method is pitched as a “general-purpose, plug-and-play visualization tool.” This goes beyond simple OOD synthesis experiments in Figure 2. I would expect the method to work well since diffusion models tend to work well on a variety of different (and complex) image datasets.” “How much are the results specific to ImageNet? While it is important to evaluate SSL methods on this dataset, I think it’s important to demonstrate that the method works well on other datasets used in SSL such as CIFAR-10/CIFAR-100, CityScapes.”
> >
> > This is an interesting suggestion but we feel it would achieve little in supporting claim 1 or claim 2. ImageNet is the reference for SSL methods whose analysis is our main focus; also from its diversity it is usually much more difficult than CIFAR-10. As you noted, it is also known that diffusion models work well on different datasets, so there is nothing to make us believe that diffusion models will fail on CIFAR10. Now, if your question was about “how are SSL insights generalizing to other datasets” we agree that this is an important question. We added an experiment in Figure 28 in the Appendix that shows the same conclusions about the projector being more invariant than the representation when using images outside ImageNet.
> >
> > > “I think Figure 1 could be explained in more detail. The authors contextualize RCDM within 2 alternative frameworks (unconditional generation, gradient-based controllable generation), and it would have been helpful to highlight this more in the main text.”
> >
> > We don’t consider that showing the limitation of gradient-based controllable generation is a contribution of our work (this result is already known). In fact, we removed this section since it was deemed confusing by other reviewers and that it did very little in supporting the claims we made in this paper. Instead, we compare RCDM with DIP in Figure 7 since this method was used by Zhao et al. (2021) to visualize SSL representation.  We also added several references in the related work to better contextualize RCDM with other methods.
> >
> > > “Although the authors performed a wide variety of experiments with the visualization of learned representations, the conclusions they drew seemed somewhat inconclusive. For example, if SSL representations do preserve information such as the object scale and color palette of the background, what are the implications of this? What should we do with the knowledge that SSL representations seem to demonstrate some signs of adversarial robustness to perturbations in the input space? I would have appreciated if the authors could have contextualized their findings better for either recommendations in practice, developments of some quantitative metrics to compare between representations, etc. At the moment, the experiments are largely qualitative visual comparisons, which are hard to assess.”
> >
> > We entirely agree with your point. The entire motivation of our study (as reminded in our general answer) is to debunk the commonly accepted fact that the output of the backbone learns invariant representations. This is an assumption that many recent studies rely on to build further downstream task solutions based on SSL trained deep networks. Hence, we believe that (among all our other observations) highlighting that only the output of the projection learns invariants (and too many of them) while the output of the backbone remains pretty much input equivariant is crucial for the community.
> >
> > > “Along this line, I think the paper would benefit from additional experiments that use the insights drawn from RCDM to verify empirical phenomena in downstream tasks. For example, if it is indeed the case that the SSL representations are “invariant to vertical shifts,” does this observation still hold in a downstream task where the test images are shifted downwards?

---

> > > ### Author Response · Authors · 2022-05-24
> > > **Answer to Reviewer GRCY**
> > >
> > > >”Experiments along these lines would be extremely helpful for convincing the reader that the qualitative observations made from using a tool like RCDM is actually useful/predictive of certain properties of the representations in practice.”
> > >
> > > We invite you to read the listed evidence in support of our claim 2) in the general answer. This list describes why these qualitative observations are essentials from a SSL point of view. Furthermore, our qualitative observations are actually already predictive of certain properties of the representations in practice. We observed visually that the representations at the projector level are actually too invariant, meaning the information needed to predict a class label is often lost. In practice, when using these representations for a downstream classification task, the validation accuracy can be as low as 30\% on these projector representations while we get more than 60% accuracy when using the representation at the backbone level. Thus, instead of training a KNN or a linear probe for classification on top of the representations, we can just visualize them with RCDM and know in advance if the representation will be suitable or not for classification. We added a small paragraph in the first part of Section 4 which highlights how RCDM visualizations are predictive of certain properties of a downstream classification task,
> > >
> > > > ​​Addition of broader impacts section (see below).
> > >
> > > We added a broader impact section at the end of the main part of the paper.

---

> > ### Comment · Reviewer_GRCY · 2022-06-14
> > **response**
> >
> > Thank you for the reply! I've read the updated PDF, the authors' responses, and the other reviewers' comments (and the corresponding replies). I think the updated version is more clear and representative of the paper's actual contribution (e.g. emphasizing that it is for visualizing the features, removing some confusing claims, etc.), which was helpful to see. However, I still think this paper really needs an experiment showcasing that the insights uncovered by the visualization tool translate to a downstream task or experiment as in my original review (which was also mentioned by Reviewer W3ik). This would greatly strengthen the paper and make the claims more convincing! I'm happy to change my score if the authors can address this point.

---

> > > ### Author Response · Authors · 2022-06-20
> > > **Update with required experiment**
> > >
> > >   Your suggestion is well-taken, thank you. To show that qualitative insights derived from RCDM visualization translate to measurable effects on downstream tasks, we have updated the paper by adding two new figures (Figure 12 and Figure 13) along with a small explanatory text at the beginning of Section 4. In these experiments, we show that simple visual inspection of the diversity of samples produced by RCDM is predictive of the degree of certainty or uncertainty that a linear classifier trained on that representation will display. We also show,  when conditioning on the representation of a given image, that if all K samples generated by RCDM  are recognizable instances of the object on the image, then the original image (used for conditioning) will typically be correctly classified by a downstream classifier, else it won’t. With these experiments, we clearly show a correlation between the insights we gain through visualization and the performances we get on a subsequent downstream task. We hope that this latest update answers your concerns.

---

### Review · Reviewer_BkYX · 2022-05-02

**Summary Of Contributions:**

This paper presents a technique for visualizing representations of fixed, pre-trained, and mostly self-supervised (SSL) image networks. Specifically, the learned representation of a neural network can be mapped back on the image space by using a conditional diffusion model, and by representing multiple samples corresponding to the same SSL representation, the invariances learned by the network which shall be interpreted can be visualized.


**Broader Impact Concerns:**

Since the paper provides a fairly powerful generative approach, it also includes all the considerations one needs to make when publishing such models. Although this has been discussed in specific publications, I think the paper should at least briefly summarize the potential societal problems associated with such models (spread of misinformation, deliberate manipulation, misrepresentation of data, ...).

**Requested Changes:**

- See weaknesses, annotated as follows:
	- (1) needs to be answered in order to recommended the paper for acceptance
	- (2) not absolutely required, but would strengthen the work


**Strengths And Weaknesses:**

**Strengths**

- This paper touches on an interesting topic: the interpretability of learned representations is and remains an important problem, especially for large-scale SSL models such as CLIP, DINO, etc. Gaining insights into these representations via visualizations has potential (and is also an established approach).

- Good qualitative samples and extensive experiments.
- Analyzing the image manipulation capabilities of the approach is an interesting extra and seems promising (albeit more for a dedicated generative model).
- The work is very related to the concurrent work unCLIP (a.k.a. DALLE-2, https://arxiv.org/abs/2204.06125), but with a different focus: it does not focus on the generative capability/synthesis quality, but rather on the interpretability that comes with this application.
  -  A side question, since this work also does some synthesis (especially in the appendix): How does the choice of SSL mesh affect the **generative** capability of the model? That is, suppose I want to build a generative model using RCDM. Which network should I choose, and how does this affect the generative application?

**Weaknesses**
-  The approach is not new, other conditional-generative approaches (as mentioned in the section on related work) have addressed the same problem. The paper would benefit from a direct comparison of these approaches, and it would be worth discussing whether such a comparison could also make clear which of the approaches has which advantages and disadvantages. (1)
-  A somewhat mixed comment: I think this paper suffers a bit from trying to do the balancing act between being an "interpretability paper" and showing (very decent) generative results (mainly in the appendix). In my opinion, it would benefit greatly from focusing on only one of these two aspects. The latter also allows for very interesting insights, e.g. what kind of "representation network" is needed to achieve the best generative performance. (2)
- The conditioning mechanism via the conditional batchnorm seems to be a bit ad-hoc. Is this more powerful than the mechanism already built into the ADM Unet (one would just have to leave out the "class embedding" layer)? (1)
- I am not quite sure what exactly we learn in this paper: What **new** insights does the method offer, especially compared to [1] and [2] (when trained with a similar computational budget) (1)?  How can we be sure that the DM does not "ignore" the conditioning (and invent new details during synthesis) (1)?
  - $\rightarrow$  Is there any way to "prove" that the initial sample contains the same information as the representation from which it was predicted? (1)
- Using a deterministic mapping from the input to the latent diffusion space (such as an inverted DDIM sampler, e.g., in https://arxiv.org/abs/2105.05233), this method could find an **explicit representation** of the invariances (rather than visualizing them implicitly via multiple decodes). I think it would be interesting to have a visualization of this. (2)
- I would be very interested to see results obtained with classifier-free guidance (https://openreview.net/pdf?id=qw8AKxfYbI). How do such "post-hoc" truncation procedures (which also exist for other types of generative models) change/influence the interpretation of the results? (2)

*References*
- [1]  Charlie Nash, Nate Kushman, and Christopher KI Williams. Inverting supervised representations with
autoregressive neural density models. In The 22nd International Conference on Artificial Intelligence and
Statistics, pp. 1620–1629. PMLR, 2019.
- [2] Robin Rombach, Patrick Esser, and Björn Ommer. Making sense of cnns: Interpreting deep representations
and their invariances with inns. In Computer Vision–ECCV 2020: 16th European Conference, Glasgow,
UK, August 23–28, 2020, Proceedings, Part XVII 16, pp. 647–664. Springer, 2020

---

> ### Author Response · Authors · 2022-05-24
> **Answer to Reviewer BkYX 1/2**
>
> Thank you for your review. Before reading our detailed answer below, we invite you to read the general answer to the reviews posted above as well as the list of changes we did in the updated version of our paper.
>
> > A side question, since this work also does some synthesis (especially in the appendix): How does the choice of SSL mesh affect the generative capability of the model? That is, suppose I want to build a generative model using RCDM. Which network should I choose, and how does this affect the generative application?
>
> > The approach is not new, other conditional-generative approaches (as mentioned in the section on related work) have addressed the same problem. The paper would benefit from a direct comparison of these approaches, and it would be worth discussing whether such a comparison could also make clear which of the approaches has which advantages and disadvantages. (1)
>
> > A somewhat mixed comment: I think this paper suffers a bit from trying to do the balancing act between being an "interpretability paper" and showing (very decent) generative results (mainly in the appendix). In my opinion, it would benefit greatly from focusing on only one of these two aspects. The latter also allows for very interesting insights, e.g. what kind of "representation network" is needed to achieve the best generative performance. (2)
>
> These are great suggestions. However, we feel that those are not aligned with our main focus which is to understand what SSL methods are learning (rather than providing a truly novel conditional generative model to be used in-place of existing ones). The conditional generative model was selected with the following argument: OpenAI’s diffusion model was the latest model with state-of-the-art generation quality. Hence, to maximize the amount of insights one can gain from generation, we opted to use this model conditionally to fit our needs. We nevertheless had to show that samples produced by this method were faithful to the representation used as conditioning (which was not something guaranteed since the model could have totally ignored the conditioning). We still compared it to IC-GAN (another recent state-of-the-art conditional generative model) to show that our quality and fidelity was competitive.. As you noted, it’s difficult to write a paper in between research domains (interpretability/generative model and SSL). We had to make a choice and we decided to focus on the “interpretability of ssl” this is why we tried to emphasize the interpretability of SSL methods in the main part of the paper while pushing the generative part to the appendix  Nevertheless, we strongly concur that doing a proper comparison with other conditional-generative approaches and analyzing which kind of "representation network" is needed to achieve the best generative performance across different generative tasks and models is a very interesting research direction on its own that we are pursuing as future work.
>
> > The conditioning mechanism via the conditional batchnorm seems to be a bit ad-hoc. Is this more powerful than the mechanism already built into the ADM Unet (one would just have to leave out the "class embedding" layer)? (1)
>
> We wanted to use the same technique as the one in IC-GAN to make a comparison with this model and then decided to stick with it. However, we have also tried the built-in conditioning mechanism of ADM. In this instance we replaced the "class embedding" layer (which takes discrete inputs) by a simple linear layer, since our representations are already continuous, to map to the same dimension of the timesteps. We did not observe any significant differences between the two methods. We added in the paper a mention that both methods can be used, as well as Figure 10 in the appendix that compares both methods.

---

> > ### Author Response · Authors · 2022-05-24
> > **Answer to Reviewer BkYX 2/2**
> >
> > > I am not quite sure what exactly we learn in this paper: What new insights does the method offer, especially compared to [1] and [2] (when trained with a similar computational budget) (1)? How can we be sure that the DM does not "ignore" the conditioning (and invent new details during synthesis) (1)?  Is there any way to "prove" that the initial sample contains the same information as the representation from which it was predicted? (1)
> >
> > This is a great and critical question! Concerning the new insights (which are exclusively targeted towards self-supervised representations), we invite you to read the list of evidence that supports our claim 2) in our general answer.  One of the main scientific value of our work is to showcase to the research community in SSL how to probe SSL representations (with a given visualization method that could be ours or [1,2]) in novel ways (visualizing invariances to specific augmentations, robustness and structure) to gain clear insights. Concerning how to be sure that our generative process doesn’t ignore the conditioning and that samples produced with this method contained the same information as the representation from which it was predicted, please refer to the list of evidence that supports our claim 1).
> >
> > > “Using a deterministic mapping from the input to the latent diffusion space (such as an inverted DDIM sampler, e.g., in https://arxiv.org/abs/2105.05233), this method could find an explicit representation of the invariances (rather than visualizing them implicitly via multiple decodes). I think it would be interesting to have a visualization of this. (2)
> >
> > Thank you for the suggestion, however we do not really grasp how visualizing the latent diffusion space could help us. Once we get the associated point in the latent diffusion space with the inverted DDIM sampler, for a variety of images, what would provide us with the explicit representation for the invariances? As the inverted DDIM is deterministic, those representations would not vary given the same image.
> >
> > > I would be very interested to see results obtained with classifier-free guidance (https://openreview.net/pdf?id=qw8AKxfYbI). How do such "post-hoc" truncation procedures (which also exist for other types of generative models) change/influence the interpretation of the results? (2)”
> >
> > Classifier-free guidance is not really “post-hoc” since it still needs a specific training, even if it consists simply in dropping randomly the class conditioning during training. We could probably use a similar technique by randomly dropping our conditioning on representations during training; however it’s not clear for us that we will get similar benefits as the one we observe with class conditioning and classifier-free guidance. Classifier (free) guidance techniques seem to work well because they are constraining the sampling process in specific directions in the diffusion latent space (this constraining is probably very important since a specific class can be associated with thousands of possible images, thus many possible directions). However, our approach is already strongly constraining the sampling process by using large continuous vectors of representation ( it’s extremely unlikely that two images will be mapped to the same representation, thus the sampling process will already be biased towards a very specific direction). As shown in our paper, this way of constraining the reverse diffusion process works very well since we are able to recover very many aspects of the original image used as conditioning.

---

### Review · Reviewer_W3ik · 2022-05-09

**Summary Of Contributions:**

This paper presents a new approach for visualizing learned representations with a conditional diffusion model. Given a fixed representation h = f(x) (from any method), a conditional diffusion model is trained on a given dataset (ImageNet) to model p(x | h). By sampling several images from p(x|h), we can then better understand what the representation is sensitive to (similar across samples) vs. invariant to (different across images). The visualization approach is applied to SOTA SSL methods Dino, Swav, SimCLR, Barlow Twins and supervised learned representations. Qualitatively, this work shows how different representations are sensitive to different things and how visualization can show the robustness of representations to adversarial perturbations, and quantitatively the resulting unconditional generative model performs well.

**Broader Impact Concerns:**

There was no broader impact statement. With any work on generative models, I believe it's important to carefully detail what datasets you are using to train it (e.g. ImageNet for your conditional diffusion models), and highlight how you do or don't expect it to fail on OOD inputs. In particular in your work you argue that your inversion works well on OOD inputs, but if images are far from ImageNet examples I do believe the method performs worse. There may also be many other naturalistic images that have the same hidden representaiton as an ImageNet image but are not in the dataset you trained on, and thus inversions will be heavily biased to be closer to ImageNet samples.

**Requested Changes:**

Critical:
* While the reconstructions are high quality, I think this paper is still missing an experiment which can show us something we didn't know before about a learned neural network representation. Leveraging these visualizations to perform some downstream task (e.g. if a human is able to detect whether or not the network will get the correct answer on an adversarially perturbed input) would help to strengthen the argument that thsese visualizations are useful.
* Baselines and discussion of related work: the related work needs to be expanded to cover the work of Dosovitsky & Brox, Mahendran & Vedaldi, Nguyen et al., and the DeepDream/feature viz work of Olah et al at a minimum. It would be great if there were additional comparisons with these approaches to highlight that they cannot yield the same qualitative conclusions as your proposed method (e.g. around invariance to vertical shift but sensitivity to scale). I also feel an explanation or further experiments to highlight why the baselines are so bad in Appendix A is critical.
* Appendix A:  I don't think the argument that gradient-based methods only occupy a K-dimensional subspace is valid (see more comments below). An extended proof or discussion is needed.

The below are a mix of suggested change and notes as I went through the paper (organized by section).

Intro:
* Intro: it may be worth mentioning that the modern family of SSL methods are similar to prior work like metric learning with triplet losses and exemplar CNNs/instance discrimination that have both of the criteria you listed (objective depends on properties of the embedding not reconstruction, and leverages data augmentation to encourage invariance).
* I don’t think “s.a” is a standard, please write out as “such as”
* For the observations mentioned in the intro, is (ii) about post or pre-projector SSL representations?

Related work:
* “Successful interpretation…. first few layers of a DN”: what does successful mean here? I think an issue throughout the paper is that there are vastly different definitions/success criteria for “understanding” so it may be worth fleshing out exactly what was understood in these earlier works and how they failed when applied to higher-level layers of DNNs
* I believe the characterization of prior work that inverts DNNs with DIP is inaccurate, at least looking at Zhao et al. (2021), they minimize d(f(x), f(g(w))) where w are the weights of the generator and the loss is in feature space. Qualitatively your results in Appendix A are far worse than Figure 2 of Zhao et al. (2021). Any ideas what is leading to this discrepancy?
* Deterministic visualization methods: almost all the methods listed have some stochastic component based on the initialization of the random image or random weights that are being optimized. There’s no baseline/comparison/discussion of how one could also use these methods to form a diverse set of images given a representation
* There is a tremendous amount of work on studying learned representations through post-hoc inversion and visualization methods, and I would not expect this paper to cover all of them. However, I do think there are a number of fundamental papers that were missed that should be briefly discussed:
* Mahendran & Vedaldi (2015, 2016): motivates the notion of natural pre-images, constructs a careful set of regularizers to yield good inversions that are more natural (e.g. jittering, smoothness, texture/style regularizer). There’s no discussion of DeepDream (2015): https://en.wikipedia.org/wiki/DeepDream and   Olah et al.’s work on feature visualization and activation atlases (e.g. https://distill.pub/2017/feature-visualization/), this Distill paper has a great table under “The Spectrum of Regularization” that mentions several other papers that are not discussed here, e.g. Nguyen et al. 2016a that uses a GAN generator to synthesize preferred inputs (but does not draw samples from the conditional, instead they optimize directly for the latent), Nguyen et al. 2016b that runs MCMC with conditional guidance based on features.
All of these approaches that optimize in image space with extra regularizers can be used to generate a diverse set of natural pre-images that could then be evaluated and compared against.
* Dosovitskiy & Brox (2016) have two papers, one training generators with L2 loss which results in blurry reconstructions, the other training with GAN discriminator losses that produces better sample quality. Here they mention that their method does not work to sample diverse sets of pre-images, citing this would be great motivation for the current paper!
* Lucic et al. (https://arxiv.org/abs/1903.02271): uses SSL to improve the quality of a generative model by clustering data using SSL + small amounts of labeled data and training a GAN conditioned on these learned clusterings. It’s similar in spirit to your work but it’s a GAN and leverages conditioning on the discrete cluster instead of continuous representation embedding.

3: Conditioning a diffusion model on representation h
   * This is the first time S(h) appears and it is not yet defined (you could define it when you introduce notation in the intro)
   * The density p(x) is not a good statistic to measure whether a sample is realistic looking (see: https://arxiv.org/abs/1810.09136), so letting S(h) be the set of sufficiently high likelihood images will not yield good lookign samples. Similarly, I don’t think the description of the goal as sampling from M \int S(h) makes sense when reasoning over high-dimensional continuous spaces.
   * Do you actually use batchnorm and introduce dependences between elements in the batch or use conditional group norm where you modulate the scale and bias of each feature map?
   * You could use an architecture closer to the super-res based approach by having a neural network that outptus a representation-dependent bias in only the first layer (corresponding to broadcasting the vector spatially after a transformation). I believe your approach is a generalization of this, but I’d be interested to see baselines of how different forms of conditioning impact the results.
   * Need to mention what dataset you are training the diffusion model on. ImageNet?

4: Experiments
   * In terms of not memorizing, Figure 2 & Figure 17 are great. The argument in Figure (b) that the samples are “close visually” I don’t find so convincing. In particular the contrast of the diffusion models is often very different from the original, and for the semantic segmentation input, changing the color of a pixel corresponds to changing the class label, and so the resulting samples I’d think about as *very* different even if to our eye they have somewhat similar characteristics. For example, semantic segmentaiton masks on roads will never have splotches of other texture Fig2b, right, and if you trained a diffusion model on segmentation masks it likely would not generate images like this. IMO, Fig 2b highlighted that we should not expect these visualization methods to work well for really OOD inputs (like segmentation masks)
   * Figure 3 should be listed as a Table not a Figure. Fig 3a results are neat and compelling!

5: Analysis
   * Fig 4 is a neat demonstration of how different heads and methods constrain the pre-iamge differently! It would be interesting to pair this with a supervised model to see whether the small differences in the representations of the sampled inputs confuse a classifier that is trained on the learned representation.
   * Fig 6: it’d be useful to note where the classifier fails on the adversarial example in each row (e.g. by having a box around each image with a color red if it’s incorrect). I think there’s also a great opportunity for a behavioral study to show the utility of your method: see whether humans are better able to predict the classifier output by visualizing your reconstructions instead of the original adversarially perturbed input.
   * Fig 7: this was an interesting idea and set of manipulations but I dont’ find the results very convincing. It does seem like this method is good at plugging in different backgrounds, but the clothes (to my eye) in the bottom row don’t match well with the initial set of clothes and the dogs in that image are of poorer quality than most your samples. I think there may be more convincing sets of manipulations in latent space that could highlight this method, e.g. taking an average of a bunch of dog images with clothes, and dogs without, and learning a single vector to add in latent space.
   * Table 3b could use more explanation / discussion in the text as it’s the only quantitative result that involves your representational-conditional model (that’s not the unconditional FID)

7: Reproducibility
   * Would be great to have accessible code to understand some of the model details (like how you handle normalization exactly).

Appendix A:
   * This is a great appendix and highlighting the failure mode of prior methods in the main paper would be useful to motivate your approach.
   * A.2 / Eqn 2: I don’t follow your logic here. The Jacobian of a deep network will change as the input is changed, and thus the space that the iterates span can be D-dimensional if you take many steps
   * A.3: prior work has found that randomc rops + jittering can be very important for the quality of results. Did you try that here as well? Having seen the quality of earlier results (e.g. DeepDream, Mahendran &Vedaldi), I’m surprised how bad the results are in Fig 8



**Strengths And Weaknesses:**

Strengths:
* Due to the excellent sample quality of diffusion models, this work presents the best visualizations of learned representations I have seen.
* The presented method is a straightforward extension of existing work on diffusion models, and enables new applications beyond visualization such as image manipulation.
* The high quality of visualizations as well as ability to successfully invert OOD images allows for more interactive probing and understanding of learned representations
* The paper is well written and the motivation and methods are clear.

Weaknesses:
* Despite the quality of inversions, I found the experiments were unconvincing of the utility of this approach for understanding neural networks. If such visualizations are to help understand neural networks, what have we learned in the qualitative experiments that we did not know previously? I would love to see identification of a failure of some representations that could be exploited by altering inputs to confuse the representation (based off unexpected invariances), or some human study indicating that these visualizations are useful for any task.
* Baselines and discussion of related work: there is tons of work on visualizing representations of DNNs, but there are no alternative methods presented or compared to in this paper. Could all the qualitative insights have come about from a visualization method that produced less realistic inversions? How do these other approaches compare in your setting? Appendix A is a start at evaluating this quesiton, but ther eis a huge gap between the quality of the baselines in Appendix A and published results in this area.
* An additional challenge is that your conditional diffusion models will have to be retrained for every representation. As another idea, I believe you could do classifier guidance in the diffusion context by combining an unconditional p(x) model with a conditional p(h | x) model using something like log p(h|x) = ||f(x) - h_target||^2, i.e. you can use classifier guidance to guide your unconditional sample to have a similar representation as your target without retraining a diffusion model.
* There’s no discussion of how the images that are used to train the conditional diffusion model will bias the inversions. Are failures on OOD inputs due to the representation failing on OOD inputs or the visualization failing?

---

> ### Author Response · Authors · 2022-05-24
> **Answer to Reviewer W3ik**
>
> Thank you for your review. Before reading our detailed answer below, we invite you to read the general answer to the reviews posted above as well as the list of changes we did in the updated version of our paper.
>
> > “​​Despite the quality of inversions, I found the experiments were unconvincing of the utility of this approach for understanding neural networks. If such visualizations are to help understand neural networks, what have we learned in the qualitative experiments that we did not know previously? I would love to see identification of a failure of some representations that could be exploited by altering inputs to confuse the representation (based off unexpected invariances), or some human study indicating that these visualizations are useful for any task.”
>
> > While the reconstructions are high quality, I think this paper is still missing an experiment which can show us something we didn't know before about a learned neural network representation. Leveraging these visualizations to perform some downstream task (e.g. if a human is able to detect whether or not the network will get the correct answer on an adversarially perturbed input) would help to strengthen the argument that thsese visualizations are useful.
>
> We highlight (more details in our general answer) that an important insight was gained from our study, that was not well known (or even thought to be the opposite to our findings) by the SSL community
> The representation at the output of the backbone is almost input equivariant i.e. no invariance is learned
> This result is critical for self-supervised learning since most of the training methods rely on learning invariances with respect to hand-crafted data augmentations. Our paper highlights a major failure of current self-supervised models by showing that their recent success is not due to learning better invariant representations by a careful choice of training criteria nor data augmentation, but by using a cheap trick (the use of a projector) that allows the backbone representation to retain as much as possible information about the inputs (despite pushing for learning invariance at the projector level).  Having a tool like RCDM is unlocking a path towards the design of better self-supervised models by visualizing how successful the SSL training is in learning invariant representations that can be useful for downstream tasks.
> Another major gain obtained with RCDM is to be able to probe visually how successful a given representation will be on a downstream task. Instead of having to train a linear classifier or a KNN over the entire training set of ImageNet, one could just visualize the representation and determine visually whether or not these representations contain the information needed to solve a given downstream task. For example in Figure 3, when we visualize the representation at the projector level with Dino, we observe that some background information is maintained while important information about the baby kangaroo shape is lost. This kind of representation is really detrimental when doing classification since the information that characterizes the class is not preserved. This result is important because instead of having to probe an SSL model on many different downstream tasks, one could just train RCDM over these representations and from qualitative visual inspection deduce which downstream tasks are the most aligned with the information that is preserved in the representation.

---

> > ### Author Response · Authors · 2022-05-24
> > **Answer to Reviewer W3ik**
> >
> > > “Baselines and discussion of related work: there is tons of work on visualizing representations of DNNs, but there are no alternative methods presented or compared to in this paper. Could all the qualitative insights have come about from a visualization method that produced less realistic inversions? How do these other approaches compare in your setting? Appendix A is a start at evaluating this quesiton, but ther eis a huge gap between the quality of the baselines in Appendix A and published results in this area.
> >
> > > ​​Baselines and discussion of related work: the related work needs to be expanded to cover the work of Dosovitsky & Brox, Mahendran & Vedaldi, Nguyen et al., and the DeepDream/feature viz work of Olah et al at a minimum. It would be great if there were additional comparisons with these approaches to highlight that they cannot yield the same qualitative conclusions as your proposed method (e.g. around invariance to vertical shift but sensitivity to scale). I also feel an explanation or further experiments to highlight why the baselines are so bad in Appendix A is critical.
> >
> > > ”I believe the characterization of prior work that inverts DNNs with DIP is inaccurate, at least looking at Zhao et al. (2021), they minimize d(f(x), f(g(w))) where w are the weights of the generator and the loss is in feature space. Qualitatively your results in Appendix A are far worse than Figure 2 of Zhao et al. (2021). Any ideas what is leading to this discrepancy?”
> >
> > > A.3: prior work has found that randomc rops + jittering can be very important for the quality of results. Did you try that here as well? Having seen the quality of earlier results (e.g. DeepDream, Mahendran &Vedaldi), I’m surprised how bad the results are in Fig 8.
> >
> > > “Mahendran & Vedaldi (2015, 2016): motivates the notion of natural pre-images, constructs a careful set of regularizers to yield good inversions that are more natural (e.g. jittering, smoothness, texture/style regularizer). There’s no discussion of DeepDream (2015): https://en.wikipedia.org/wiki/DeepDream and Olah et al.’s work on feature visualization and activation atlases (e.g. https://distill.pub/2017/feature-visualization/), this Distill paper has a great table under “The Spectrum of Regularization” that mentions several other papers that are not discussed here, e.g. Nguyen et al. 2016a that uses a GAN generator to synthesize preferred inputs (but does not draw samples from the conditional, instead they optimize directly for the latent), Nguyen et al. 2016b that runs MCMC with conditional guidance based on features. All of these approaches that optimize in image space with extra regularizers can be used to generate a diverse set of natural pre-images that could then be evaluated and compared against.”
> >
> > We think there might be a possible misunderstanding here (Please correct us if we are wrong). We used an entirely unconstrained optimization process in Appendix A. We just minimized the mean squared error between two representations by doing small, unconstrained and unregularized gradient steps in the input space, there is nothing else. The main point of this section was just to say that “If you want to get useful visualizations, you need to impose a more natural structure using some kind of prior, regularizer, or constraint” (As noted in https://distill.pub/2017/feature-visualization/). So, to answer your question, the results were bad in Appendix A because there was no prior, no regularizer and no constraint in the optimization process we presented. The work you are citing are all using some sort of regularization/constraint or prior which explains why their results are better.

---

> > > ### Author Response · Authors · 2022-05-24
> > > **Answer to Reviewer W3ik**
> > >
> > > We thank you for the references you provided, we updated the related work accordingly. We agree that interpretability is an important research area and doing proper comparison with other methods could be useful. However writing a interpretability paper, would require analyzing our model under different experimental setups outside SSL. For example some interpretability methods you cited analyze what a neural network learns at different layers. Some others try to visualize what a specific neuron has learned to detect (DeepDream, Nguyen et al. 2016a). All the methods you mentioned are doing some kind of optimization in the image space, but there are major differences as to what objective they are optimizing and what task they want to solve. It is not straightforward to compare directly a method designed to optimize a specific neuron activation with methods that minimize a distance between two large continuous vectors. It is also not straightforward to compare a method designed for visualizing features at different intermediates layers with a method that is designed and optimized for class label conditioning generative modeling. Doing a proper comparison of all of these methods when optimizing the same objective towards a given task could be really interesting but it should be a work on its own. Nevertheless, since Zhao et al. (2021) provides a visual analysis of the representations learned with a self-supervised model by using DIP (thus using a setup very close to ours), we added new Figure 7 in the appendix that makes a direct comparison between RCDM and DIP.
> > >
> > > > An additional challenge is that your conditional diffusion models will have to be retrained for every representation. As another idea, I believe you could do classifier guidance in the diffusion context by combining an unconditional p(x) model with a conditional p(h | x) model using something like log p(h|x) = ||f(x) - h_target||^2, i.e. you can use classifier guidance to guide your unconditional sample to have a similar representation as your target without retraining a diffusion model.
> > >
> > > It's an interesting idea, thanks ! It’s true that having to retrain the model each time is a limitation of the approach we presented. However, the conditioning by being done inside the model should be much faster when sampling than the methods you are suggesting that requires to compute a gradient at each step of the sampling chain. Another point is that classifier guidance requires training the classifier on the inputs generated by the diffusion process. Unfortunately, most self-supervised learning models are extremely costly to train and it’s not clear how much these noisy inputs will affect the training and performances of the self-supervised model. Our goal was to be able to take checkpoints of already trained SSL models and use their representations directly without having to go through a costly retraining.
> > >
> > > > “There’s no discussion of how the images that are used to train the conditional diffusion model will bias the inversions. Are failures on OOD inputs due to the representation failing on OOD inputs or the visualization failing?”
> > >
> > > > “In terms of not memorizing, Figure 2 & Figure 17 are great. The argument in Figure (b) that the samples are “close visually” I don’t find so convincing. In particular the contrast of the diffusion models is often very different from the original, and for the semantic segmentation input, changing the color of a pixel corresponds to changing the class label, and so the resulting samples I’d think about as very different even if to our eye they have somewhat similar characteristics. For example, semantic segmentaiton masks on roads will never have splotches of other texture Fig2b, right, and if you trained a diffusion model on segmentation masks it likely would not generate images like this. IMO, Fig 2b highlighted that we should not expect these visualization methods to work well for really OOD inputs (like segmentation masks)”
> > >
> > > This is a really interesting question ! To answer it, we trained RCDM on the segmentation mask of cityscapes (while using the representation of an SSL model trained on ImageNet). When using it conditionally on a segmentation mask , we don’t observe any failure in the reconstruction. Thus, the failures on OOD inputs, in this instance, are due to the visualizations that are failing and not the representation. We added Figure 13 in the Appendix which describes this experiment. We also removed the statement that OOD samples are close visually, in Fig 2b.

---

> > > > ### Author Response · Authors · 2022-05-24
> > > > **Answer to Reviewer W3ik**
> > > >
> > > > > Appendix A: I don't think the argument that gradient-based methods only occupy a K-dimensional subspace is valid (see more comments below). An extended proof or discussion is needed. A.2 / Eqn 2: I don’t follow your logic here. The Jacobian of a deep network will change as the input is changed, and thus the space that the iterates span can be D-dimensional if you take many steps
> > > >
> > > > At each gradient step, the direction of the update applied to the input-space image will consist of a linear combination of the model’s Jacobian matrix row which is a K x D matrix, and is thus constrained to live within a K-dimensional subspace of the data space (D-dimensional). That being said, it is clear that this only holds for a single update; for repeated updates, the K-dimensional subspaces can freely vary (since the Jacobian matrix of the model depends on where is the current input, and would only stay constant through the space if the model was linear) and thus overall produce updates that together span the entire D-dimensional space.
> > > >
> > > > > “Do you actually use batchnorm and introduce dependences between elements in the batch or use conditional group norm where you modulate the scale and bias of each feature map?”
> > > >
> > > > As mentioned in Section 3, we use a conditional version of batch normalization, thus introducing dependencies between elements in the batch. However we can also use the conditioning built-in inside ADM which doesn’t require the use of batch normalization (as shown by Figure 10).
> > > >
> > > > > “You could use an architecture closer to the super-res based approach by having a neural network that outptus a representation-dependent bias in only the first layer (corresponding to broadcasting the vector spatially after a transformation). I believe your approach is a generalization of this, but I’d be interested to see baselines of how different forms of conditioning impact the results.”
> > > >
> > > > We didn’t really see any differences between the conditional batch normalization conditioning and the ADM built-in conditioning (In Figure 10) so, it’s not clear that this other type of conditioning will achieve anything more than the one we already tried.
> > > > > “Need to mention what dataset you are training the diffusion model on. ImageNet?”
> > > >
> > > > We mention it in the experiment part that we have now merged with Section 3 in the updated version of the paper.
> > > >
> > > > > Fig 4 is a neat demonstration of how different heads and methods constrain the pre-iamge differently! It would be interesting to pair this with a supervised model to see whether the small differences in the representations of the sampled inputs confuse a classifier that is trained on the learned representation.
> > > >
> > > > It’s a great idea actually, however we unfortunately did not have the time to do it for this rebuttal.
> > > >
> > > > > Fig 6: it’d be useful to note where the classifier fails on the adversarial example in each row (e.g. by having a box around each image with a color red if it’s incorrect). I think there’s also a great opportunity for a behavioral study to show the utility of your method: see whether humans are better able to predict the classifier output by visualizing your reconstructions instead of the original adversarially perturbed input.
> > > >
> > > > Thanks, we updated the figure according to your suggestion!
> > > >
> > > > > “Fig 7: this was an interesting idea and set of manipulations but I dont’ find the results very convincing. It does seem like this method is good at plugging in different backgrounds, but the clothes (to my eye) in the bottom row don’t match well with the initial set of clothes and the dogs in that image are of poorer quality than most your samples. I think there may be more convincing sets of manipulations in latent space that could highlight this method, e.g. taking an average of a bunch of dog images with clothes, and dogs without, and learning a single vector to add in latent space.”
> > > >
> > > > We tried this actually ! But we didn’t have any success in finding a single vector in latent (representation space) that matches a meaningful transformation in the input space. Current SSL representation vectors seem still very entangled. There are some cases as shown in Fig 7 where we can observe very locally some structure but they don’t generalize very well across different images. The purpose of this paper is to showcase how to use a specific tool, so if SSL researchers developed a SSL criteria that constrains the representation to have a given structure, doing these kinds of experiments will be really useful to evaluate them.

---

> > > > > ### Comment · Reviewer_W3ik · 2022-06-29
> > > > > **response**
> > > > >
> > > > > Thank you to the authors for the thorough rebuttal, new experiments, and updated paper. Having read the other reviews, I maintain my concern that there is not enough utility shown of the visualization method for probing SSL representations and am leaning towards rejecting the paper in its current version. Given the complexity of training a diffusion model on representations, I feel that such a method may not be of widespread general use to the ML audience. However, this paper does present an interesting tool that may lead to discoveries and new understandings of representations in the future.
> > > > >
> > > > > The main argument for the utility of the diffusion-based visualization method (claim 2) lies in the finding that the backbone outputs are not invariant. However, that same finding was discussed and presented in SimCLR (Table 3) where they show how transformations can be predicted from the backbone with reasonable fidelity but not from the projector output. The proposed visualization method provides another way of identifying this, but I do not agree with the assertion that this is a widely held belief which this paper corrects. If the main difference is that the diffusion-based visualization tool can access all information (not just the linearly accessible bits), then you could train a nonlinear readout to evaluate whether the backbone or projector was able to predict the transformation.
> > > > >
> > > > > > Instead of having to train a linear classifier or a KNN over the entire training set of ImageNet, one could just visualize the representation and determine visually whether or not these representations contain the information needed to solve a given downstream task
> > > > >
> > > > > The visualization approach requires training a diffusion model on ImageNet, which will be far more costly than training a probe or running kNN. The complexitwy of training many probes will still be cheaper than training a diffusion model. I do agree that the method presented here could be useful in the future for probing representations, but I am not convinced that there has been anything discovered in this paper about existing representations that was not previously known or could not be identified without this tool.
> > > > >
> > > > > The new Figure 7 comparing with DIP is great, but this single example doesn’t really show whether you could not assess “supervised representation is invariant to background” without DIP. The main reason for asking about APpendix A / comparison to other visualization methods was to evaluate whether they could be used instead to answer the same questions you ask here. And it's not clear that DIP would not, in particular you can see how DIP shows that the backbone preserves far more information about the inputs than the projection head.

---

> > > > > > ### Author Response · Authors · 2022-06-29
> > > > > > **Answer to Reviewer W3ik**
> > > > > >
> > > > > > Thank you for your response.
> > > > > >
> > > > > > > Thank you to the authors for the thorough rebuttal, new experiments, and updated paper. Having read the other reviews, I maintain my concern that there is not enough utility shown of the visualization method for probing SSL representations and am leaning towards rejecting the paper in its current version. Given the complexity of training a diffusion model on representations, I feel that such a method may not be of widespread general use to the ML audience.
> > > > > >
> > > > > > Training this model is not as difficult as you seem to think. In terms of code, it is only a matter of loading the weights of the model used as conditioning. We can also use low resolution (64x64) to speed up the training. By doing that, in less than two days of training (on 4 gpu), one already gets informative visualizations. Also the perceived complexity of a novel and superior or complementary imaging tool is not a good argument to play down its potential usefulness (otherwise medical imaging would be content with x-rays and would not have also embraced cumbersome magnetic resonance imaging).
> > > > > >
> > > > > > > The main argument for the utility of the diffusion-based visualization method (claim 2) lies in the finding that the backbone outputs are not invariant. ... The complexity of training many probes will still be cheaper than training a diffusion model. I do agree that the method presented here could be useful in the future for probing representations, but I am not convinced that there has been anything discovered in this paper about existing representations that was not previously known or could not be identified without this tool.
> > > > > >
> > > > > > Your argument relies on the fact that training a linear (or non linear) readout to probe the representations will be cheaper than training a diffusion model. It is true but:
> > > > > >
> > > > > > 1)  One only needs to train the diffusion model once whereas if you use linear/non linear readout to probe the representation, you’ll need to train them on each task you want to probe your representation of.
> > > > > > 2) When using these readouts, you need to have access to labels which might not be available. So if you want to probe the representation to see if it contains information about the background, you won’t be able to do it since the background label information is not available on ImageNet.
> > > > > > 3) One might not think in advance of the entire space of factors of variation it might be useful to probe. Supervised probing requires one to know ahead of time what they are looking for: it can serve to verify a hypothesis, but is hardly a tool for discovery.
> > > > > > So, when probing a representation on different modalities, it might be less time consuming to use the diffusion model  than asking a team of people to manually label data in order to train linear/non linear readouts on a different number of tasks.
> > > > > >
> > > > > > > The new Figure 7 comparing with DIP is great, but this single example doesn’t really show whether you could not assess “supervised representation is invariant to background” without DIP. ... it's not clear that DIP would not, in particular you can see how DIP shows that the backbone preserves far more information about the inputs than the projection head.
> > > > > >
> > > > > > The main point is that the quality of the images obtained with DIP can be bad enough for it to be much harder to assess what information is and isn’t contained in a representation. If you look at examples obtained with the supervised model we find it extremely hard to conclude anything from DIP generated images, whereas with our diffusion model the invariance to background is very obvious. However if we focus only on the backbone vs projector matter, it is true that we can get a similar conclusion with DIP (which is also a good thing since it also confirms one of our claims).
> > > > > >
> > > > > > It seems that your main concern about the paper is that we weren’t able to show unique and novel insights gained by using the diffusion method, that we couldn’t also get by other means ((linear) readout/DIP etc..). Of course other methods may be able to similarly probe what is or isn’t retained in a representation. But RCDM is currently the visualization method that yields superior image quality, enabling easier interpretation. It seems unfair to trivialize the contribution because similar insights might be obtained by other means, which have different limitations (as we just mentioned). Now it was important to show that we can reach the same conclusions with other methods, since it's proof that our method actually does work !
> > > > > >
> > > > > > So the remaining question is whether our diffusion method is useful for the research community as an additional tool to probe representation, whether it advances the state-of-the-art in this. Based on the arguments written above, we think that it is clearly a useful tool – superior on several aspects to what had been previously used – . And in our paper we showcased many diverse ways it could be used imaginatively, paving the way for potentially exciting research directions.

---

### Author Response · Authors · 2022-05-13
**General answer to the reviews**

We would like to thank all the reviewers for carefully reading our paper and providing insightful comments and useful suggestions. We are working on improving our work based on these, and will update the paper soon. We are planning to answer each reviewer's concerns individually in the coming days along with the updated paper; however, to answer the main concerns, we wanted to start the discussion as soon as possible.

For now, we would like to clarify the main contribution and claims of our work:

The main **purpose of this paper is to advocate and showcase the usefulness of conditional generative models to probe and gain insights about properties of self-supervised-learned (SSL) representations** of images. We strongly believe that if SSL practitioners and researchers added such visualizations to their workflow, it could help prevent making erroneous claims. Of course visualization is not sufficient on its own, but when used alongside numerical benchmarks, it can provide a much sharper understanding of what the algorithms are really learning. This is what our paper shows.

There remain many mysteries and misconceptions about the properties of the representations learned with current SSL methods, and our work makes an essential contribution in clarifying these. For example the belief that SSL representations are invariant to the data augmentations used during training (reinforced by broad statements in SSL papers such as  “[SSL] approach makes the representations to become invariant under transformation applied to the images. It was shown that the quality of the learned representations benefits from such a form of pretraining.” [1])  While it is true that SSL training criteria push to learn invariances, it is definitely not true (with the current frameworks) that the representations that are commonly used for downstream tasks (i.e the one at the backbone level) are indeed invariant to these transformations, as we clearly show visually in our paper.

We also need to emphasize that our paper is **not about proposing a fundamentally novel or superior conditional generative modeling or interpretability method**. Thus, we **do not claim that this conditional diffusion model is the best one could imagine to get high fidelity visualization**, and we **do not claim that this specific visualization method is the best that everyone should use for interpretability**. Some reviewers highlighted that having a paper in between many worlds (Interpretability/Conditional generative mode/SSL) is a weakness. But we think that sometimes there is a need to create bridges between different research domains and toolings. The aim of this paper is to raise awareness in the self-supervised community about how a visualization tool (that we built on recent published advances in conditional generative models, with a simple change), can usefully help in analyzing and designing SSL methods, thus complementing the traditional numerical benchmarks. In order to demonstrate the usefulness of this tool for the community and its scientific value, we needed to show that **samples produced by such techniques are faithful to the conditioning representation (claim 1)**, and to showcase how **it can be used to probe SSL representations in novel ways to gain clear insights (claim 2)**. If reviewers see any other claim in this paper unrelated to claim 1 or claim 2 or if you think the provided evidence is insufficient to back claim 1 or claim 2, please let us know.
Since TMLR puts an emphasis on

> Are the claims made in the submission supported by accurate, convincing and clear evidence?

We want to highlight the evidence that supports claim 1 and 2.

Evidence in support of Claim 1:
- Computing the rank of the conditioning representation within the set of nearest neighbors of the model’s sample representation in Figure 18. A rank of 1 implies that all generated samples have their representation that is the closest to the conditioning representation used to generate these exemples.
- Computing the distances between the representations of the generated images with the representations of different transformations applied on the images used for conditioning. This shows that for whatever transformations we applied, the generated images are still the closest ones to the representation.
- Visualization of the model’s samples with respect to a given conditioning Fig . 2, 12, 13. We show that in different regimes (In distribution and Out of distribution), the model is able to reproduce the main features of the image whose representation was used as conditioning.
- Visualization of the model’s samples with respect to an interpolation between different conditioning Fig 29,30. This shows that such a model is able to generalize to interpolations at the representation level, which is evidence that the model is not just overfitting its training the data.

---

> ### Author Response · Authors · 2022-05-13
> **General answer to the reviews [2/2]**
>
> Evidence in support of Claim 2:
> - In Figures 4, 25, 26, we show that RCDM can be used to visualize differences in learned representations. With such visualizations, we are able to show that the representations at the projector level are much more invariant than the ones at the backbone level which suggests why ssl practitioners are not able to use the representation at the projector level to obtain the best performance on downstream tasks. This is an important result because most papers in SSL have reported accuracy results at the projector and representation level with a linear probe, however such experiments don't reveal whether the information was removed from the representation or if it’s merely too entangled to be easily extracted by a linear probe.
> - In Figures 27,28, we show that RCDM can also be used to verify how much a representation is invariant to a given data augmentation. This point is crucial since self-supervised learning relies nowadays on handcrafted augmentation. There is also a misconception that SSL representations are invariant to the data augmentations that were used during training. Our results show that this is clearly not the case. Our paper shows that SSL representations contain much more information about their inputs than what is believed in the community.
> - We show visually in Figure 6 that representations learned with self-supervised methods appear to be more robust to adversarial examples than the ones obtained through supervised learning. This opens a really interesting research direction that we hope researchers in self-supervised learning will explore.
> - We showcase in Figure 7 how RCDM can be used to reveal how a self-supervised representation is structured. We strongly believe that a really interesting research direction would be to devise SSL criteria that impose some structure at the representational level. RCDM will be a major tool to evaluate if such structures are present inside the representation.
>
> We would like to re-emphasize that the above results will be useful for the community as a whole since recent papers base their work/research on unverified assumptions (representations learned with SSL are invariant to augmentations) that we are now in the capacity to debunk, hence we strongly believe that we fulfill one of the fundamental TMLR requirement:
>
> > Would at least some individuals in TMLR's audience be interested in knowing the findings of this paper?
>
> Lastly, evaluating what is contained in a learned representation is hard. If one wanted to have a numerical benchmark, one could use a dataset labeled with known factors of variations to be able to check whether information about these is contained or not in the representation. It is possible to create an artificial or small real controlled dataset with a limited number of factors of variations to evaluate the invariances or equivariances of representations to these. However when considering natural images, it is near impossible to label every conceivable factor of variation present in them. So when researchers are writing papers with experiments on toy dataset with controlled factors of variation, they can in addition use ImageNet with RCDM, to probe whether the same invariances and equivariances translate to natural images. This, without the need to create an expensive specifically labeled natural image dataset. This ability is really critical for researchers to be able to evaluate structured representations. Crucially, where careful numerical benchmarking may allow to probe for predefined factors, conditional generation has *the potential to reveal the presence in the representation of unexpected factors*, that researchers might not have thought of beforehand.
>
> [1] https://www.mdpi.com/2079-9292/9/11/1930/pdf

---

### Author Response · Authors · 2022-05-24
**Update of the paper**

We would like to thank the reviewers again for their valuable recommendations which have helped clarify and strengthen our work.
We updated the main paper accordingly and list the changes below:
- We clarified our contributions in the abstract and introduction (i) the fact that we do not claim to introduce a new generative model but merely showcase the use of diffusion models for interpretability of SSL (ii) that we obtain novel insights regarding SSL representations.
- We added clear definitions related to self-supervised learning and its terminology, in particular the distinction between the backbone network and projector/head network, and the representations associated with these two levels.
- We added references related to studies on interpretability and visualization of representations while contrasting their approach with ours e.g. contrasting the generation of DIP based solutions that were commonly used to gain insight about SSL representations, with an example in the newly added Fig. 7.
- We merged Sections 3 and 4, to reduce the focus on the generative model in favor of insights gained into SSL models. The section title was changed accordingly. We also moved the schematic figure explaining the method to the appendix.
- We emphasized that our conditional batch normalization variant of OpenAI diffusion model was motivated by the  goal to compare with IC-GAN. However, one can instead use the built-in conditioning of ADM as illustrated in the newly added Figure 10. We also added a small paragraph (Section 3) to explain why Classifier Guidance is not suitable for visualization of SSL representations (As it requires retraining each SSL model on the input diffusion process).
- We added a small paragraph (Section 4) explaining how our visualization method can be used to predict the downstream performances of a set of (SSL) representations by comparing the invariances that appear present in the representation to the ones suited for the downstream task.
- We removed the claim that samples remain close visually in OOD.
- We added a red square around the adversarial example that is fooling a trained classifier in Figure 4, to visually support the conclusion that SSL and supervised methods see adversarial examples differently.
- We updated Section titles to clarify the gained insights.
- We added the broader impact statement at the end.

Concerning the Appendix, we made the following updates:
- We removed Appendix A that dealt with unconstrained gradient optimization as this result is widely known in the community e.g. in https://distill.pub/2017/feature-visualization/. This point also wasn’t directly aligned with our main contributions.
Since Zhao et al. (2021) tried to visualize SSL representations with DIP, we added Figure 7 that directly compares RCDM and DIP, highlighting the advantage of RCDM to obtain high-quality samples that enable drawing precise visual conclusions on the learned invariants.
- We added Figure 10 that compares different types of conditioning for RCDM (Cond. Batch Norm and ADM built-in conditioning)
- We added Figure 11 that investigates the failure cases in the OOD scenario (with segmentation masks from cityscapes). We trained RCDM with segmentation masks from cityscape while the SSL model (Dino) is trained on ImageNet. We observe that the segmentation samples are faithful to the conditioning, meaning that the previous failures in OOD generation were due to what the generative model had been trained on and not to the SSL representation.
- Finally, we added Figure 28 emphasizing that our conclusions concerning the ssl representations (e.g. nearly no invariants at the backbone level and a large number of invariants at the projector level) holds for other datasets than Imagenet.

We hope that the above changes answer the reviewers’ concerns, and we will be happy to answer any further questions that the reviewers and action editor might have.

---

### Decision · Action_Editors · 2022-07-03

**Recommendation:** Accept with minor revision

**Comment:**

Many thanks to the authors for working with the reviewers on getting extra experiments and clarifications into the subsequent versions of this work.

The main point of contention in the reviews is regarding the claim from the submission, that the insights uncovered by the visualization tool are good for a downstream task/experiment. I appreciate that the authors added some figures & experiments showing that in some circumstances the visualization tool can show whether the representation will be useful (for classification) or not. I am not necessarily convinced that using RCDM is in fact "easier" than training kNN (as the authors claim in their comments).

Taking a look at Figures 12 and 13, it is hard to know whether the conclusions spelled out by the authors are cherry-picked based on those samples, even if the discussion at the end of Section 4 suggests there are some metrics to bolster those claims.

So, in rendering my decision, I am guiding myself on the two TMLR policies:

1. "Are the claims made in the submission supported by accurate, convincing and clear evidence?": It is clear that the claim about the usefulness of the visualization tool towards a downstream task is not believed by the reviewers to be very supported by accurate, convincing and clear evidence. But the authors have shown some good-faith attempts at clearing this up.

2. "Would at least some individuals in TMLR's audience be interested in knowing the findings of this paper?": I believe a non-trivial number of TMLR readers would be interested in knowing the findings of this paper.

For this to be accepted as is, I would want the authors to refine the experiments in figures 12 & 13 and the discussion at the end of Section 4:

  * At the very least, I would like to see a few more numbers and analysis. More specifically, I'm concerned that the Dino linear probing result on page 6 may not generalize at all to other models, so it'd be good to see more data on this (ideally with as many encoders as the authors used for the other experiments). I think this could go a long way towards understanding the extent to which the proposed method is "useful".
  * If those results don't quite generalize beyond Dino, I think the paper can still be accepted, provided the authors downplay the claims about "usefulness" (which were the most contentious with the reviewers).

Beyond that, I think the authors have done a good job at answering the reviewers' concerns.

---

> ### Author Response · Authors · 2022-08-09
> **Thank you**
>
> We are grateful to the decision and for all the reviewers’ recommendations during the entire review process. In order to make our camera ready as complete as possible, we have implemented the following updates:
>
> First, we have updated Figure 2 as well as Section 4 to highlight the correlation between the downstream classification performances evaluated across all of our models and the samples produced by RCDM. We hope that the new Figure 2 on which we added a Table with the various linear probe evaluation results will help in bolstering our claim that visual assessment of the generated samples correlates with downstream task performance.
>
> Second, we improved Figure 13 (which is now Figure 12)  by adding additional models to be sure that none of the presented insights are model specific unless pointed out explicitly.
>
> Third, we removed Figure 12 to avoid overextending the RCDM interpretation since Figure 2 and 11 already establish that there exists a correlation between the performance in the classification downstream and the samples generated when looking at the differences between the backbone versus the projector. Removing this figure allows us to downplay our claim on this point that was controversial during the review process.
>
> We also added the link to the code and the trained models.
>
> Please let us know if there are any remaining concerns.